# LANGUAGE MODELS ARE INJECTIVE AND HENCE INVERTIBLE

**Giorgos Nikolaou**[1,5,*]  **Tommaso Mencattini**[1,2,*]

**Donato Crisostomi**[2]   **Andrea Santilli**[2]   **Yannis Panagakis**[4,5]   **Emanuele Rodolà**[2,3]

[1]EPFL   [2]Sapienza University of Rome   [3]Paradigma
[4]University of Athens   [5]Archimedes/Athena RC, Greece

## ABSTRACT

Transformer components such as non-linear activations and normalization are inherently non-injective, suggesting that different inputs could map to the same output and prevent exact recovery of the input from a model's representations. In this paper, we challenge this view. First, we prove mathematically that transformer language models mapping discrete input sequences to their corresponding sequence of continuous representations are injective and therefore lossless, a property established at initialization and preserved during training. Second, we confirm this result empirically through billions of collision tests on six state-of-the-art language models, and observe no collisions. Third, we operationalize injectivity: we introduce SIPIT, the first algorithm that provably and efficiently reconstructs the **exact** input text from hidden activations, establishing linear-time guarantees and demonstrating exact invertibility in practice. Overall, our work establishes injectivity as a fundamental and exploitable property of language models, with direct implications for transparency, interpretability, and safe deployment.

## 1   INTRODUCTION

A core question in understanding large language models is whether their internal representations faithfully preserve the information in their inputs. Since Transformer architectures rely heavily on non-linearities, normalization, and many-to-one attention mechanisms, it is often assumed that they discard information: different inputs could collapse to the same hidden state, making exact recovery of the input impossible. This view motivates concerns around transparency, robustness, and safe deployment, as it suggests that the link between text and representation is inherently *lossy*.

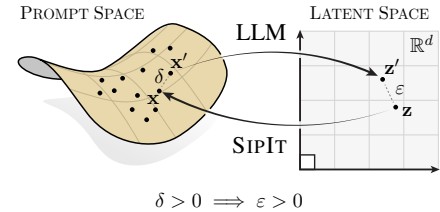

Figure 1: The map from prompts to latent space is injective. SIPIT inverts it.

In this paper, we show that this intuition is misleading. Despite their apparent complexity, standard decoder-only Transformer language models (seen as maps from prompts to hidden states) are in fact **almost-surely injective**; for essentially all parameter settings and during the course of training, different prompts yield different last-token representations (e.g., see Figure 1).

Building upon this property, we further provide a practical algorithm, SIPIT, that reconstructs the *exact* input from hidden activations. To our knowledge, it is the first to guarantee exact recovery in provable linear time (worst case bound), often faster in practice, turning injectivity from a theoretical property into an operational tool.

**Our approach.**   To establish our result, we take a rigorous mathematical view of Transformers as functions. The key idea is that their components (embeddings, LayerNorm, causal attention,

---

*Equal contribution; author order settled via Mario Kart.

Corresponding authors: {georgios.nikolaou,tommaso.mencattini}@epfl.ch

MLPs, and residual wiring) are smooth and structured enough that the model, as a whole, behaves predictably with respect to its parameters. Using tools from real analysis, we show that collisions (two different prompts producing the exact same representation) can only occur on a set of parameter values that has measure zero; that is, they are mathematical *exceptions* rather than possibilities one should expect in practice. Moreover, we prove that common training procedures (gradient descent with standard step sizes) never move parameters into this exceptional set. In layman's terms, almost all models at initialization are injective, and training preserves this property.

Technically, our proofs rely on two ingredients. First, we establish that Transformers are real-analytic functions of their parameters, which allows us to reason precisely about when and where collisions could occur. Second, we construct parameter settings where no two prompts collide, and show that gradient descent (GD) does not collapse such separation, i.e., collisions remain a measure-zero event. The end result is a finite-horizon *guarantee*: after any fixed number of training steps, and under mild assumptions, injectivity holds with probability one. We provide complete formal proofs of these statements.

**Main result.** Our central finding is that causal decoder-only Transformer language models are injective almost surely. Formally, consider one such model with embedding width $d$, at least one attention head per block, real-analytic components, finite vocabulary $\mathcal{V}$, and finite context length $K$. Initialize its parameters $\boldsymbol{\theta}$ at random, using any distribution that has a density[1] (such as Gaussian, uniform, or Xavier/Glorot), and train for any finite number $T$ of GD steps with step sizes in $(0, 1)$. Then, with probability one over the random initialization,

$$\mathrm{s} \neq \mathrm{s}' \implies \mathbf{r}(\mathrm{s}\,;\boldsymbol{\theta}_T) \neq \mathbf{r}(\mathrm{s}'\,;\boldsymbol{\theta}_T),$$

i.e., the map from prompts $\mathrm{s}$ to *last-token* representations $\mathbf{r}(\mathrm{s}\,;\boldsymbol{\theta}_T)$ is injective across all prompts in $\mathcal{V}^{\leq K}$. In short, collisions in practical settings form a measure-zero set, and neither initialization nor training will ever place a model inside that set.

**Significance.** Our result shows that in standard decoder-only Transformers, different prompts almost surely yield different last-token representations across all practically relevant parameter settings and training procedures. The guarantee is both *generic* (it fails only on a measure-zero set of pathological parameters) and *practical* (it holds at finite width, depth, and training time under common initializations).

Conceptually, we replace a long-assumed property with a rigorous theorem, showing that injectivity is not an asymptotic idealization but a structural consequence of the architecture itself. Technically, our analytic framework pinpoints when collisions can arise (through deliberate non-analytic choices such as quantization or tying), and clarifies that otherwise the model is inherently lossless. Importantly, it establishes that last-token states almost everywhere *identify* the input.

Finally, we turn this theoretical guarantee into an operational tool: our algorithm SIPIT uses gradient-based reconstruction to recover prompts *exactly* from internal activations, efficiently and with provable *linear-time* guarantees. This confirms empirically that collisions do not occur in practice. Beyond transparency and safety, this elevates *invertibility* to a first-class property of Transformer language models, enabling stronger interpretability, probing, and causal analyses.

## 2 TRANSFORMERS ARE INJECTIVE

**Summary.** In this section we show that decoder-only Transformers almost surely map different prompts to different hidden states. Collisions can only occur under measure-zero parameter choices, and gradient-based training never creates them. In simple terms, Transformer representations are structurally lossless.

**Approach.** We consider causal decoder-only Transformer language models with vocabulary $\mathcal{V}$, finite context window $K$, and embedding dimension $d$. For an input sequence $\mathrm{s} \in \mathcal{V}^{\leq K}$, let $\mathbf{r}(\mathrm{s}\,;\boldsymbol{\theta})$ denote the final hidden representation at the *last* token position[2], given parameters $\boldsymbol{\theta}$.

---

[1] Put simply, parameters are not drawn from a degenerate or hand-crafted set.

[2] We focus on the last-token state, since it alone drives next-token prediction; earlier rows matter only insofar as they shape this final state. Injectivity at the last token is the property of real operational interest.

Our analysis relies on three facts:

(i) *Real-analyticity.* Each component of the architecture (embeddings, positional encodings, LayerNorm with $\varepsilon > 0$, causal attention, MLPs with analytic activations, residuals) is real-analytic in its parameters (see Appendix A.2 for the mathematical background). This smoothness implies that the set of parameter values causing two distinct prompts to collide is extremely thin (measure zero).

(ii) *Initialization.* Standard initialization schemes (Gaussian, uniform, Xavier/Glorot, etc.) draw parameters from continuous distributions with densities, so they avoid measure-zero sets with probability one.

(iii) *Training.* Gradient-based updates (including SGD and mini-batch/full-batch GD) preserve absolute continuity of the parameter distribution after any finite number of steps; thus, training cannot generate collisions.

These facts allow us to state and prove injectivity results without relying on asymptotics. We begin by establishing the analytic structure of the architecture.

**Theorem 2.1** (Transformers are real-analytic). *Fix embedding dimension $d$ and context length $K$. Assume the MLP activation is real-analytic (e.g. tanh, GELU). Then for every input sequence $\mathrm{s} \in \mathcal{V}^{\leq K}$, the map*

$$(\mathrm{s}, \boldsymbol{\theta}) \mapsto \mathbf{r}(\mathrm{s} \,;\, \boldsymbol{\theta}) \in \mathbb{R}^d \tag{1}$$

*is real-analytic jointly in the parameters $\boldsymbol{\theta}$ and the input embeddings.*

*Sketch of proof (full proof in Appendix B, Proposition B.3).* Each building block is real-analytic: polynomials (embeddings, projections), exponential and softmax (attention), reciprocal square root (LayerNorm with $\varepsilon > 0$), analytic activations in the MLP, and affine maps. Real-analytic functions are closed under addition, multiplication, quotient, and composition. Since the Transformer is a finite composition of such blocks, the entire map is real-analytic. □

This smoothness result drives everything that follows: it ensures that collisions, if they exist, are confined to measure-zero parameter sets. We now ask: what happens at initialization?

**Theorem 2.2** (Almost-sure injectivity at initialization). *Let $\boldsymbol{\theta}$ be drawn from any distribution with a density (e.g. Gaussian or uniform). Then for any two distinct prompts $\mathrm{s}, \mathrm{s}' \in \mathcal{V}^{\leq K}$,*

$$\Pr[\mathbf{r}(\mathrm{s} \,;\, \boldsymbol{\theta}) = \mathbf{r}(\mathrm{s}' \,;\, \boldsymbol{\theta})] = 0 \,. \tag{2}$$

Figure 2: Two real-analytic functions $f_1$ and $f_2$ and their difference $f_1 - f_2$. Black contours show the zero sets, which form thin curves (measure zero) rather than regions of positive measure.

*Sketch of proof (full proof in Appendix C, Theorem C.2).* Fix $\mathrm{s} \neq \mathrm{s}'$ and consider

$$h(\boldsymbol{\theta}) = \|\mathbf{r}(\mathrm{s} \,;\, \boldsymbol{\theta}) - \mathbf{r}(\mathrm{s}' \,;\, \boldsymbol{\theta})\|_2^2 \,. \tag{3}$$

By Theorem 2.1, $h$ is real-analytic. A fundamental dichotomy of real-analytic functions states that either $h$ is identically zero, or its zero set has Lebesgue measure zero (see Figure 2 for an illustration). Therefore, to rule out the pathological case $h \equiv 0$ it suffices to exhibit a single parameter setting where $\mathbf{r}(\mathrm{s} \,;\, \boldsymbol{\theta}) \neq \mathbf{r}(\mathrm{s}' \,;\, \boldsymbol{\theta})$.

This can always be done: if $\mathrm{s}$ and $\mathrm{s}'$ differ at the last position (symbol or length), freeze the network so that the last state reduces to embedding plus position, and choose distinct rows; this already separates $\mathbf{r}(\mathrm{s})$ and $\mathbf{r}(\mathrm{s}')$. If instead they differ earlier, let $i^\star$ be the first mismatch and set one attention head so the last position attends almost entirely to $i^\star$, encoding its token in the value; this forces different outputs for $\mathrm{s}$ and $\mathrm{s}'$.

Hence $h$ is not identically zero, and so the collision set $\{\boldsymbol{\theta} : h(\boldsymbol{\theta}) = 0\}$ has Lebesgue measure zero. Since standard initializations have densities, the probability of sampling such $\boldsymbol{\theta}$ is zero, and $\mathbf{r}(\mathrm{s} \,;\, \boldsymbol{\theta}) \neq \mathbf{r}(\mathrm{s}' \,;\, \boldsymbol{\theta})$ (injectivity) holds almost surely at initialization. □

According to Theorem 2.2, at initialization, collisions are mathematically impossible except on a vanishingly small set of parameter values. Finally, with the following Theorem we ensure training does not break injectivity.

**Theorem 2.3** (Injectivity preserved under training). *Let $\boldsymbol{\theta}_0$ be initialized from a distribution with a density, and let $\boldsymbol{\theta}_T$ be the parameters after $T$ steps of gradient descent with step sizes in $(0,1)$. Then with probability one,*

$$\mathrm{s} \neq \mathrm{s}' \quad \Longrightarrow \quad \mathbf{r}(\mathrm{s}\,;\,\boldsymbol{\theta}_T) \neq \mathbf{r}(\mathrm{s}'\,;\,\boldsymbol{\theta}_T), \tag{4}$$

*Sketch of proof (full proof in Theorems C.1 and C.5).* At initialization, $\boldsymbol{\theta}_0$ is drawn from a distribution with a density, hence absolutely continuous. To break injectivity during training, GD would need to map this continuous law onto the measure-zero collision set identified in Theorem 2.2. We show this cannot happen.

A single GD step is the map $\phi(\boldsymbol{\theta}) = \boldsymbol{\theta} - \eta\nabla\mathcal{L}(\boldsymbol{\theta})$, where $\mathcal{L}$ is the training loss. Because the network and the softmax cross-entropy loss are real-analytic, $\phi$ is also real-analytic. Its Jacobian determinant $\det D\phi(\boldsymbol{\theta})$ is itself real-analytic and not identically zero (one can check this by evaluating at a simple parameter setting). Hence the set where $\det D\phi = 0$ has measure zero. Away from that set, the Inverse Function Theorem applies: $\phi$ is a smooth, locally invertible change of coordinates that can stretch or bend space but cannot collapse regions of positive volume onto lower-dimensional sets. Therefore, pushing forward an absolutely continuous distribution through $\phi$ yields another absolutely continuous distribution.

Since this argument holds for each step, any finite sequence of GD updates preserves absolute continuity of the parameter law. Combining with Theorem 2.2, which shows that collision sets are measure-zero, we conclude that $\mathbf{r}(\mathrm{s}\,;\,\boldsymbol{\theta}_T) \neq \mathbf{r}(\mathrm{s}'\,;\,\boldsymbol{\theta}_T)$ almost surely for all $\mathrm{s} \neq \mathrm{s}'$. $\square$

Thus injectivity is not just an initialization property but remains true throughout training. A simple but important corollary follows.

**Corollary 2.3.1** (SGD and mini-batch GD). *Under the assumptions of Theorem 2.3, the same conclusion holds when the updates are $\boldsymbol{\theta}_{t+1} = \boldsymbol{\theta}_t - \eta_t\,\nabla_\theta\mathcal{L}_{\mathcal{B}_t}(\boldsymbol{\theta}_t)$ with arbitrary (possibly random or adversarial) batch selections $\mathcal{B}_t$, thus including the singleton case of SGD and the full dataset.*

*Proof.* The proof argument of Theorem 2.3 is unchanged: for each fixed batch $\mathcal{B}$, the update map $\phi_{\mathcal{B}}(\boldsymbol{\theta}) = \boldsymbol{\theta} - \eta\nabla\mathcal{L}_{\mathcal{B}}(\boldsymbol{\theta})$ is real-analytic with a Jacobian that is not identically zero. Indeed, the batch loss is the average $\mathcal{L}_{\mathcal{B}} = \frac{1}{|\mathcal{B}|}\sum_{i=1}^{|\mathcal{B}|}\mathcal{L}_i$, so at the point $\boldsymbol{\theta}_\star$ from the single-sample proof (where the Jacobian determinant is sample-independent and nonzero) the batch Jacobian coincides with the single-sample one by linearity of differentiation, and its determinant is therefore also nonzero. Thus, the finite composition of such maps preserves absolute continuity of the parameter law. $\square$

Together with this robustness to different training regimes, we can also strengthen the guarantee itself: injectivity holds not just pairwise, but globally across finite sets of prompts.

**Corollary 2.3.2** (Distinctness for finite sets). *For any finite set of prompts $\mathcal{S} \subseteq \mathcal{V}^{\leq K}$, the representations $\{\mathbf{r}(\mathrm{s}\,;\,\boldsymbol{\theta}_T) : \mathrm{s} \in \mathcal{S}\}$ are almost surely all distinct.*

*Proof.* See Appendix C, Corollary C.2.1. $\square$

These results show that decoder-only Transformer language models are structurally injective: different prompts almost surely yield different last-token states. Collisions can be manufactured, e.g., through deliberate non-analytic choices (quantization, non-smooth activations), but in practical training pipelines, injectivity is guaranteed; extensive experiments in §4.1 confirm this empirically.

**Failure cases.** We showed that non-injective transformers are overwhelmingly unlikely, though it is still possible for an adversary to construct collisions by hand. For instance, if two vocabulary items $v_i \neq v_j$ are assigned *exactly* the same embedding vector, then any prompts differing only by swapping $v_i$ and $v_j$ yield identical representations. Likewise, if two absolute positional embeddings are made exactly equal and the remaining weights are tuned to suppress other positional signals,

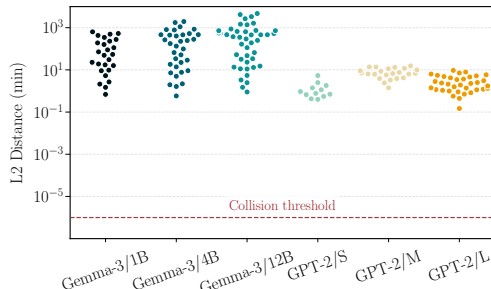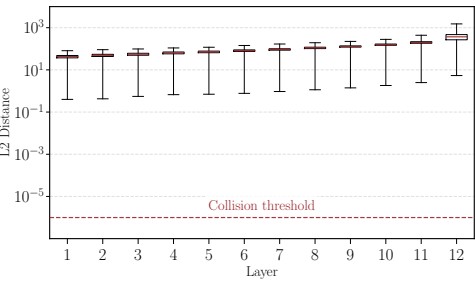

Figure 3: Seeking collisions in a large-scale prompt set: The minimum distances between last-token states are far above the collision threshold $10^{-6}$: (left) across layers for `GPT-2` and `Gemma-3` families (one dot per layer), (right) across depth for `GPT-2 Small`; distances grow with depth.

one can force collisions between sequences that differ only at those positions. These scenarios, however, require deliberately engineered parameter choices: under continuous random initialization and standard training, the probability of such coincidences is zero.

## 3 EXACT PROMPT RECOVERY VIA SIPIT

In the previous section, we have proven that decoder-only Transformers are almost surely injective, i.e., different prompts map to different hidden states. We now show how this property can be used in practice to **reconstruct the exact input prompt** given hidden states at some layer. We call this algorithm SIPIT (Sequential Inverse Prompt via ITerative updates).[3]

**Threat model.** This paper focuses on the injectivity result and its algorithmic consequence; we do not define a full adversarial model. A natural setting where SIPIT applies is one in which an adversary obtains the hidden-state sequence—for instance, through a leaked KV-cache, a shared-inference pipeline, or an API that exposes intermediate representations. Our injectivity result guarantees that exact recovery from only the final embedding is possible in principle, but designing an efficient algorithm for that setting is nontrivial and left to future work; here we assume access to all per-position states at a given layer $\ell$.

Recall from §2 that the mapping from a prompt $s$ to its last-token state is almost surely injective. Since the last state is itself a deterministic function of the hidden matrix at any layer $\ell$, injectivity extends to the full representation

$$s \mapsto \mathbf{H}^{(\ell)}(s) \in \mathbb{R}^{T \times d}. \tag{5}$$

We denote by $\mathbf{h}_t(s)$ the row of $\mathbf{H}^{(\ell)}(s)$ at position $t$. In the following, the parameters $\boldsymbol{\theta}$ and target layer $\ell$ are considered fixed and omitted for simplicity.

The algorithm exploits the causal structure of Transformers: the hidden state at position $t$ depends only on the prefix $\langle s_1, \ldots, s_{t-1} \rangle$ and the current token $s_t$. This means that if we already know the prefix, then the hidden state at position $t$ uniquely identifies $s_t$.

**Example.** Suppose the vocabulary is $a, b, c$ and the true prompt is $\langle a, b \rangle$. At $t = 1$, the hidden state depends only on $s_1$. By comparing the observed state with the three candidate states produced by trying $a$, $b$, and $c$, we can tell exactly which one matches, thus recovering $s_1 = a$. Then at $t = 2$, we know the prefix $\langle a \rangle$, so we try appending each candidate token and again match the resulting hidden state to recover $s_2 = b$. Iterating this procedure reconstructs the full sequence.

More generally, we can look at the "one-step" map

$$v_j \mapsto \mathbf{h}_t(\pi \oplus v_j), \quad v_j \in \mathcal{V}, \tag{6}$$

which gives the hidden state at step $t$ for each possible next token, given the fixed prefix $\pi = \langle s_1, \ldots, s_{t-1} \rangle$ (here $\oplus$ denotes concatenation).

---

[3]Implementation available at https://github.com/giorgosnikolaou/SIPIT.

**Remark.** By the analytic arguments of §2, the one-step map is almost surely injective: with a fixed prefix, any two distinct tokens almost surely yield distinct hidden states.

This property makes sequence recovery straightforward. At each step $t$, given the hidden state $\widehat{\mathbf{h}}_t$ and the already recovered prefix, we simply check which candidate token produces a matching hidden state. That token must be the true $s_t$. Repeating this process recovers the entire sequence.

This leads to the SIPIT algorithm, shown in Algorithm 1. At every position, the algorithm cycles through vocabulary candidates (according to some policy such as random order or gradient-guided search) until it finds the unique match[4], then appends it to the reconstructed prefix and moves on.

---

**Algorithm 1** SIPIT: Sequential Inverse Prompt via ITerative updates

---

**Require:** Observed layer-$\ell$ states $\widehat{\mathbf{H}}^{(\ell)} \in \mathbb{R}^{T \times d}$; vocabulary $\mathcal{V}$; tolerance $\varepsilon \geq 0$.
**Ensure:** Recovered sequence $\widehat{s} = \langle \hat{s}_1, \ldots, \hat{s}_T \rangle$.
1: $\widehat{s} \leftarrow \langle \rangle$
2: **for** $t = 1$ **to** $T$ **do**
3:    $\mathcal{C} \leftarrow \emptyset$                                                         ▷ tested candidates
4:    **for** $j = 1$ **to** $|\mathcal{V}|$ **do**
5:       $v_j \leftarrow \text{POLICY}(\mathcal{V}, \mathcal{C}, \widehat{s}, \ell)$                  ▷ new candidate token $v_j$ (see Alg. 2 and 3)
6:       **if** $\widehat{\mathbf{h}}_t \in \mathcal{A}_{\pi,t}(v_j\,;\varepsilon)$ **then**                  ▷ verify $v_j$ (see Def. D.2)
7:          $\widehat{s} \leftarrow \widehat{s} \oplus v_j$                                                    ▷ hit!
8:          **break**
9:       **else**
10:         $\mathcal{C} \leftarrow \mathcal{C} \cup \{v_j\}$
11:       **end if**
12:    **end for**
13: **end for**
14: **return** $\widehat{s}$

---

To rule out edge cases and analyze the computational cost of SIPIT, we now state a formal guarantee.

**Theorem 3.1** (Correctness of SIPIT). *Under the assumptions of Theorem 2.3, given observed hidden states $\widehat{\mathbf{H}}^{(\ell)}$, SIPIT recovers the true input sequence $s$ with probability one in at most $T|\mathcal{V}|$ steps.*

*Sketch of proof (full proof in Appendix D, Thm. D.2, Prop. D.4).* At each step, local injectivity ensures a unique token matches the observed state. As the policy spans the vocabulary, this token will be found in at most $|\mathcal{V}|$ trials. Induction over $t = 1, \ldots, T$ completes the argument. □

**Theorem 3.2** (Robustness of SIPIT). *Under the assumptions of Theorem 2.3, fix a layer $\ell$ and define, for any prefix $\pi$ and time $t$,*

$$\Delta_{\pi,t} := \min_{v \neq v' \in \mathcal{V}} \left\| \mathbf{h}_t(\pi \oplus v) - \mathbf{h}_t(\pi \oplus v') \right\|_2.$$

*Let $s = \langle s_1, \ldots, s_T \rangle$, define the prefixes $\pi_t = \langle s_1, \ldots, s_{t-1} \rangle$ and suppose access to the perturbed hidden states*

$$\widehat{\mathbf{h}}_t(\pi_t \oplus s_t) = \mathbf{h}_t(\pi_t \oplus s_t) + \mathbf{e}_t, \qquad \|\mathbf{e}_t\|_2 < \frac{\Delta_{\pi_t,t}}{2}, \quad t \in [T].$$

*Then SIPIT recovers the true sequence $s$ with probability one, and terminates in at most $T|\mathcal{V}|$ steps.*

*Proof in Appendix D, Thm. D.2, Prop. D.2.* □

In short, SIPIT turns the almost-sure injectivity of Transformer representations into a constructive procedure: not only are hidden states unique identifiers of prompts, but the exact input sequence can be efficiently *recovered* in linear time, and often faster in practice. It is a structural property of Transformer representations, not a quirk of initialization or training.

---

[4]In practice, we accept matches if the observed hidden state is within an $\varepsilon$-ball around the predicted one.

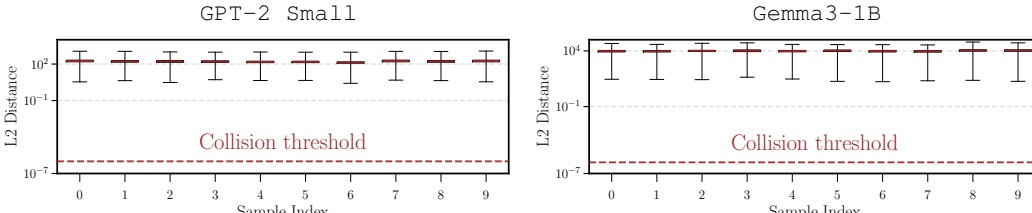

Figure 4: Exhaustive collision search on the 10 closest prefix prompts. The boxplots look flat and uneventful, and that is the point: even under stress-test conditions with billions of candidate pairs, all minima stay well above the collision threshold, showing that nothing collapses.

## 4 EXPERIMENTS

We previously proved that decoder-only Transformers are injective (§2) and introduced an algorithm, SIPIT, that leverages this property to recover the exact input prompt from hidden states at a given layer (§3). We now provide extensive empirical evidence supporting our theory by showing that distinct prompts yield distinct embeddings, i.e., no collisions occur by a large margin (§4.1). We then demonstrate that SIPIT successfully reconstructs the original input prompt (§4.2).

**Environment.** All experiments were run on a single NVIDIA A100-SXM (64 GB) GPU. Python 3.11, CUDA 12.2, PyTorch 2.8.0, and `transformers` 4.50.0 were used for all experiments. Reported runtimes refer to this setup.

### 4.1 SEARCHING FOR COLLISIONS

We collected 100k prompts by uniformly sampling from a mixture of four datasets: `wikipedia-en`[5], `C4` (Raffel et al., 2020), `The Pile` (Gao et al., 2020), and `python-github-code`[6]. For each prompt, we extracted the last-token representation and systematically checked whether any two distinct prompts produced identical embeddings. This process required around **5 billion** pairwise comparisons.

| Model | $\ell_2$ Distance (min) | | |
|---|---|---|---|
| | layer 1 | layer $\frac{L}{2}$ | layer $L$ |
| Llama-3.1-8B | 0.001 | 0.129 | 0.620 |
| Mistral-7B-v0.1 | 0.002 | 0.187 | 1.274 |
| Phi-4-mini-ins | 0.014 | 1.336 | 9.020 |
| TinyStories-33M | 0.029 | 1.434 | 2.793 |

Table 1: Minimum pairwise distance between last-token states in the first, middle, and final layers of four models. All values are well above the collision threshold $10^{-6}$.

We observed **no collisions** across all models and layers: distinct prompts always yielded distinct last-token states. Figure 3 (left) shows the per-layer minimum distances for the Gemma3 pretrained (Team et al., 2025) and GPT-2 (Radford et al., 2019) families, with strictly positive values throughout. Table 1 complements this by reporting the same statistic for Llama-3.1-8B (Grattafiori et al., 2024), Mistral-7B-v0.1 (Jiang et al., 2023), Phi-4-mini-instruct (Microsoft et al., 2025) and TinyStories-33M (Eldan & Li, 2023), again showing clear separation at the first, middle, and last layers. Finally, Figure 3 (right) zooms in on GPT-2 Small, revealing that these distances typically increase with depth. Additional results for GPT-2 Medium, GPT-2 Large and Gemma3 (1B, 4B, 12B) appear in Appendix E, confirming the same trend.

| Model | $\ell_2$ Distance (min) | | |
|---|---|---|---|
| | FP4 | INT8 | FP32 |
| Llama-3.1-8B | 2.281 | 6.597 | 1.274 |
| Mistral-7B-v0.1 | 1.748 | 2.692 | 1.136 |
| Phi-4-mini-instruct | 18.368 | 20.956 | 8.780 |

| Model | Size | $\ell_2$ Distance (min) | | |
|---|---|---|---|---|
| | | layer 1 | layer $L/2$ | layer $L$ |
| phi-4 | 14B | 0.010 | 1.025 | 8.759 |
| Llama-3.1-70B | 70B | 0.005 | 0.465 | 3.975 |

Table 2: **Quantized Models:** Minimum pairwise distance between last-token states in the final layer of three quantized models.

Table 3: **Large Models**: Minimum pairwise distance between last-token states in the first, middle, and final layers of two large models.

[5]https://huggingface.co/datasets/wikimedia/wikipedia
[6]https://huggingface.co/datasets/angie-chen55/python-github-code

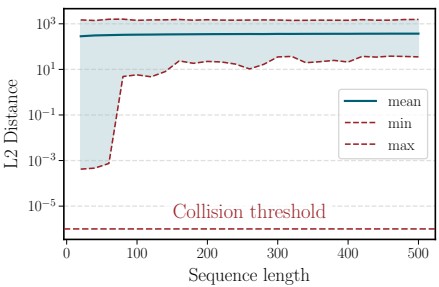

Figure 5: Sequence length vs. pairwise distance for `GPT-2`. Min, mean, and max distances rise at short lengths and then stabilize, indicating consistent separability.

Figure 5 shows how pairwise distances between last-token states vary with prompt length in `GPT-2` `Small`. Three patterns emerge: (i) the *minimum* distance is never close to zero at all lengths, and (ii) it grows rapidly at short lengths but then levels off, suggesting that beyond a moderate context size, adding tokens does not affect separability; (iii) the overall spread (min-max) stays bounded, with no sign of pathological collapses. Similar behavior is seen in `Gemma3` (see Appendix E, Figure 9). Overall, clear margins emerge quickly and then stabilize, making collisions unlikely at any sequence length.

**Exhaustive collision test.** Different from previous experiments, in this setting (Figure 4), we restrict our analysis to the 10 prompts from the dataset mixture whose embeddings have the smallest last-token distances. For each of these prompts, we appended *every* vocabulary token and computed all pairwise distances between the resulting last-token states, effectively performing an exhaustive search over continuations and yielding more than **343 billion** prompt pairs per model. This exhaustive experiment helps rule out the possibility that earlier observations were simply due to chance in random sampling rather than a true absence of collisions. While a complete search over all possible prompts would be ideal, it is computationally infeasible. The number of unique prompts grows exponentially with sequence length, and the number of pairwise comparisons grows even faster. For context, even with single-token prompts and the vocabulary size of `Gemma3-1B`, there are already over 34 billion possible prompt pairs, making exhaustive evaluation entirely impractical. Our compromise still revealed structure: we identified 5 prompt pairs with highly similar last-token embeddings, suggesting overlapping semantic content and motivating us to ask whether distinct next tokens could preserve meaning, i.e., yield essentially identical last-token hidden states.

Figure 4 reports the resulting distributions as boxplots for both `GPT-2` `Small` and `Gemma3-1B`, with distances far from zero (no collision), **confirming local injectivity** as predicted by our theory.

**FP4 and INT8 weight quantization.** To assess how weight quantization affects pairwise representation distances, we conducted additional experiments with FP4 and INT8 quantization on several models (`Llama-3.1-8B`, `Phi-4-mini-instruct`, and `Mistral-7B-v0.1`). We further extended this analysis to FP4-quantized **14B** and **70B** parameter models, namely `Phi-4` (14B) and `Llama-3.1-70B`. As shown in Tables 2 and 3, across all tested models quantization **(1)** does not introduce any collisions, **(2)** more than doubles the minimum distance between representations, thereby preserving the integrity of the representation space, and **(3)** maintains this separation even as model size increases substantially.

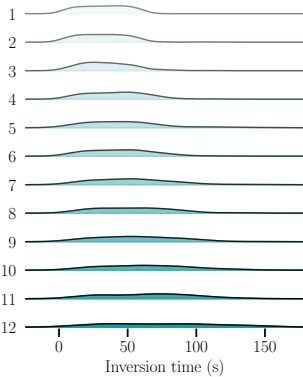

Figure 6: Inversion time as a function of depth. Runtimes rise only mildly across layers.

## 4.2 INVERTIBILITY RESULTS

We now test whether the theoretical injectivity translates into exact recovery on pre-trained models. Using SIPIT with only the hidden states at a fixed layer, we attempt to reconstruct the full prompt token-by-token for `GPT-2` `Small`. We sample 100 prompts, with a 90%-10% split between meaningful sentences and random token sequences (to test robustness in unstructured cases), and attempt to reconstruct them from hidden states. We compare against HARDPROMPTS (Wen et al., 2023), which leverages gradient signals for approximate prompt discovery, and against a SIPIT ablation that replaces the gradient-guided candidate policy with the uniformly random policy (BRUTEFORCE).

Other inversion approaches (Morris et al., 2023a;b; Nazir et al., 2025) tackle a different setting altogether: they operate in black box access, using sequences of next-token logprobs or encoder logits rather than hidden states, and train auxiliary inverters to reconstruct text, at high computational cost. Their outputs are typically approximate and not guaranteed exact. These differences make

| Model | Vocab size | Inversion Performance | | |
|---|---|---|---|---|
| | | Accuracy | Time (s) | Vocab explored (%) |
| Mistral-7B-v0.1 | 32000 | 100% | $111.78 \pm 46.50$ | $0.19 \pm 0.08\,\%$ |
| Llama-3.1-8B | 128255 | 100% | $549.48 \pm 265.75$ | $0.21 \pm 0.10\,\%$ |

Table 4: Inversion performance on FP4-quantized models with different vocabulary sizes. SIPIT recovers all tokens with 100% accuracy while exploring less than $0.22\%$ of the vocabulary on average.

them complementary but not directly comparable to our setting of training-free, *exact* inversion from *hidden states* in decoder-only LMs.

Results are reported in Table 5. Across all prompts (20 tokens each), SIPIT recovers the **exact** sequence with $100\%$ token-level accuracy (no errors, no collisions), matching the theoretical guarantee of linear-time convergence. In contrast, HARDPROMPTS completely fails to recover the input, while BRUTEFORCE eventually succeeds but at a prohibitive computational cost, requiring several orders of magnitude longer.

| Method | Mean Time (s) | Accuracy |
|---|---|---|
| HARDPROMPTS | $6132.59 \pm 104.61$ | 0.00 |
| BRUTEFORCE (ours) | $3889.61 \pm 691.17$ | 1.00 |
| SIPIT (ours) | $\mathbf{28.01 \pm 35.87}$ | $\mathbf{1.00}$ |

Table 5: SIPIT ensures efficient exact recovery, unlike HARDPROMPTS (no recovery) or BRUTEFORCE (infeasible runtime).

**Robustness and vocabulary scaling.** The theoretical analysis in Theorem 3.2 shows that our inversion algorithm is robust to a certain level of noise while maintaining linear scaling in vocabulary size. To empirically validate this, we use FP4-quantized versions of Mistral-7B-v0.1 ($\approx 32$K vocabulary size) and Llama-3.1-8B ($\approx 128$K). We sample 50 prompts (10 tokens each) and attempt to reconstruct them from hidden states corrupted by FP4 weight quantization. As shown in Table 4, SIPIT reconstructs all inputs with perfect accuracy while exploring, on average, less than $0.22\%$ of the vocabulary, demonstrating that the gradient-based heuristic is both robust to quantization noise and highly efficient. From a complexity perspective, the nearly constant percentage of tokens explored across the two vocabulary scales empirically confirms the predicted linear scaling.

**Effect of layer depth.** Finally, Figure 6 shows inversion times by layer for longer prompts (ranging from 20 to 200 tokens). Although deeper layers are costlier in principle (since verifying a candidate and computing gradients require traversing more blocks), the effect is minor: runtimes rise only slightly from first to last layer, and the scaling remains graceful overall. Likely, earlier layers need more iterations to converge, while deep layers store richer information that reduces the search effort. As a result, the net cost remains stable, confirming SIPIT is efficient across depth.

## 5 RELATED WORK

Our results connect to two active lines of research: theoretical analyses of Transformer architectures, and inverse problems in language modeling. We briefly review both to position our contributions.

**Analytical properties of Transformers.** Viewed as functions on $\mathbb{R}^d$, individual Transformer components are clearly non-injective: LayerNorm collapses along per-example statistics (Ba et al., 2016), residual connections can cancel, and in attention-only stacks, rank decays doubly-exponentially with depth (Dong et al., 2021). Likewise, on the output side, the softmax bottleneck constrains the distributions reachable by language models (Yang et al., 2018). From this algebraic perspective, Transformers seem inherently many-to-one, an intuition echoed by classical completeness and universal-approximation theorems for Transformers, which show that highly many-to-one maps can be represented in principle; we briefly review these results in Appendix F.

Our focus is different: we study the discrete-to-continuous map from *prompts* $s \in \mathcal{V}^{\leq K}$ to *hidden states* in $\mathbb{R}^d$. In this setting, analytic viewpoints on Transformer computation become powerful: treating each layer as a real-analytic map yields almost-sure guarantees that hold at finite width, depth, and training horizon (Appendix F surveys which modern LLMs satisfy this assumption and proves the analyticity for all activation functions encountered in practice). Recent work has adopted this angle for related properties: Jiang & Haghtalab (2025) show that building blocks of modern architectures are *almost always surjective*, while Sutter et al. (2025) prove that Transformers at random initialization are *almost surely injective* with respect to the entire hidden-state matrix (and only at initialization). Differently, we prove injectivity with respect to the *parameters* and at the task-relevant *last-token state*; crucially, we show that injectivity is not an initialization artifact but *persists under training*.

**Inverse problems in language modeling.** Model inversion asks whether one can reconstruct a model's input prompt from outputs or internal signals (Sun et al., 2021). In the context of language models, this question has motivated a growing body of work exploring practical inversion strategies. Output-to-prompt methods infer prompts from generated continuations but yield only approximate reconstructions (Zhang et al., 2024). Recent work shows that even black-box outputs are information-rich: Morris et al. (2023b) train a separate inverter to map next-token probability vectors to text, and Nazir et al. (2025) extend this by taking sequences of logprobs, applying a linear compression to embedding dimension, and training an encoder-decoder inverter; this achieves higher exact-match rates but still without guarantees. Complementarily, Morris et al. (2023a) reconstruct text from encoder logits via a trained iterative inverter. These contributions highlight privacy risks when probabilities or embeddings are exposed, but they differ from our setting: they rely on trained inverters, remain approximate, and do not invert *hidden states* of decoder-only LMs.

A related line of work frames the task as automated prompt optimization, casting prompt design as discrete sequence optimization aligned with downstream performance (Guo et al., 2025; Sun et al., 2022; Deng et al., 2022); methods such as AutoPrompt (Shin et al., 2020) and Hard Prompts Made Easy (Wen et al., 2023) use gradient signals to discover effective, but approximate, prompts. Most closely related to ours, Thomas et al. (2025) recover prompts from hidden states via a sequential algorithm that uses an LLM-based policy to rank candidates; lacking injectivity guarantees, however, it must score all vocabulary tokens before committing, with no formal exactness guarantees.

Unlike all prior work, our approach is training-free, efficient, and comes with *provable* linear-time guarantees for *exact* recovery from internal states.

# 6 DISCUSSION AND CONCLUSIONS

This work establishes that decoder-only Transformers are almost surely injective: distinct prompts produce distinct hidden states under standard initialization and training. Building on this structural result, we introduced SIPIT, the first algorithm that can recover the *exact* input sequence from hidden activations, with provable linear-time guarantees. Together, these contributions move injectivity from an informal belief to a rigorously grounded and operational property of language models.

The scientific impact is clear. Our findings reconcile two competing views in the community: Transformers as "lossy" due to nonlinearities, normalization, and many-to-one attention, versus language models as injective in their hidden representations. We advocate viewing language models as maps on the *sequence* space rather than the embedding space; under this perspective, we prove that all information about the input sequence is almost surely preserved end-to-end. The constructive inversion offered by SIPIT strengthens this point in practice, establishing a clean baseline for interpretability and auditing: if probes or inversion methods fail, it is not because the information is missing. For mechanistic interpretability in particular, injectivity guarantees that last-token states faithfully encode the full input, giving a sound foundation for causal and probing analyses.

Beyond theory, the findings carry practical and legal implications. Hidden states are not abstractions but the prompt in disguise: any system that stores or transmits them is effectively handling user text itself, with direct consequences for privacy, deletion, and compliance (Miranda et al., 2025). The evolving regulatory landscape has not yet fully reckoned with this fact. The Hamburg Data Protection Commissioner, for instance, argued that LLM *parameters* do not constitute personal data because training data is transformed into abstract mathematical representations during learning, and that it "remains doubtful whether any extractable data records constitute personal data" (HmbBfDI, 2024). That analysis, however, concerns training data encoded in model weights; it does not address the hidden representations computed at inference time. Our results reveal that these representations are lossless encodings of the user's exact input, recoverable in full via SIPIT. Consequently, any system that stores, caches, or transmits hidden states is effectively handling the user's verbatim text, and the corresponding pipelines should be subject to the same data-protection obligations as the raw prompts they encode.

Finally, this work opens several directions. Extending the analysis to multimodal architectures such as music and vision Transformers is an open problem. Studying approximate inversion under noise or quantization will clarify how robust invertibility remains in practice. Bridging these technical insights with evolving regulatory frameworks will be crucial for safe and responsible deployment.

## REPRODUCIBILITY STATEMENT

The experimental setup (hardware, software versions, and dataset construction) is described in §4; the 100k-prompt benchmark uses uniform sampling from four public datasets detailed in §4.1. On the theory side, every theorem stated in the main text is accompanied by a complete proof in the appendix: analytic preliminaries in Appendix A, the formal definition of the Transformer language model and the proof that it is real-analytic in Appendix B, almost-sure injectivity and its preservation under training in Appendix C, and SIPIT correctness and robustness in Appendix D. Appendix E provides Implementation Details and Additional Experiments, and Appendix F verifies that all activation functions used in all modern LLMs satisfy the analyticity assumption.

## ACKNOWLEDGMENTS

Figure 1 is adapted from *Autoencoder Diagrams* by Keenan Crane (2025), used under CC0 1.0 Universal. We further acknowledge Adam Barla for the initial discussions on LLMs invertibility.

This work has been supported by project MIS 5154714 of the National Recovery and Resilience Plan Greece 2.0 funded by the European Union under the NextGenerationEU Program, the MUR FIS2 grant n. FIS-2023-00942 "NEXUS" (cup B53C25001030001), and partly by Sapienza University of Rome via the Seed of ERC grant "MINT.AI" (cup B83C25001040001).

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

APPENDIX OVERVIEW

## A  PRELIMINARIES

This section fixes the notation used throughout the main paper and the appendix (subsection A.1), and it introduces *real-analyticity* as the organizing theme (subsection A.2). We first review the vector-space notion and its basic closure/composition properties (subsubsection A.2.1), together with a zero-set principle used in measure-zero arguments. We then extend these ideas to maps between matrix spaces (subsubsection A.2.2) via vectorization/matricization and note that analyticity is preserved under matrix compositions. To streamline later proofs, we summarize real-analytic building blocks commonly used in transformer layers–polynomials, exponential/logarithm, softmax, row normalization, matrix products, Hadamard scaling, and stacking (subsubsection A.2.3). Finally, in subsection A.3, we collect differential and topological tools–Fréchet derivatives and the Hessian, standard facts on $\mathbb{R}^p$, the inverse function theorem, and pushforwards/absolute continuity–which we use for local invertibility and absolute-continuity arguments. Readers already comfortable with these topics can skim now and return to specific subsections as needed.

### A.1  NOTATION

For arbitrary $T \in \mathbb{N}$, we write $[T] = \{1, 2, \ldots, T\}$ to denote the set of positive integers up to $T$. Additionally, we denote the strictly positive real numbers as $\mathbb{R}^+ = (0, \infty)$ and the non-negative real numbers as $\mathbb{R}_0^+ = [0, \infty)$. Similarly, we let $\mathbb{N}_0 = \mathbb{N} \cup \{0\}$.

Discrete sets are denoted by uppercase calligraphic letters $\mathcal{V}$, and a sequence of length $K$ is denoted by lowercase letters: $\mathrm{s} = \langle s_1, \ldots, s_K \rangle \in \mathcal{V}^K$. We write $|\mathrm{s}| = K$ to denote the length of the sequence. The set of non-empty sequences of length at most $K$ is denoted as $\mathcal{V}^{\leq K} = \bigcup_{k=1}^{K} \mathcal{V}^k$. Non-discrete sets are denoted by uppercase calligraphic bold-face letters $\boldsymbol{\mathcal{B}}$.

**Remark 1.** *We will often refer to a discrete set $\mathcal{V}$ as the vocabulary and to an element $\mathrm{s} \in \mathcal{V}^{\leq K}$ as an input, context, or prompt.*

Matrices (vectors) are denoted by uppercase (lowercase) bold-face letters: $\mathbf{X} \in \mathbb{R}^{d_1 \times d_2}$ ($\mathbf{x} \in \mathbb{R}^d$). For vectors and matrices, we frequently use standard norms and common matrix operations. The Hadamard and Kronecker products are defined following [Kolda & Bader (2009)](#):

- **$p$-norm:** For a vector $\mathbf{x} \in \mathbb{R}^d$, the $\ell_p$ norm is defined as

$$\|\mathbf{x}\|_p = \left( \sum_{i=1}^{d} |\mathbf{x}_i|^p \right)^{\frac{1}{p}}.$$

- **Frobenius norm:** For a matrix $\mathbf{X} \in \mathbb{R}^{d_1 \times d_2}$, the Frobenius norm is defined as

$$\|\mathbf{X}\|_{\mathrm{F}} = \sqrt{\mathrm{tr}(\mathbf{X}\mathbf{X}^{\top})} = \sqrt{\sum_{i=1}^{d_1} \sum_{j=1}^{d_2} \mathbf{X}_{ij}^2}.$$

- **Hadamard product:** The Hadamard (element-wise) product is defined for vectors and matrices of the same shape:

$$
\begin{aligned}
(\mathbf{x} \odot \mathbf{y})_i &= \mathbf{x}_i \mathbf{y}_i, & &\text{for all } i \in [d], \\
(\mathbf{X} \odot \mathbf{Y})_{ij} &= \mathbf{X}_{ij} \mathbf{Y}_{ij}, & &\text{for all } i \in [d_1], \ j \in [d_2],
\end{aligned}
$$

where $\mathbf{x}, \mathbf{y} \in \mathbb{R}^d$ and $\mathbf{X}, \mathbf{Y} \in \mathbb{R}^{d_1 \times d_2}$.

- **Kronecker product:** The Kronecker product of $\mathbf{X} \in \mathbb{R}^{d_1 \times d_2}$ and $\mathbf{Z} \in \mathbb{R}^{d_3 \times d_4}$ is denoted $\mathbf{X} \otimes \mathbf{Z}$ and defined blockwise as

$$\mathbf{X} \otimes \mathbf{Z} = \begin{bmatrix} \mathbf{X}_{11}\mathbf{Z} & \cdots & \mathbf{X}_{1d_2}\mathbf{Z} \\ \vdots & \ddots & \vdots \\ \mathbf{X}_{d_1 1}\mathbf{Z} & \cdots & \mathbf{X}_{d_1 d_2}\mathbf{Z} \end{bmatrix} \in \mathbb{R}^{(d_1 d_3) \times (d_2 d_4)}.$$

We denote the all-zeros matrix of size $m \times n$ as $\mathbf{0}_{m \times n}$, and the all-zeros vector of length $m$ as $\mathbf{0}_m$. Similarly, we write $\mathbf{1}_m$ for the all-ones vector of length $m$, and $\mathbf{I}_m$ (or $\mathbf{I}_{m \times m}$ when dimensions must be explicit) for the $m \times m$ identity matrix.

Let $f : \mathcal{V}^{\leq K} \times \mathbb{R}^p \to \mathbb{R}^d$ be a function over a finite vocabulary $\mathcal{V}$ and $K \in \mathbb{N}$. We refer to $f$ as the *model*, to its first argument as the *input sequence*, and to its second argument as the *parameters*.

**Remark 2.** *Throughout our analysis, we assume a finite set of possible input sequences, reflecting the practical limitations and design choices of modern LLMs, specifically the bounded context length.*

**Remark 3.** *We take the codomain of the model to be $\mathbb{R}^d$, corresponding to the space of token embeddings. This allows us to study how the final embedding (typically used to compute next-token probabilities) depends on both the input sequence and the model parameters.*

## A.2 REAL-ANALYTICITY

We now introduce the central notion for our analysis: real-analyticity. In its standard form, real-analyticity is defined for functions $f : \mathcal{U} \to \mathbb{R}^n$, where $\mathcal{U} \subseteq \mathbb{R}^m$ is an open set. Since the transformer architecture is naturally expressed in terms of matrices, it will be convenient to extend this notion to maps of the form $f : \mathbb{R}^{m \times n} \to \mathbb{R}^{a \times b}$.

**Multi-index notation.** We use multi-index notation for both vectors and matrices.

*Vector case.* Let $\boldsymbol{\alpha} = (\alpha_1, \ldots, \alpha_m)^\top \in \mathbb{N}_0^m$ and $\mathbf{x}, \mathbf{y} \in \mathbb{R}^m$. Define:

$$|\boldsymbol{\alpha}| = \sum_{j=1}^m \alpha_j, \qquad \boldsymbol{\alpha}! = \prod_{j=1}^m \alpha_j!, \qquad (\mathbf{x} - \mathbf{y})^{\boldsymbol{\alpha}} = \prod_{j=1}^m (\mathbf{x}_j - \mathbf{y}_j)^{\alpha_j}.$$

*Matrix case.* Let $\mathbf{A} = (\alpha_{uv}) \in \mathbb{N}_0^{m \times n}$ and $\mathbf{X}, \mathbf{Y} \in \mathbb{R}^{m \times n}$. Define:

$$|\mathbf{A}| = \sum_{u=1}^m \sum_{v=1}^n \alpha_{uv}, \qquad \mathbf{A}! = \prod_{u=1}^m \prod_{v=1}^n \alpha_{uv}!, \qquad (\mathbf{X} - \mathbf{Y})^{\mathbf{A}} = \prod_{u=1}^m \prod_{v=1}^n (\mathbf{X}_{uv} - \mathbf{Y}_{uv})^{\alpha_{uv}}.$$

Given an open set $\mathcal{U} \subseteq \mathbb{R}^m$ and a map $f : \mathcal{U} \to \mathbb{R}$, we write

$$\mathbf{d}^{\boldsymbol{\alpha}} f(\mathbf{x}) := \frac{\partial^{|\boldsymbol{\alpha}|} f}{\partial \mathbf{x}_1^{\alpha_1} \cdots \partial \mathbf{x}_m^{\alpha_m}}(\mathbf{x})$$

for the mixed partial derivative (when it exists). Unless stated otherwise, we assume $f \in C^\infty(\mathcal{U})$, so $\mathbf{d}^{\boldsymbol{\alpha}} f$ exists and is continuous for all $\boldsymbol{\alpha} \in \mathbb{N}_0^m$; for vector-valued maps $f = (f_1, \ldots, f_n)$ the operator $\mathbf{d}^{\boldsymbol{\alpha}}$ acts componentwise. We also use the convention $\mathbf{d}^{\mathbf{0}} f = f$.

### A.2.1 REAL-ANALYTIC FUNCTIONS WITH VECTOR INPUTS

We begin with the standard vector-space definition and its basic algebraic properties. These are the building blocks from which all later analyticity arguments are assembled.

**Definition A.1** (Real-analytic functions, Lewis 2014, Definition 1.1.3)**.** *Let $\mathcal{U} \subseteq \mathbb{R}^m$ be open. A function $f : \mathcal{U} \to \mathbb{R}$ is **real-analytic** on $\mathcal{U}$ if, for every $\mathbf{y} \in \mathcal{U}$, there exist coefficients $\{c_{\boldsymbol{\alpha}} \in \mathbb{R}\}_{\boldsymbol{\alpha} \in \mathbb{N}_0^m}$ and $r > 0$ such that*

$$f(\mathbf{x}) = \sum_{\boldsymbol{\alpha} \in \mathbb{N}_0^m} c_{\boldsymbol{\alpha}} (\mathbf{x} - \mathbf{y})^{\boldsymbol{\alpha}}$$

*for all $\mathbf{x} \in \mathcal{U}$ with $\|\mathbf{x} - \mathbf{y}\|_2 < r$. The set of real-analytic functions on $\mathcal{U}$ is denoted by $C^\omega(\mathcal{U})$.*

*A map $f : \mathcal{U} \to \mathbb{R}^n$ is real-analytic on $\mathcal{U}$ if each of its components $f_1, \ldots, f_n : \mathcal{U} \to \mathbb{R}$ is real-analytic. The set of such maps is denoted $C^\omega(\mathcal{U} ; \mathbb{R}^n)$.*

**Remark 4.** *To establish real-analyticity of a vector-valued mapping (e.g., an MLP, attention mechanism, or LayerNorm), it suffices to prove real-analyticity of each scalar component.*

**Proposition A.1** (Closure properties, Lewis 2014, Proposition 1.2.1)**.** *Let $f, g : \mathbb{R}^m \to \mathbb{R}$ be real-analytic maps. Then, the following hold:*

1. ***Addition:*** *$f + g \in C^\omega(\mathbb{R}^m)$.*

2. ***Product:*** *$fg \in C^\omega(\mathbb{R}^m)$.*

3. ***Quotient:*** *If $g(\mathbf{x}) \neq 0$ for all $\mathbf{x} \in \mathbb{R}^m$, then $f/g \in C^\omega(\mathbb{R}^m)$.*

**Proposition A.2** (Composition, Lewis 2014, Proposition 1.2.2)**.** *Let $f : \mathbb{R}^m \to \mathbb{R}^n$ and $g : \mathbb{R}^n \to \mathbb{R}^k$ be real-analytic maps. Then, the composition $g \circ f : \mathbb{R}^m \to \mathbb{R}^k$ is real-analytic.*

**Remark 5.** *For simplicity, we do not state the closure properties in their most general form, where $f$ and $g$ may be defined on different open subsets of $\mathbb{R}^m$. This avoids additional notation involving intersections of domains. Since every function of interest in our later analysis is defined on the whole space $\mathbb{R}^m$, this restriction entails no loss of generality.*

**Theorem A.1** (Zero sets of nontrivial real-analytic maps Mityagin 2015)**.** *Let $\mathcal{U} \subseteq \mathbb{R}^m$ be connected and open, and let $f \in C^\omega(\mathcal{U} ; \mathbb{R}^n)$. If $f \not\equiv \mathbf{0}_n$, then its zero set*

$$Z(f) := f^{-1}(\{\mathbf{0}_n\}) = \{\mathbf{x} \in \mathcal{U} : f(\mathbf{x}) = \mathbf{0}_n\}$$

*has Lebesgue measure zero in $\mathbb{R}^m$ (i.e. $\mathrm{Leb}_m\big(Z(f)\big) = 0$). Equivalently, if there exists $\mathbf{x} \in \mathcal{U}$ with $f(\mathbf{x}) \neq \mathbf{0}_n$, then $\mathrm{Leb}_m\big(f^{-1}(\{\mathbf{0}_n\})\big) = 0$.*

**Remark 6.** *The result in Mityagin (2015) is stated for scalar-valued maps $f : \mathcal{U} \to \mathbb{R}$. The extension to vector-valued maps $f = (f_1, \ldots, f_n) : \mathcal{U} \to \mathbb{R}^n$ is immediate: the zero set of $f$ is the intersection of the zero sets of its scalar components,*

$$Z(f) = \bigcap_{i=1}^{n} Z(f_i),$$

*and if $f \not\equiv \mathbf{0}_n$, then at least one component $f_j \not\equiv 0$, so $Z(f) \subseteq Z(f_j)$, which has measure zero by the scalar case.*

## A.2.2 REAL-ANALYTIC FUNCTIONS WITH MATRIX INPUTS

Since transformer layers operate on matrices (e.g., $\mathbf{X} \in \mathbb{R}^{T \times d}$), we need to extend real-analyticity from vector spaces to matrix spaces. The key tool is the vectorization operator, which lets us reduce matrix-analytic questions to the vector case treated above.

**Definition A.2** (Real-analyticity on matrix spaces). *Let $\mathcal{U} \subseteq \mathbb{R}^{m \times n}$ be open. A function $f : \mathcal{U} \to \mathbb{R}$ is **real-analytic** on $\mathcal{U}$ if, for every $\mathbf{Y} \in \mathcal{U}$, there exist coefficients $\{c_{\mathbf{A}} \in \mathbb{R}\}_{\mathbf{A} \in \mathbb{N}_0^{m \times n}}$ and $r > 0$ such that*

$$f(\mathbf{X}) = \sum_{\mathbf{A} \in \mathbb{N}_0^{m \times n}} c_{\mathbf{A}} (\mathbf{X} - \mathbf{Y})^{\mathbf{A}}$$

*for all $\mathbf{X} \in \mathcal{U}$ with $\|\mathbf{X} - \mathbf{Y}\|_{\mathrm{F}} < r$.*

*A map $f : \mathcal{U} \to \mathbb{R}^{a \times b}$ is real-analytic on $\mathcal{U}$ if each of its components $f_{ij} : \mathcal{U} \to \mathbb{R}$ is real-analytic. The set of such maps is denoted $C^{\omega}(\mathcal{U} ; \mathbb{R}^{a \times b})$.*

**Remark 7.** *In the special case where $n = b = 1$, the domain and codomain reduce to $\mathbb{R}^m$ and $\mathbb{R}^a$, respectively. Then Definition A.2 recovers Definition A.1. Thus, Definition A.2 generalizes real-analyticity to functions between matrix spaces.*

**Definition A.3** (Vectorization and matricization Operators). *Let $\mathrm{vec}_{m,n} : \mathbb{R}^{m \times n} \to \mathbb{R}^{mn}$ denote the standard **vectorization operator**, which stacks the columns of a matrix into a single column vector (Henderson & Searle, 1981).*

*We also define the corresponding **matricization operator** $\mathrm{mat}_{m,n} : \mathbb{R}^{mn} \to \mathbb{R}^{m \times n}$. As shown in Chacón & Duong 2020, the vectorization and matricization operators are mutual inverses:*

$$\mathrm{mat}_{m,n}\big(\mathrm{vec}_{m,n}(\mathbf{X})\big) = \mathbf{X} \quad \forall \mathbf{X} \in \mathbb{R}^{m \times n} \tag{7}$$

$$\mathrm{vec}_{m,n}\big(\mathrm{mat}_{m,n}(\mathbf{x})\big) = \mathbf{x} \quad \forall \mathbf{x} \in \mathbb{R}^{mn} \tag{8}$$

*Furthermore, if $\mathbf{x} \in \mathbb{R}^{mn}$ and $\mathbf{X} \in \mathbb{R}^{m \times n}$ are related by vectorization and matricization, i.e., $\mathbf{x} = \mathrm{vec}_{m,n}(\mathbf{X})$ and $\mathbf{X} = \mathrm{mat}_{m,n}(\mathbf{x})$, then their norms coincide:*

$$\|\mathbf{x}\|_2 = \|\mathbf{X}\|_{\mathrm{F}}.$$

**Definition A.4** (Vectorized Form of Function). *Let $\mathcal{U} \subseteq \mathbb{R}^{m \times n}$ be open and $\tilde{\mathcal{U}} = \mathrm{vec}_{m,n}(\mathcal{U})$ (also open since $\mathrm{vec}$ is a linear homeomorphism). We denote the **vectorized form** of a function $f : \mathcal{U} \to \mathbb{R}^{a \times b}$ as*

$$\tilde{f} := \mathrm{vec}_{a,b} \circ f \circ \mathrm{mat}_{m,n} : \tilde{\mathcal{U}} \to \mathbb{R}^{ab}.$$

*Equivalently, for all $\mathbf{X} \in \mathcal{U}$:*

$$f(\mathbf{X}) = \mathrm{mat}_{a,b}\bigg( \tilde{f}\big(\mathrm{vec}_{m,n}(\mathbf{X})\big) \bigg) \tag{9}$$

**Lemma A.1** (Equivalence real-analyticity). *Let $\mathcal{U} \subseteq \mathbb{R}^{m \times n}$ be open, $\tilde{\mathcal{U}} = \mathrm{vec}_{m,n}(\mathcal{U})$, and let $f : \mathcal{U} \to \mathbb{R}^{a \times b}$ with its vectorized form $\tilde{f} : \tilde{\mathcal{U}} \to \mathbb{R}^{ab}$.*

*Fix $\mathbf{Y} \in \mathcal{U}$ and set $\mathbf{y} = \mathrm{vec}_{m,n}(\mathbf{Y}) \in \tilde{\mathcal{U}}$. Then the following are equivalent:*

*1. $f$ is real-analytic at $\mathbf{Y}$ (in the sense of Definition A.2).*

*2. $\tilde{f}$ is real-analytic at $\mathbf{y}$ (in the sense of Definition A.1).*

*Proof.* We begin by establishing the correspondence between matrix and vector indices in $\mathbb{R}^{k \times \ell}$ and $\mathbb{R}^{k\ell}$. For $s \in [k\ell]$, define:

$$u(s) := 1 + (s - 1) \bmod k \qquad \text{(row index)}$$

$$v(s) := 1 + \left\lfloor \frac{s - 1}{k} \right\rfloor \qquad \text{(column index)}$$

Then $(u(s), v(s)) \in [k] \times [\ell]$ gives the matrix coordinates corresponding to the $s$th entry of the vectorization. Conversely, for $(u, v) \in [k] \times [\ell]$, define:

$$s(u, v) := u + (v - 1)k \in [k\ell]$$

to recover the linear index.

When clear from context, we omit arguments and simply write $u$, $v$, or $s$ for readability.

Let $\mathbf{X}, \mathbf{Y} \in \mathbb{R}^{m \times n}$, with vectorizations $\mathbf{x} = \mathrm{vec}_{m,n}(\mathbf{X})$ and $\mathbf{y} = \mathrm{vec}_{m,n}(\mathbf{Y})$. For a vector multi-index $\boldsymbol{\alpha} \in \mathbb{N}_0^{mn}$, define the corresponding matrix multi-index $\mathbf{A}_{\boldsymbol{\alpha}} := \mathrm{mat}_{m,n}(\boldsymbol{\alpha})$, so that:

$$(\mathbf{x} - \mathbf{y})^{\boldsymbol{\alpha}} = \prod_{s=1}^{mn} (\mathbf{x}_s - \mathbf{y}_s)^{\boldsymbol{\alpha}_s} = \prod_{u=1}^{m} \prod_{v=1}^{n} (\mathbf{X}_{uv} - \mathbf{Y}_{uv})^{(\mathbf{A}_{\boldsymbol{\alpha}})_{uv}} = (\mathbf{X} - \mathbf{Y})^{\mathbf{A}_{\boldsymbol{\alpha}}}. \qquad (10)$$

Similarly, for a matrix multi-index $\mathbf{A} \in \mathbb{N}_0^{m \times n}$, define the corresponding vector multi-index $\boldsymbol{\alpha}_{\mathbf{A}} := \mathrm{vec}_{m,n}(\mathbf{A})$, giving:

$$(\mathbf{X} - \mathbf{Y})^{\mathbf{A}} = \prod_{u=1}^{m} \prod_{v=1}^{n} (\mathbf{X}_{uv} - \mathbf{Y}_{uv})^{\mathbf{A}_{uv}} = \prod_{s=1}^{mn} (\mathbf{x}_s - \mathbf{y}_s)^{(\boldsymbol{\alpha}_{\mathbf{A}})_s} = (\mathbf{x} - \mathbf{y})^{\boldsymbol{\alpha}_{\mathbf{A}}}. \qquad (11)$$

Now let $\mathbf{M} \in \mathcal{U}$, and let $\mathbf{m} = \mathrm{vec}_{m,n}(\mathbf{M}) \in \tilde{\mathcal{U}}$. By definition of the vectorization,

$$f_{uv}(\mathbf{M}) = \tilde{f}_s(\mathbf{m}), \quad \text{where } s = s(u, v).$$

This coordinate-wise correspondence underlies the equivalence stated in the lemma.

($\Rightarrow$) Assume $f$ is real-analytic at $\mathbf{Y}$. Then by Definition A.2, there exists $r > 0$ and, for each $(u, v)$, coefficients $\{c_{\mathbf{A}}^{(uv)}\}_{\mathbf{A} \in \mathbb{N}_0^{m \times n}}$ such that:

$$f_{uv}(\mathbf{X}) = \sum_{\mathbf{A} \in \mathbb{N}_0^{m \times n}} c_{\mathbf{A}}^{(uv)} (\mathbf{X} - \mathbf{Y})^{\mathbf{A}}, \qquad \forall \mathbf{X} \in \mathcal{U} : \|\mathbf{X} - \mathbf{Y}\|_{\mathrm{F}} < r. \qquad (12)$$

Using Equation 11, each component $\tilde{f}_s$ of $\tilde{f}$ can be expressed as:

$$\tilde{f}_s(\mathbf{x}) = \sum_{\boldsymbol{\alpha} \in \mathbb{N}_0^{mn}} \tilde{c}_{\boldsymbol{\alpha}}^{(s)} (\mathbf{x} - \mathbf{y})^{\boldsymbol{\alpha}}, \quad \text{where } \tilde{c}_{\boldsymbol{\alpha}_{\mathbf{A}}}^{(s)} := c_{\mathbf{A}}^{(u(s), v(s))}.$$

This series converges for all $\mathbf{x} \in \tilde{\mathcal{U}}$ with $\|\mathbf{x} - \mathbf{y}\|_2 = \|\mathbf{X} - \mathbf{Y}\|_{\mathrm{F}} < r$. Hence, each scalar component of $\tilde{f}$ has a convergent power series at $\mathbf{y}$, proving that $\tilde{f}$ is real-analytic there.

($\Leftarrow$) The reverse direction follows by symmetry: assume $\tilde{f}$ is real-analytic at $\mathbf{y}$, write the expansion at $\mathbf{y}$ using definition Definition A.1, and repeat the argument using Equation 10 to construct component-wise expansions for $f_{uv}$ at $\mathbf{Y}$. $\qquad \square$

**Remark 8.** *Consider the function $f = \mathrm{vec}_{m,n} : \mathbb{R}^{m \times n} \to \mathbb{R}^{mn \times 1}$, which vectorizes an $m \times n$ matrix by stacking its columns. Its corresponding vectorized form is*

$$\tilde{f}(\mathbf{x}) = (\mathrm{vec}_{mn,1} \circ \mathrm{vec}_{m,n} \circ \mathrm{mat}_{m,n})(\mathbf{x}) = \mathrm{vec}_{mn,1}(\mathbf{x}) = \mathbf{x},$$

*since $\mathbf{x} \in \mathbb{R}^{mn}$ is already a column vector . This composition yields the identity map on $\mathbb{R}^{mn}$, which is clearly real analytic. Therefore, by Lemma A.1, both $\mathrm{vec}_{m,n}$ is real analytic, and similarly, so is $\mathrm{mat}_{m,n}$. It is now evident that the composition of two matrix-valued real-analytic function is real-analytic, and we will prove it.*

**Proposition A.3** (Composition on matrix spaces is real-analytic). *Suppose $f : \mathbb{R}^{m \times n} \to \mathbb{R}^{a \times b}$ and $g : \mathbb{R}^{a \times b} \to \mathbb{R}^{p \times q}$ are real-analytic (in the sense of Definition A.2). Then $g \circ f : \mathbb{R}^{m \times n} \to \mathbb{R}^{p \times q}$ is real-analytic.*

*Proof.* Consider the vectorized forms

$$\tilde{f} := \mathrm{vec}_{a,b} \circ f \circ \mathrm{mat}_{m,n} : \mathbb{R}^{mn} \to \mathbb{R}^{ab}, \qquad \tilde{g} := \mathrm{vec}_{p,q} \circ g \circ \mathrm{mat}_{a,b} : \mathbb{R}^{ab} \to \mathbb{R}^{pq}.$$

By Lemma A.1, $f$ is real-analytic iff $\tilde{f}$ is, and $g$ is real-analytic iff $\tilde{g}$ is. Hence $\tilde{f}$ and $\tilde{g}$ are real-analytic maps between Euclidean spaces.

The vectorized form of the composition is

$$\widetilde{g \circ f} = \mathrm{vec}_{p,q} \circ (g \circ f) \circ \mathrm{mat}_{m,n} = \underbrace{\left( \mathrm{vec}_{p,q} \circ g \circ \mathrm{mat}_{a,b} \right)}_{\tilde{g}} \circ \underbrace{\left( \mathrm{vec}_{a,b} \circ f \circ \mathrm{mat}_{m,n} \right)}_{\tilde{f}} = \tilde{g} \circ \tilde{f},$$

where we inserted the identity $(\mathrm{mat}_{a,b} \circ \mathrm{vec}_{a,b})(\mathbf{X}) = \mathbf{X}$. By the vector-space composition property (Proposition A.2), $\tilde{g} \circ \tilde{f}$ is real-analytic on $\mathbb{R}^{mn}$. Applying Lemma A.1 once more, we get that $g \circ f$ is real-analytic. $\qquad\square$

### A.2.3 REAL ANALYTICITY OF COMMON COMPONENTS

We now catalogue the specific functions that appear inside transformer layers, proving each one is real-analytic. These building blocks—polynomials, exponentials, softmax, row normalization, matrix products, Hadamard scaling, and stacking—will be composed in Appendix B to establish the real-analyticity of the full model. Throughout, all maps are defined on $\mathbb{R}^{m \times n}$ (or an open subset thereof), so Definition A.2 applies.

**Proposition A.4** (Polynomials are real-analytic). *Let $p : \mathbb{R}^m \to \mathbb{R}$ be a polynomial in the coordinates of $\mathbf{x} \in \mathbb{R}^m$, i.e., $p(\mathbf{x}) = \sum_{|\boldsymbol{\alpha}| \leq d} a_{\boldsymbol{\alpha}} \mathbf{x}^{\boldsymbol{\alpha}}$ for some $d \in \mathbb{N}_0$ and coefficients $a_{\boldsymbol{\alpha}} \in \mathbb{R}$. Then $p \in C^\omega(\mathbb{R}^m)$.*

*Proof.* Polynomials are $C^\infty$, and $\mathbf{d}^{\boldsymbol{\alpha}} p \equiv 0$ whenever $|\boldsymbol{\alpha}| > d$. Hence the Taylor expansion of $p$ at any $\mathbf{y} \in \mathbb{R}^m$ truncates:

$$p(\mathbf{x}) = \sum_{|\boldsymbol{\alpha}| \leq d} \frac{\mathbf{d}^{\boldsymbol{\alpha}} p(\mathbf{y})}{\boldsymbol{\alpha}!} (\mathbf{x} - \mathbf{y})^{\boldsymbol{\alpha}},$$

which holds for all $\mathbf{x} \in \mathbb{R}^m$ (radius $r = +\infty$). Therefore $p$ is real-analytic. $\qquad\square$

**Proposition A.5** (The exponential is real-analytic). *The map $\exp : \mathbb{R} \to (0, \infty)$ is real-analytic on $\mathbb{R}$.*

*Proof.* Define $E(x) := \sum_{k=0}^{\infty} \frac{x^k}{k!}$. By the ratio test this power series has infinite radius of convergence, hence converges absolutely for all $x \in \mathbb{R}$. Standard results on power series imply that $E$ is $C^\infty$ on $\mathbb{R}$ and can be differentiated termwise within its radius of convergence; in particular, for every $j \in \mathbb{N}_0$,

$$E^{(j)}(x) = \sum_{k=j}^{\infty} \frac{k(k-1)\cdots(k-j+1)}{k!} x^{k-j} = \sum_{r=0}^{\infty} \frac{x^r}{r!} = E(x).$$

Fix $y \in \mathbb{R}$. Taylor's theorem for power series then yields

$$E(x) = \sum_{j=0}^{\infty} \frac{E^{(j)}(y)}{j!} (x - y)^j = E(y) \sum_{j=0}^{\infty} \frac{(x - y)^j}{j!},$$

which is a convergent power series in $x - y$ with infinite radius of convergence. Hence $E$ is real-analytic at every $y \in \mathbb{R}$. As $E$ is the usual exponential function defined by its power series, $\exp$ is real-analytic on $\mathbb{R}$. $\qquad\square$

**Proposition A.6** (The logarithm is real-analytic). *The map $\log : (0, \infty) \to \mathbb{R}$ is real-analytic on $(0, \infty)$.*

*Proof.* For brevity, we present only a proof sketch;

The exponential map $\exp : \mathbb{R} \to (0, \infty)$ is real-analytic with $\exp'(y) \neq 0$ for all $y$. By the real-analytic inverse function theorem (see Krantz & Parks 2002, Thm. 2.3.1), its local inverse $\log$ is real-analytic on $(0, \infty)$. $\qquad \square$

The next three results handle the attention-specific operations: softmax, row normalization (used in the causal projection form), and entrywise matrix polynomials.

**Proposition A.7** (Softmax is real-analytic). *The map* $\mathrm{softmax} : \mathbb{R}^m \to \mathbb{R}^m$ *with components*

$$\mathrm{softmax}_i(\mathbf{x}) \;=\; \frac{e^{\mathbf{x}_i}}{\sum_{j=1}^m e^{\mathbf{x}_j}}, \qquad i = 1, \dots, m,$$

*is real-analytic on* $\mathbb{R}^m$.

*Proof.* Fix $i$. The numerator $\mathbf{x} \mapsto e^{\mathbf{x}_i}$ is the composition of the coordinate projection $\pi_i(\mathbf{x}) = \mathbf{x}_i$ (a linear, hence real-analytic, map) with $\exp$; by Proposition A.5 and the composition rule in Proposition A.1, it is real-analytic. The denominator

$$H(\mathbf{x}) = \sum_{j=1}^m e^{\mathbf{x}_j}$$

is a finite sum of real-analytic functions, hence real-analytic. Moreover, $H(\mathbf{x}) > 0$ for all $\mathbf{x} \in \mathbb{R}^m$ because $e^{x_j} > 0$. Therefore, by the quotient rule in Proposition A.1, the map

$$\mathbf{x} \mapsto \frac{e^{\mathbf{x}_i}}{H(\mathbf{x})}$$

is real-analytic on $\mathbb{R}^m$. Since this holds for each $i = 1, \dots, m$, the vector-valued map $\mathrm{softmax}$ is real-analytic. $\qquad \square$

**Proposition A.8** (Row normalization is real-analytic on positive row-sum domain). *Let*

$$\mathcal{D}_T := \left\{ \mathbf{Y} \in \mathbb{R}^{T \times T} : \mathbf{Y}\mathbf{1}_T \in (0, \infty)^T \right\}.$$

*Define* $\mathrm{RN}(\mathbf{Y}) = \mathrm{diag}(\mathbf{Y}\mathbf{1}_T)^{-1}\mathbf{Y}$ *on* $\mathcal{D}_T$. *Then* $\mathrm{RN} : \mathcal{D}_T \to \mathbb{R}^{T \times T}$ *is real-analytic (in the sense of Definition A.2).*

*Proof.* The map $\mathbf{Y} \mapsto \mathbf{s} := \mathbf{Y}\mathbf{1}_T$ is linear, hence real-analytic. On $(0, \infty)^T$, the entrywise reciprocal $\mathbf{s} \mapsto \mathbf{s}^{\odot(-1)}$ is real-analytic (componentwise $t \mapsto 1/t$). The map $\mathbf{s} \mapsto \mathrm{diag}(\mathbf{s})$ is linear. Matrix multiplication $(\mathbf{A}, \mathbf{Y}) \mapsto \mathbf{A}\mathbf{Y}$ is real-analytic (Proposition A.10). Composing these gives $\mathrm{RN}(\mathbf{Y}) = \mathrm{diag}(\mathbf{Y}\mathbf{1}_T)^{-1}\mathbf{Y}$ real-analytic on the open set $\mathcal{D}_T$. $\qquad \square$

**Proposition A.9** (Entrywise matrix polynomials are real-analytic). *Fix* $m, n \in \mathbb{N}$. *For coefficients* $\{c_\mathbf{A} \in \mathbb{R}\}_{\mathbf{A} \in \mathbb{N}_0^{m \times n}}$ *and some* $d \in \mathbb{N}_0$, *define the function* $p : \mathbb{R}^{m \times n} \to \mathbb{R}$ *by:*

$$p(\mathbf{X}) = \sum_{|\mathbf{A}| \leq d} c_\mathbf{A} \, \mathbf{X}^\mathbf{A}, \tag{13}$$

*where* $\mathbf{X}^\mathbf{A} = \prod_{u=1}^m \prod_{v=1}^n \mathbf{X}_{uv}^{\mathbf{A}_{uv}}$ *as defined in the multi-index notation above. Then* $p$ *is real-analytic on* $\mathbb{R}^{m \times n}$ *(in the sense of Definition A.2).*

*Moreover, if* $f : \mathbb{R}^{m \times n} \to \mathbb{R}^{a \times b}$ *has component functions* $f_{ij}$ *of the form Equation 13, then* $f$ *is real-analytic.*

*Proof.* Consider the vectorized form $\tilde{p} := p \circ \mathrm{mat}_{m,n} : \mathbb{R}^{mn} \to \mathbb{R}$. Using the coordinate identification from equation 11-equation 10, each monomial satisfies

$$\left( \mathrm{mat}_{m,n}(\mathbf{x}) \right)^\mathbf{A} = \mathbf{x}^{\boldsymbol{\alpha}_\mathbf{A}},$$

where $\boldsymbol{\alpha_A} = \mathrm{vec}_{m,n}(\mathbf{A})$. Hence:

$$\tilde{p}(\mathbf{x}) = \sum_{|\mathbf{A}| \leq d} c_{\mathbf{A}}\, \mathbf{x}^{\boldsymbol{\alpha_A}},$$

which is a standard multivariate polynomial in $\mathbf{x} \in \mathbb{R}^{mn}$. By Proposition A.4, such functions are real-analytic on all of $\mathbb{R}^{mn}$, so $\tilde{p} \in C^{\omega}(\mathbb{R}^{mn})$. By Lemma A.1, this implies $p$ is real-analytic on $\mathbb{R}^{m \times n}$.

For the second claim, observe that if each $f_{ij}$ is a scalar polynomial of the form Equation 13, then each $f_{ij}$ is real-analytic by the argument above. Hence, by Definition A.2, $f$ is real analytic. $\qquad\square$

Finally, we record the algebraic operations—matrix multiplication, Hadamard scaling, concatenation, and noncommutative matrix polynomials—that allow us to compose the above primitives into full transformer layers.

**Proposition A.10** (Matrix product of real-analytic factors)**.** *Let the functions* $f : \mathbb{R}^{m \times n} \to \mathbb{R}^{p \times r}$ *and* $g : \mathbb{R}^{m \times n} \to \mathbb{R}^{r \times q}$ *be real-analytic. Then,* $h : \mathbb{R}^{m \times n} \to \mathbb{R}^{p \times q}$ *defined as* $h(\mathbf{X}) = f(\mathbf{X})\, g(\mathbf{X})$, *is real-analytic on* $\mathbb{R}^{m \times n}$.

*Proof.* For each $(i,j) \in [p] \times [q]$, it holds that $h_{ij}(\mathbf{X}) = \sum_{k=1}^{r} f_{ik}(\mathbf{X})\, g_{kj}(\mathbf{X})$.

Each factor $f_{ik}$ and $g_{kj}$ is a real-analytic scalar map by assumption; their product is real-analytic by Proposition A.1, and a finite sum of real-analytic functions is real-analytic. Thus every $h_{ij}$ is real-analytic, hence $h$ is real-analytic. $\qquad\square$

**Proposition A.11** (Hadamard (element-wise) scaling)**.** *Let* $\mathbf{A} \in \mathbb{R}^{m \times n}$ *be a fixed matrix. Then, the map* $f : \mathbb{R}^{m \times n} \to \mathbb{R}^{m \times n}$ *defined as* $f(X) = \mathbf{A} \odot \mathbf{X}$ *is real-analytic on* $\mathbb{R}^{m \times n}$.

*Proof.* Componentwise, $(\mathbf{A} \odot \mathbf{X})_{ij} = \mathbf{A}_{ij}\, \mathbf{X}_{ij}$ is a product of a constant and a coordinate function, hence a polynomial (degree $\leq 1$) and thus real-analytic. $\qquad\square$

**Proposition A.12** (Concatenation/stacking of real-analytic blocks)**.** *Let* $f_{\ell} : \mathbb{R}^{m \times n} \to \mathbb{R}^{p \times q_{\ell}}$ *be real-analytic for* $\ell \in [L]$. *The horizontal concatenation operation* $g : \mathbb{R}^{m \times n} \to \mathbb{R}^{p \times (q_1 + \cdots + q_L)}$, *defined as:*

$$g(\mathbf{X}) = \begin{bmatrix} f_1(\mathbf{X}) & f_2(\mathbf{X}) & \cdots & f_L(\mathbf{X}) \end{bmatrix}$$

*is real-analytic. Likewise, if* $f_{\ell} : \mathbb{R}^{m \times n} \to \mathbb{R}^{p_{\ell} \times q}$ *are real-analytic, then the vertical stacking operation* $h : \mathbb{R}^{m \times n} \to \mathbb{R}^{(p_1 + \cdots + p_L) \times q}$, *defined as:*

$$h(\mathbf{X}) = \begin{bmatrix} f_1(\mathbf{X})^{\top} & f_2(\mathbf{X})^{\top} & \cdots & f_L(\mathbf{X})^{\top} \end{bmatrix}^{\top}$$

*is real-analytic.*

*Proof.* Each scalar component of $g$ (respectively $h$) is exactly one scalar component of some $f_{\ell}$, hence real-analytic. Therefore $g$ and $h$ are real-analytic by definition Definition A.2. $\qquad\square$

**Proposition A.13** (Noncommutative matrix polynomials are real-analytic)**.** *Let* $n, p, q \in \mathbb{N}$, *let* $\mathbf{X} \in \mathbb{R}^{n \times n}$, *and fix coefficient matrices* $\mathbf{A}_k \in \mathbb{R}^{p \times n}$ *and* $\mathbf{B}_k \in \mathbb{R}^{n \times q}$ *for* $k = 0, \ldots, d$. *Define*

$$f(\mathbf{X}) := \sum_{k=0}^{d} \mathbf{A}_k\, \mathbf{X}^k\, \mathbf{B}_k \;\in\; \mathbb{R}^{p \times q}, \qquad \mathbf{X}^0 := \mathbf{I}_n, \;\; \mathbf{X}^{k+1} := \mathbf{X}^k \mathbf{X}.$$

*Then $f$ is real analytic in the sense of Definition A.2.*

*Proof.* The identity map $\mathbf{X} \mapsto \mathbf{X}$ is linear, hence a degree-1 entrywise polynomial; by Proposition A.9 it is real-analytic. Assume $\mathbf{X} \mapsto \mathbf{X}^k$ is real-analytic. With $f(\mathbf{X}) = \mathbf{X}^k$ and $g(\mathbf{X}) = \mathbf{X}$, Proposition A.10 yields $\mathbf{X}^{k+1} = f(\mathbf{X})g(\mathbf{X})$ real-analytic; by induction, all powers $\mathbf{X} \mapsto \mathbf{X}^k$ are real-analytic.

For each $k$, left/right multiplication by fixed matrices preserves real-analyticity via Proposition A.10: since the constant maps $\mathbf{X} \mapsto \mathbf{A}_k$ and $\mathbf{X} \mapsto \mathbf{B}_k$ are real-analytic (components are constant polynomials), the composition $\mathbf{X} \mapsto \mathbf{A}_k \mathbf{X}^k \mathbf{B}_k$ is real-analytic. Finally, $f$ is a finite sum of real-analytic maps, hence real-analytic by closure under addition (apply Proposition A.1 componentwise). $\qquad\square$

**Remark 9.** *We highlight several standard constructions that yield real-analytic maps, omitting proofs for brevity:*

- *Affine and bilinear maps. Functions of the form $\mathbf{X} \mapsto \mathbf{A}\mathbf{X}\mathbf{B} + \mathbf{C}$ are real-analytic, as they are obtained via matrix multiplication and addition of constant matrices (Proposition A.10, Proposition A.1).*

- *Algebraic expressions in $\mathbf{X}$. Any expression constructed from $\mathbf{X}$ using finitely many additions and matrix multiplications with fixed coefficient matrices, e.g. $\mathbf{A}_0 + \mathbf{A}_1\mathbf{X}\mathbf{B}_1 + \mathbf{A}_2\mathbf{X}\mathbf{B}_2\mathbf{X}\mathbf{C}_2$- defines a real-analytic map. This follows from repeated application of Proposition A.10 and closure under addition.*

- *Scalar polynomial invariants. Coordinate functions $\mathbf{X}_{ij}$, the trace $\mathrm{tr}(\mathbf{X})$, all principal and non-principal minors, and the determinant $\det(\mathbf{X})$ are scalar polynomials in the entries of $\mathbf{X}$, and hence real-analytic by Proposition A.9.*

## A.3   DIFFERENTIAL, MEASURE-THEORETIC, AND TOPOLOGICAL TOOLS

This subsection collects the minimal calculus, measure, and topology we will use later. In finite dimensions, Fréchet derivatives let us speak uniformly about Jacobians and Hessians; basic Euclidean topology lets us control neighborhoods and compactness; the inverse function theorem gives local invertibility; and pushforwards/absolute continuity formalize how distributions transform under measurable maps.

**Definition A.5** (Fréchet derivative (Luenberger, 1997, §7.2-§7.3)). *Let $\mathcal{U} \subseteq \mathbb{R}^m$ open, and consider a function $f : \mathcal{U} \to \mathbb{R}^n$. We say that $f$ is Fréchet differentiable at $\mathbf{x} \in \mathcal{U}$ if there exists a bounded linear map $\mathbf{A} : \mathbb{R}^m \to \mathbb{R}^n$ such that*

$$\lim_{\|\mathbf{h}\|_2 \to 0} \frac{\|f(\mathbf{x}+\mathbf{h}) - f(\mathbf{x}) - \mathbf{A}\mathbf{h}\|_2}{\|\mathbf{h}\|_2} = 0.$$

*The unique operator $\mathbf{A}$ is denoted by $Df(\mathbf{x})$ and called the (Fréchet) derivative of $f$ at $\mathbf{x}$.*

**Definition A.6** (Second Fréchet derivative (Magnus & Neudecker, 2019, Ch. 18)). *Let $\mathcal{U} \subseteq \mathbb{R}^m$ open, and consider a function $f : \mathcal{U} \to \mathbb{R}^n$. Suppose $f$ is Fréchet differentiable at $\mathbf{x}$. The second Fréchet derivative of $f$ at $\mathbf{x}$ is the bounded bilinear map $D^2 f(\mathbf{x}) : \mathbb{R}^m \times \mathbb{R}^m \to \mathbb{R}^n$ defined as:*

$$D^2 f(\mathbf{x})[\mathbf{h}, \mathbf{k}] := \lim_{t \to 0} \frac{Df(\mathbf{x}+t\mathbf{h})[\mathbf{k}] - Df(\mathbf{x})[\mathbf{k}]}{t}.$$

**Proposition A.14** (Connection to the Hessian). *If $f : \mathcal{U} \to \mathbb{R}$ is $C^2$, then $D^2 f(\mathbf{x})$ is symmetric (Arora et al., 2021, Thm. 5.1) and can represented by the Hessian matrix $\nabla^2 f(\mathbf{x})$:*

$$D^2 f(\mathbf{x})[\mathbf{h}, \mathbf{k}] \;=\; \mathbf{h}^\top \left(\nabla^2 f(\mathbf{x})\right) \mathbf{k},$$

*as noted in Magnus & Neudecker 2019, Ch. 18.*

**Euclidean topology.**   The following standard definitions and facts about $\mathbb{R}^p$ are used in the local-to-global measure argument in Appendix C.

**Definition A.7** (Closure of a set in $\mathbb{R}^p$). *Let $\mathcal{U} \subseteq \mathbb{R}^p$. The closure of $\mathcal{U}$, denoted $\overline{\mathcal{U}}$, is the smallest closed subset of $\mathbb{R}^p$ containing $\mathcal{U}$.*

**Definition A.8** (Euclidean balls in $\mathbb{R}^p$). *Fix $p \in \mathbb{N}$ and equip $\mathbb{R}^p$ with the Euclidean norm $\|\cdot\|_2$. For $\mathbf{x} \in \mathbb{R}^p$ and $r > 0$ we define:*

$$B(\mathbf{x}, r) := \{\, \mathbf{y} \in \mathbb{R}^p : \|\mathbf{y} - \mathbf{x}\|_2 < r \,\}$$
$$\overline{B}(\mathbf{x}, r) := \{\, \mathbf{y} \in \mathbb{R}^p : \|\mathbf{y} - \mathbf{x}\|_2 \le r \,\}$$

*In $\mathbb{R}^p$ with the Euclidean topology one has $\overline{B}(\mathbf{x}, r) = \overline{B(\mathbf{x}, r)}$, i.e. the closed ball equals the topological closure of the open ball.*

**Definition A.9** (Second-countable subspace of $\mathbb{R}^p$ (Munkres, 2000, §30)). *Let $\mathcal{X} \subseteq \mathbb{R}^p$ be equipped with the subspace topology $\tau_{\mathcal{X}} := \{\mathcal{U} \cap \mathcal{X} : \mathcal{U} \text{ open in } \mathbb{R}^p\}$. We say $\mathcal{X}$ is second-countable if there exists a countable family $\mathcal{F} \subseteq \tau_X$ such that every $\mathcal{O} \in \tau_{\mathcal{X}}$ is a union of members of $\mathcal{F}$. Equivalently, the countable family*

$$\mathcal{F}_{\mathbb{Q}} \;:=\; \{\, B(\mathbf{x}, r) \cap \mathcal{X} : \mathbf{x} \in \mathbb{Q}^p, r \in \mathbb{Q}_{>0} \,\},$$

*is a basis for $\tau_{\mathcal{X}}$.*

**Proposition A.15** (Standard facts for $\mathbb{R}^p$)**.** *Fix $p \in \mathbb{N}$. The following hold:*

1. **Hausdorff** *(Aitken, 2020, Prop. 18): $\mathbb{R}^p$ with its Euclidean metric is Hausdorff.*

2. **Heine-Borel** *(Munkres, 2000, Thm. 27.3): A subset of $\mathbb{R}^p$ is compact iff it is closed and bounded; in particular, each closed Euclidean ball $\overline{B}(x, r)$ is compact.*

3. **Second countability** *(Munkres, 2000, §13 and Thm. 30.2) : $\mathbb{R}$ has a countable base (intervals with rational endpoints); hence $\mathbb{R}^p$, being a finite product of second-countable spaces, is second-countable. Moreover, subspaces of second-countable spaces are second-countable.*

4. **Lindelöf consequence***(Munkres, 2000, Thm. 30.3(a)): Every second-countable space is Lindelöf; consequently, every open cover of any subspace of $\mathbb{R}^p$ admits a countable sub-cover.*

5. **Local compactness of** $\mathbb{R}^p$*(Munkres, 2000, Thm. 29.2): For any $\mathbf{x} \in \mathbb{R}^p$ and open neighborhood $\mathcal{W} \ni \mathbf{x}$, there exists $\varepsilon > 0$ with $\overline{B}(\mathbf{x}, \varepsilon) \subseteq \mathcal{W}$, and $\overline{B}(\mathbf{x}, \varepsilon)$ is compact by Heine-Borel; hence $\mathbb{R}^p$ is locally compact. Furthermore, in a Hausdorff space, local compactness is equivalent to shrinking neighborhoods with compact closures: for every neighborhood $\mathcal{W} \ni \mathbf{x}$ there exists an open $\mathcal{V}$ with $\mathbf{x} \in \mathcal{V} \subseteq \overline{\mathcal{V}} \subseteq \mathcal{W}$ and $\overline{\mathcal{V}}$ compact.*

**Invertibility and measure transport.** The inverse function theorem and the pushforward formalism are the two tools that connect local diffeomorphism charts to global measure-preservation statements.

**Definition A.10** ($C^k$ diffeomorphism Spivak 1971, Ch. 5)**.** *Let $U, V \subseteq \mathbb{R}^p$ be open sets and let $k \in \mathbb{N} \cup \{\infty\}$. A map $f : U \to V$ is a $C^k$ **diffeomorphism** if:*

1. *$f$ is bijective;*

2. *$f$ is $C^k$ (all partial derivatives up to order $k$ exist and are continuous);*

3. *the inverse map $f^{-1} : V \to U$ is $C^k$.*

*When $k = 1$ we simply say* diffeomorphism. *Equivalently, a $C^k$ diffeomorphism is a bijective $C^k$ map whose inverse is also $C^k$.*

**Theorem A.2** (Inverse Function Theorem Rudin 1976, Thm. 9.24)**.** *Let $\mathcal{U} \subset \mathbb{R}^p$ be open and $f : \mathcal{U} \to \mathbb{R}^p$ be $C^1$. Suppose $\mathbf{a} \in \mathcal{U}$ satisfies $\det Df(\mathbf{a}) \neq 0$. Then there exist open sets $\mathcal{U}_0 \subset \mathcal{U}$ with $\mathbf{a} \in \mathcal{U}_0$ and $\mathcal{V}_0 \subset \mathbb{R}^p$ with $f(\mathbf{a}) \in \mathcal{V}_0$ such that*

$$f\big|_{\mathcal{U}_0} : \mathcal{U}_0 \to \mathcal{V}_0$$

*is a $C^1$-diffeomorphism. Moreover, the inverse $f^{-1} : \mathcal{V}_0 \to \mathcal{U}_0$ is $C^1$ and*

$$D\big(f^{-1}\big)(f(\mathbf{x})) \;=\; \big(Df(\mathbf{x})\big)^{-1} \qquad \forall \, \mathbf{x} \in \mathcal{U}_0.$$

**Remark 10.** *In Theorem A.2 we assume $f : \mathcal{U} \subseteq \mathbb{R}^p \to \mathbb{R}^p$, so the Jacobian $Df(\mathbf{a})$ is a $p \times p$ (square) matrix. In this setting,*

$$\det Df(\mathbf{a}) \neq 0 \quad \Longleftrightarrow \quad Df(\mathbf{a}) \text{ is invertible,}$$

*and this is exactly the hypothesis that yields a local $C^1$ inverse.*

**Definition A.11** (Pushforward and absolute continuity (Folland, 1999, §3.2))**.** *Consider a Borel-measurable map $T : \mathbb{R}^p \to \mathbb{R}^p$ and let $\mu$ be a Borel measure on $\mathbb{R}^p$. The pushforward measure $T_{\#}\mu$ is the Borel measure on $\mathbb{R}^p$ defined by*

$$T_{\#}\mu(\mathcal{U}) \;:=\; \mu\big(T^{-1}(\mathcal{U})\big), \qquad \mathcal{U} \in \mathcal{B}(\mathbb{R}^p).$$

*If $\nu$ is another Borel measure on $\mathbb{R}^p$, we say $T_{\#}\mu$ is absolutely continuous with respect to $\nu$, and write $T_{\#}\mu \ll \nu$, if for every Borel set $\mathcal{U} \in \mathcal{B}(\mathbb{R}^p)$:*

$$\nu(\mathcal{U}) = 0 \Longrightarrow T_{\#}\mu(\mathcal{U}) = 0.$$

*In particular, for Lebesgue measure $\mathrm{Leb}_p$, to prove $T_{\#}\mu \ll \mathrm{Leb}_p$ for every $\mu \ll \mathrm{Leb}_p$, it suffices to verify that*

$$\mathrm{Leb}_p(\mathcal{U}) = 0 \;\Longrightarrow\; \mathrm{Leb}_p\big(T^{-1}(\mathcal{U})\big) = 0 \;\; \text{for all Borel } \mathcal{U} \subseteq \mathbb{R}^p.$$

# B  TRANSFORMER LANGUAGE MODEL

This appendix section gives a concise, shape-accurate specification of the decoder-only Transformer we analyze. We include it both to keep the paper self-contained and because the measure-zero arguments later hinge on architecture-dependent witnesses and exact dimension bookkeeping. We begin with token and positional embeddings (Definition B.3), define self-attention and its causal variants (Definition B.5, Definition B.6, Definition B.7), assemble multi-head attention, layer normalization, and an MLP into a pre-LN residual block (Definition B.8, Definition B.9, Definition B.4, Definition B.11), stack $L$ such blocks to obtain the model (Definition B.12), and conclude with the unembedding+softmax head (Definition B.10), isolating the last-token representation used in downstream proofs (Equation 29).

**Input processing.** The first stage of the model maps a discrete token sequence into a continuous matrix representation via learned embeddings.

**Definition B.1** (Token Embedding Layer). *Let $\mathcal{V}$ be a vocabulary, and let $d \in \mathbb{N}$ be the embedding dimension. For any input sequence $s = \langle s_1, \ldots, s_T \rangle \in \mathcal{V}^{\leq K}$, the Token Embedding Layer is the function defined as:*

$$\mathrm{E}(s) = (\mathbf{E}_{s_1}, \ldots, \mathbf{E}_{s_T})^\top \in \mathbb{R}^{T \times d}, \tag{14}$$

*where $\mathbf{E} \in \mathbb{R}^{|\mathcal{V}| \times d}$ is a trainable embedding matrix indexed by elements of $\mathcal{V}$, and $\mathbf{E}_{s_i} \in \mathbb{R}^d$ denotes the embedding vector for token $s_i$.*

*This mapping is applied element-wise and is independent of the sequence length $T$.*

**Definition B.2** (Positional Embedding Layer). *Let $\mathcal{V}$ be a vocabulary, and let $d \in \mathbb{N}$ be the embedding dimension. For any input sequence $s = \langle s_1, \ldots, s_T \rangle \in \mathcal{V}^{\leq K}$ with $T = |s|$, the (learned absolute) Positional Embedding Layer is the function defined as:*

$$\mathrm{PE}(s) = (\mathbf{P}_1, \ldots, \mathbf{P}_T)^\top \in \mathbb{R}^{T \times d}, \tag{15}$$

*where $\mathbf{P} \in \mathbb{R}^{K \times d}$ is a trainable matrix indexed by positions $i \in [K]$, and $\mathbf{P}_i \in \mathbb{R}^d$ denotes the embedding vector for position $i$. This mapping depends only on positions (not on token identities) and returns the first $T$ rows of $\mathbf{P}$.*

**Definition B.3** (Embedding Layer). *Let $\mathcal{V}$ be a vocabulary, $K \in \mathbb{N}$ a context bound, and $d \in \mathbb{N}$ the embedding width. For any input sequence $s = \langle s_1, \ldots, s_T \rangle \in \mathcal{V}^{\leq K}$ with $T = |s|$, define the embedding layer as the sum of the token and positional embeddings:*

$$\mathrm{Emb}(s) := \mathrm{E}(s) + \mathrm{PE}(s) = \left( \mathbf{E}_{s_1} + \mathbf{P}_1, \ldots, \mathbf{E}_{s_T} + \mathbf{P}_T \right)^\top \in \mathbb{R}^{T \times d}, \tag{16}$$

*where $\mathbf{E} \in \mathbb{R}^{|\mathcal{V}| \times d}$ is the trainable token-embedding matrix and $\mathbf{P} \in \mathbb{R}^{K \times d}$ is the trainable positional-embedding matrix.*

**Sub-layer modules.** The Transformer block is built from four reusable sub-layers—an MLP, (causal) self-attention, multi-head attention, and layer normalization—each defined next.

**Definition B.4** (Multi-Layer Perceptron). *A Multi-Layer Perceptron (MLP) with $M$ layers is a function $\mathrm{mlp}_M : \mathbb{R}^{d_0} \to \mathbb{R}^{d_M}$, defined recursively as:*

$$\mathbf{h}^{(1)} = \mathbf{W}^{(1)} \mathbf{x} + \mathbf{b}^{(1)} \tag{17}$$

$$\mathbf{h}^{(m)} = \mathbf{W}^{(m)} \sigma(\mathbf{h}^{(m-1)}) + \mathbf{b}^{(m)}, \ m \geq 2 \tag{18}$$

$$\mathrm{mlp}_M(\mathbf{x}) = \mathbf{h}^{(M)} \tag{19}$$

*where $\mathbf{x} \in \mathbb{R}^{d_0}$ is the input, $\{\mathbf{W}^{(m)} \in \mathbb{R}^{d_m \times d_{m-1}}\}_{m=1}^M$ and $\{\mathbf{b}^{(m)} \in \mathbb{R}^{d_m}\}_{m=1}^M$ are trainable parameters and $\sigma$ is an activation function.*

**Definition B.5** (Self-Attention). *A Self-Attention module is a function $\boldsymbol{\eta} : \mathbb{R}^{T \times d_{\mathrm{in}}} \to \mathbb{R}^{T \times d_\eta}$, defined as:*

$$\boldsymbol{\eta}(\mathbf{X} \,;\, \mathbf{Q}, \mathbf{K}, \mathbf{V}) = \mathrm{softmax}\left( \frac{(\mathbf{X}\mathbf{Q})(\mathbf{X}\mathbf{K})^\top}{\sqrt{d_\eta}} \right) \mathbf{X}\mathbf{V}, \tag{20}$$

*where $\mathbf{X} \in \mathbb{R}^{T \times d_{\mathrm{in}}}$ is the input, $\mathbf{Q}, \mathbf{K}, \mathbf{V} \in \mathbb{R}^{d_{\mathrm{in}} \times d_\eta}$ are trainable parameters (query, key, and value matrices), $\mathrm{softmax}$ is applied row-wise, $d_\eta$ is the attention dimension (typically $d_\eta < d_{\mathrm{in}}$), and $T$ is the sequence length.*

**Definition B.6** (Causal Self-Attention, masked form). *Define the "causal mask" $\mathbf{M} \in \overline{\mathbb{R}}^{T \times T}$ as:*

$$\mathbf{M}_{ij} = \begin{cases} 0, & j \leq i, \\ -\infty, & j > i \end{cases}$$

*Then, a Causal Self-Attention module is a function $\tilde{\boldsymbol{\eta}} : \mathbb{R}^{T \times d_{\mathrm{in}}} \to \mathbb{R}^{T \times d_\eta}$, defined as:*

$$\tilde{\boldsymbol{\eta}}(\mathbf{X}\,;\mathbf{Q}, \mathbf{K}, \mathbf{V}) = \mathrm{softmax}\left( \frac{(\mathbf{X}\mathbf{Q})(\mathbf{X}\mathbf{K})^\top}{\sqrt{d_\eta}} + \mathbf{M} \right) \mathbf{X}\mathbf{V}, \tag{21}$$

*where $\mathbf{X} \in \mathbb{R}^{T \times d_{\mathrm{in}}}$ is the input, $\mathbf{Q}, \mathbf{K}, \mathbf{V} \in \mathbb{R}^{d_{\mathrm{in}} \times d_\eta}$ are trainable parameters (query, key, and value matrices), $\mathrm{softmax}$ is applied row-wise, $d_\eta$ is the attention dimension (typically $d_\eta < d_{\mathrm{in}}$), and $T$ is the sequence length.*

**Definition B.7** (Causal Self-Attention, projection form). *Define the unit lower-triangular matrix $\mathbf{L} \in \mathbb{R}^{T \times T}$ as $\mathbf{L}_{ij} = \mathbb{I}_{\{j \leq i\}}$ and consider the row normalization operation $\mathrm{RN} : \boldsymbol{\mathcal{D}}_T \to \mathbb{R}^{T \times T}$ of Proposition A.8. Then, a Causal Self-Attention module is a function $\tilde{\boldsymbol{\eta}} : \mathbb{R}^{T \times d_{\mathrm{in}}} \to \mathbb{R}^{T \times d_\eta}$, defined as:*

$$\tilde{\boldsymbol{\eta}}(\mathbf{X}\,;\mathbf{Q}, \mathbf{K}, \mathbf{V}) = \mathrm{RN}\left( \mathbf{L} \odot \exp\left( \frac{(\mathbf{X}\mathbf{Q})(\mathbf{X}\mathbf{K})^\top}{\sqrt{d_\eta}} \right) \right) \mathbf{X}\mathbf{V}, \tag{22}$$

*where $\mathbf{X} \in \mathbb{R}^{T \times d_{\mathrm{in}}}$ is the input, $\mathbf{Q}, \mathbf{K}, \mathbf{V} \in \mathbb{R}^{d_{\mathrm{in}} \times d_\eta}$ are trainable parameters (query, key, and value matrices), $\mathrm{RN}$ is applied row-wise, $d_\eta$ is the attention dimension (typically $d_\eta < d_{\mathrm{in}}$), and $T$ is the sequence length.*

**Remark 11.** *Consider $\mathbf{Z} = \frac{1}{\sqrt{d_\eta}}(\mathbf{X}\mathbf{Q})(\mathbf{X}\mathbf{K})^\top$. Since $\mathbf{L}_{ii} = 1$ for all $i \in [T]$, we have that $\left[\mathbf{L} \odot \exp \mathbf{Z}\right]_{ii} = e^{\mathbf{Z}_{ii}} > 0$, hence the row sum $\sum_{j \leq i} e^{\mathbf{Z}_{ij}} \geq e^{\mathbf{Z}_{ii}} > 0$ and $\mathrm{RN}$ is well-defined.*

**Definition B.8** (Multi-Head Self-Attention). *A Multi-Head Self-Attention module with $H$ heads is a function $\mathrm{attn}_H : \mathbb{R}^{T \times d_{\mathrm{in}}} \to \mathbb{R}^{T \times d_{\mathrm{out}}}$, defined using the Self-Attention map from Definition B.5 or Definition B.7 with different parameter sets per head:*

$$\boldsymbol{\eta}_h(\mathbf{X}) = \boldsymbol{\eta}(\mathbf{X}\,;\mathbf{Q}^{(h)}, \mathbf{K}^{(h)}, \mathbf{V}^{(h)}), \quad h \in [H], \tag{23}$$

$$\mathrm{attn}_H(\mathbf{X}) = \left[\boldsymbol{\eta}_1(\mathbf{X}), \ldots, \boldsymbol{\eta}_H(\mathbf{X})\right] \mathbf{W}^O, \tag{24}$$

*where $\{\mathbf{Q}^{(h)}, \mathbf{K}^{(h)}, \mathbf{V}^{(h)} \in \mathbb{R}^{d_{\mathrm{in}} \times d_\eta}\}_{h=1}^H$ are the head-specific parameters and $\mathbf{W}^O \in \mathbb{R}^{Hd_\eta \times d_{\mathrm{out}}}$ is the output projection matrix.*

**Definition B.9** (Layer Normalization). *Layer Normalization is a function $\mathrm{LN} : \mathbb{R}^d \to \mathbb{R}^d$, defined as:*

$$\mathrm{LN}(\mathbf{x}) = \boldsymbol{\gamma} \odot \frac{\mathbf{x} - \mu_\mathbf{x} \mathbf{1}_d}{\sqrt{\sigma_\mathbf{x}^2 + \varepsilon}} + \boldsymbol{\beta}, \tag{25}$$

*where $\mathbf{x} \in \mathbb{R}^d$ is the input, $\mu_\mathbf{x} = \frac{1}{d}\sum_{i=1}^d \mathbf{x}_i$ and $\sigma_\mathbf{x}^2 = \frac{1}{d}\sum_{i=1}^d (\mathbf{x}_i - \mu_\mathbf{x})^2$ are the mean and variance of $\mathbf{x}$, vectors $\boldsymbol{\beta}, \boldsymbol{\gamma} \in \mathbb{R}^d$ are learnable parameters, and $\varepsilon \in \mathbb{R}^+$ is a small constant that ensures we don't divide by zero.*

**Definition B.10** (Unembedding Layer). *Let $\mathcal{V}$ be a vocabulary and $d \in \mathbb{N}$ and $\mathbf{U} \in \mathbb{R}^{|\mathcal{V}| \times d}$ be a trainable projection matrix. Define the unembedding map $\mathrm{UnEmb} : \mathbb{R}^d \to \mathbb{R}^{|\mathcal{V}|}$ by*

$$\mathrm{UnEmb}(\mathbf{h}) := \mathrm{softmax}\big(\mathbf{U}\,\mathrm{LN}(\mathbf{h})\big), \qquad \mathbf{h} \in \mathbb{R}^d.$$

**Full architecture assembly.** With all sub-layers in place, we assemble them into a single pre-LN residual block, stack $L$ such blocks into the Transformer backbone, and append the unembedding head to form the complete language model.

**Definition B.11** (Transformer Block). *A Transformer Block consists of a composition of a Multi-Head Self-Attention layer with $H$ heads (Definition B.8) and an MLP with $M$ layers (Definition B.4), each preceded by layer normalization (Definition B.9) and wrapped with residual connections. Given an input $\mathbf{X} \in \mathbb{R}^{T \times d}$, the output $\mathrm{TB}(\mathbf{X}) \in \mathbb{R}^{T \times d}$ is computed as:*

$$\mathbf{H} = \mathbf{X} + \mathrm{attn}_H(\overline{\mathbf{X}}) \tag{26}$$

$$\mathrm{TB}(\mathbf{X}) = \mathbf{H} + \mathrm{mlp}_M(\overline{\mathbf{H}}), \tag{27}$$

where $\overline{\mathbf{X}}, \overline{\mathbf{H}} \in \mathbb{R}^{T \times d}$ are the results of applying layer normalization row-wise to $\mathbf{X}$ and $\mathbf{H}$, respectively, each with its own set of learnable parameters and $\mathrm{mlp}_M$ is applied row-wise. All sub-layer parameters are dimensioned appropriately.

**Definition B.12** (Transformer). *Fix $L \in \mathbb{N}$. For each $\ell \in [L]$, let $\mathrm{TB}^{(\ell)} : \mathbb{R}^{T \times d} \to \mathbb{R}^{T \times d}$ denote a Transformer Block (Definition B.11) with its own parameters. Define the module*

$$\mathrm{Tr}_T := \mathrm{TB}^{(L)} \circ \cdots \circ \mathrm{TB}^{(1)}.$$

*Each $\mathrm{TB}^{(\ell)}$ maps $\mathbb{R}^{T \times d} \to \mathbb{R}^{T \times d}$, so the residual additions in Definition B.11 are dimensionally valid at every depth.*

**Definition B.13** (Transformer Language Model). *Let $\mathcal{V}$ denote a finite vocabulary and $K \in \mathbb{N}$ a fixed context length. A* Transformer Language Model *with $L$ layers is the composition of an embedding layer (Definition B.3), a Transformer with $L$ blocks (Definition B.12), and an Unembedding Layer (Definition B.10).*

*Formally, it is a parameterized function*

$$f : \mathcal{V}^{\leq K} \times \mathbb{R}^p \to \Delta^{|\mathcal{V}|-1}$$

*defined as follows. Without loss of generality, consider $\boldsymbol{\theta} = (\boldsymbol{\theta}_1 \in \mathbb{R}^{p_1}, \boldsymbol{\theta}_2 \in \mathbb{R}^{p_2}, \boldsymbol{\theta}_3 \in \mathbb{R}^{p_3}) \in \mathbb{R}^p$, which collects all the model parameters.*

*For an input sequence $\mathrm{s} = \langle \mathrm{s}_1, \ldots, \mathrm{s}_T \rangle$ with $T \leq K$:*

$$\mathbf{H}(\mathrm{s}\,;\boldsymbol{\theta}) = \mathrm{Emb}(\mathrm{s}\,;\boldsymbol{\theta}_1) \qquad \text{(embedding)} \tag{28}$$

$$\mathbf{r}(\mathrm{s}\,;\boldsymbol{\theta}) = \left( \mathrm{Tr}_{|\mathrm{s}|}\Big( \mathbf{H}(\mathrm{s}\,;\boldsymbol{\theta})\,;\boldsymbol{\theta}_2 \Big) \right)_{|\mathrm{s}|} \qquad \text{(last-token representation)} \tag{29}$$

$$f(\mathbf{s}\,;\boldsymbol{\theta}) = \mathrm{UnEmb}\Big( \mathbf{r}(\mathrm{s}\,;\boldsymbol{\theta})\,;\boldsymbol{\theta}_3 \Big) \qquad \text{(next-token prediction)} \tag{30}$$

*Then, the probability of the next-token being $\mathcal{V}_i$ is given by:*

$$\Pr[\, s_{T+1} = \mathcal{V}_i \mid \mathrm{s}\,] = \big(f(\mathrm{s}\,;\boldsymbol{\theta})\big)_i, \quad \forall i \in [|\mathcal{V}|]. \tag{31}$$

**Verification of real-analyticity.** We close this section by showing that every module defined above is jointly real-analytic in its inputs and parameters. This is the technical property that lets the measure-zero arguments in Appendix C go through. We first record the equivalence between the two causal-softmax formulations, then verify analyticity of the embedding layer and of each sub-layer and their compositions.

**Proposition B.1** (Equivalence of masked and projection causal softmax). *For any logits $\mathbf{Z} \in \mathbb{R}^{T \times T}$, let $\mathbf{M}$ and $\mathbf{L}$ be as in Definitions B.6–B.7. Then, row-wise,*

$$\mathrm{softmax}(\mathbf{Z} + \mathbf{M}) = \mathrm{RN}\big( \mathbf{L} \odot \exp \mathbf{Z} \big).$$

*Consequently, the two definitions of the Causal Self-Attention are identical.*

*Proof.* Fix a row $i$. By the mask:

$$\big[\mathrm{softmax}(\mathbf{Z} + \mathbf{M})\big]_{ij} = \begin{cases} \dfrac{e^{\mathbf{Z}_{ij}}}{\sum_{k \leq i} e^{\mathbf{Z}_{ik}}}, & j \leq i, \\ 0, & j > i, \end{cases}$$

interpreting $-\infty$ via a limit. On the other hand, it holds that:

$$[\mathbf{L} \odot \exp \mathbf{Z}]_{ij} = \mathbb{I}_{j \leq i}\, e^{\mathbf{Z}_{ij}}.$$

Therefore, $\mathbf{L} \odot \exp \mathbf{Z}$ keeps exactly the entries with $j \leq i$. Then, for each row, row normalization divides the kept entries by the same positive sum $\sum_{k \leq i} e^{\mathbf{Z}_{ik}}$ and leaves the others at 0, yielding the same row as above. This holds for every row $i$, proving the identity. $\square$

**Proposition B.2** (Embedding layer is real-analytic in the parameters). *Fix a sequence* $s = \langle s_1, \ldots, s_T \rangle \in \mathcal{V}^{\leq K}$ *with* $T = |s|$. *Consider the map*

$$(\mathbf{E}, \mathbf{P}) \longmapsto \mathrm{Emb}(s) = \mathrm{E}(s) + \mathrm{PE}(s) \in \mathbb{R}^{T \times d}, \qquad \mathbf{E} \in \mathbb{R}^{|\mathcal{V}| \times d}, \ \mathbf{P} \in \mathbb{R}^{K \times d}.$$

*Then this map is real-analytic on* $\mathbb{R}^{|\mathcal{V}| \times d} \times \mathbb{R}^{K \times d}$ *(in the sense of Definition A.2).*

*Proof.* Let $S_s \in \{0,1\}^{T \times |\mathcal{V}|}$ select rows $\{s_i\}_{i=1}^{T}$, and $R_T \in \{0,1\}^{T \times K}$ select the first $T$ rows. Then

$$\mathrm{E}(s) = S_s \mathbf{E}, \qquad \mathrm{PE}(s) = R_T \mathbf{P}, \qquad \mathrm{Emb}(s) = S_s \mathbf{E} + R_T \mathbf{P}.$$

Each map $(\mathbf{E}, \mathbf{P}) \mapsto S_s \mathbf{E}$ and $(\mathbf{E}, \mathbf{P}) \mapsto R_T \mathbf{P}$ is a matrix product of a *constant* matrix with the variable (*constant maps are real-analytic* as degree-0 polynomials by Proposition A.9; the product is real-analytic by Proposition A.10). Their sum is real-analytic by closure under addition (Proposition A.1). Hence $(\mathbf{E}, \mathbf{P}) \mapsto \mathrm{Emb}(s)$ is real-analytic. $\square$

**Proposition B.3** (Joint real-analyticity of core modules and stacks). *Assume the pointwise activation* $\sigma : \mathbb{R} \to \mathbb{R}$ *used in the MLP is real-analytic (e.g.,* $\tanh$, GELU). *Fix* $T \in [K]$. *For notational convenience define the parameter tuples*

$$\Theta_{\mathrm{attn}} := \left( \{\mathbf{Q}^{(h)}, \mathbf{K}^{(h)}, \mathbf{V}^{(h)}\}_{h=1}^{H}, \mathbf{W}^O \right), \quad \Theta_{\mathrm{LN}}^{(1)} := (\boldsymbol{\gamma}^{(1)}, \boldsymbol{\beta}^{(1)}), \quad \Theta_{\mathrm{LN}}^{(2)} := (\boldsymbol{\gamma}^{(2)}, \boldsymbol{\beta}^{(2)}),$$

$$\Theta_{\mathrm{mlp}} := \left( \{\mathbf{W}^{(m)}, \mathbf{b}^{(m)}\}_{m=1}^{M} \right), \qquad \Theta_{\mathrm{TB}} := \left( \Theta_{\mathrm{attn}}, \Theta_{\mathrm{LN}}^{(1)}, \Theta_{\mathrm{LN}}^{(2)}, \Theta_{\mathrm{mlp}} \right), \quad \Theta_{\mathrm{Tr}, T} := \left( \Theta_{\mathrm{TB}}^{(1)}, \ldots, \Theta_{\mathrm{TB}}^{(L)} \right).$$

*Then the following maps are jointly real-analytic in their inputs and parameters:*

1. ***MLP.*** $(\mathbf{x}, \Theta_{\mathrm{mlp}}) \mapsto \mathrm{mlp}_M(\mathbf{x})$ *is real-analytic: each affine layer* $(\mathbf{W}, \mathbf{b}, \mathbf{x}) \mapsto \mathbf{W}\mathbf{x} + \mathbf{b}$ *is a matrix product plus addition (Proposition A.10 and Proposition A.1); the activation* $\sigma$ *is real-analytic by assumption, and composition preserves real-analyticity (Proposition A.2). Iteration over* $M$ *layers is repeated composition (Proposition A.2).*

2. ***Layer Normalization.*** $(\mathbf{x}, \boldsymbol{\gamma}, \boldsymbol{\beta}) \mapsto \mathrm{LN}(\mathbf{x}) = \boldsymbol{\gamma} \odot \frac{\mathbf{x} - \mu_{\mathbf{x}}}{\sqrt{\sigma_{\mathbf{x}}^2 + \varepsilon}} + \boldsymbol{\beta}$ *is real-analytic:* $\mu_{\mathbf{x}}$ *and* $\sigma_{\mathbf{x}}^2$ *are (entrywise) polynomials in* $\mathbf{x}$ *(Proposition A.9);* $g(\mathbf{x}) = \sigma_{\mathbf{x}}^2 + \varepsilon$ *satisfies* $g(\mathbf{x}) > 0$ *(definition of* $\varepsilon > 0$), *and the scalar map* $h(t) = t^{-1/2}$ *is real-analytic on* $(0, \infty)$ *(classical binomial series). Thus* $h \circ g$ *is real-analytic (Proposition A.2); division by* $g^{1/2}$ *is a quotient by a nonvanishing real-analytic function (Proposition A.1); Hadamard scaling by* $\boldsymbol{\gamma}$ *and addition of* $\boldsymbol{\beta}$ *preserve real-analyticity (Proposition A.11 and Proposition A.1). Row-wise application is handled by stacking (Proposition A.12) and the vectorization equivalence (Lemma A.1).*

3. ***Unembedding.*** $(\mathbf{h}, \mathbf{U}, \boldsymbol{\gamma}, \boldsymbol{\beta}) \mapsto \mathrm{softmax}\left( \mathbf{U} \, \mathrm{LN}(\mathbf{h}) \right)$ *is real-analytic:* LN *is real-analytic by (2); multiplication by* $\mathbf{U}$ *is real-analytic (Proposition A.10);* softmax *is real-analytic (Proposition A.7); the overall map is a composition (Proposition A.2) and stacking across coordinates (Proposition A.12).*

4. ***Self-Attention (vanilla or causal) and Multi-Head.*** *Let* $\mathbf{Z} = \frac{1}{\sqrt{d_\eta}} (\mathbf{XQ})(\mathbf{XK})^\top$.

   (a) Vanilla SA: $(\mathbf{X}, \mathbf{Q}, \mathbf{K}, \mathbf{V}) \mapsto \mathrm{softmax}(\mathbf{Z})\mathbf{XV}$ *is real-analytic by: matrix products (Proposition A.10), scaling, row-wise softmax (Proposition A.7 with stacking, Proposition A.12, and Lemma A.1), and a final matrix product.*

   (b) Causal SA (projection form): *With* $\mathbf{L}$ *unit lower-triangular and using Definition B.7,*
   $$(\mathbf{X}, \mathbf{Q}, \mathbf{K}, \mathbf{V}) \longmapsto \mathrm{RN}\left( \mathbf{L} \odot \exp \mathbf{Z} \right) \mathbf{XV}$$
   *is real-analytic:* $\exp$ *is real-analytic (Proposition A.5); Hadamard scaling by fixed* $\mathbf{L}$ *is real-analytic (Proposition A.11); by Remark 11, every row of* $\mathbf{L} \odot \exp(\mathbf{Z})$ *sums to a strictly positive value (the diagonal term), so the argument lies in the domain* $\mathcal{D}_T$ *of Proposition A.8; hence* RN *is real-analytic there; the final multiplication by* $\mathbf{XV}$ *is real-analytic (Proposition A.10).*

   *Therefore, each* single *attention head is real-analytic whether it is vanilla or causal (projection). For Multi-Head Self-Attention (Definition B.8), horizontal concatenation across heads is real-analytic (Proposition A.12), and the output projection by* $\mathbf{W}^O$ *is a matrix product (Proposition A.10). Hence* $(\mathbf{X}, \Theta_{\mathrm{attn}}) \mapsto \mathrm{attn}_H(\mathbf{X})$ *is real-analytic regardless of which attention variant each head uses.*

5. **Transformer Block (fixed $T$).** $(\mathbf{X}, \Theta_{\mathrm{TB}}) \mapsto \mathrm{TB}(\mathbf{X}) \in \mathbb{R}^{T \times d}$ *is real-analytic: apply LN row-wise to get $\overline{\mathbf{X}}$ (item 2 with stacking,* Proposition A.12, *and* Lemma A.1*); apply attention (item 4) to $\overline{\mathbf{X}}$; add the residual (closure under addition,* Proposition A.1*); apply LN row-wise to get $\overline{\mathbf{H}}$ (item 2 with stacking and* Lemma A.1*); apply the row-wise MLP (item 1 with stacking,* Proposition A.12*); add the residual again (*Proposition A.1*). All intermediate matrix multiplications use* Proposition A.10*, and the overall structure is a composition (*Proposition A.3 *via* Lemma A.1*).*

6. **Transformer (fixed $T$).** $(\mathbf{X}, \Theta_{\mathrm{Tr},T}) \mapsto \mathrm{Tr}_T(\mathbf{X}) = \mathrm{TB}^{(L)} \circ \cdots \circ \mathrm{TB}^{(1)}(\mathbf{X})$ *is a composition of real-analytic maps from (5), hence real-analytic by* Proposition A.3*.*

*All statements extend from vector-valued to matrix-valued, row-wise applications via* Proposition A.12 *and* Lemma A.1*, and every sum/product/quotient/composition step above invokes* Proposition A.1*,* Proposition A.10*, and* Proposition A.3 *as indicated.*

## C  ALMOST SURE INJECTIVITY

This section establishes a foundational structural result: for causal Transformer Language Models with standard architectural widths and at least one attention head per block, the final hidden state at the last token is almost surely injective with respect to the input sequence, assuming the model parameters are drawn from any absolutely continuous distribution at initialization. Crucially, we show this injectivity is preserved after any finite number of gradient descent (GD) updates.

We organize the section in two parts; **(i)** Measure-zero collisions via real-analyticity and a witness construction and **(ii)** Preservation of absolute continuity under gradient descent. Each piece builds toward the main theorem, which asserts that under mild width and head assumptions, the Transformer map from input sequences to last-token representations is injective almost surely, even after multiple rounds of training. The main theorem follows.

**Assumption C.1** (Minimum Embedding Dimension). *We assume the embedding dimension satisfies $d \geq 4$ and $d_\eta \geq 1$. Furthermore, we assume that each transformer block has at least one attention head. These conditions are trivially satisfied in practice: for modern large language models, embedding dimensions are typically in the hundreds or thousands, and each layer has multiple attention heads, so the assumptions impose no practical restrictions on the models under consideration.*

**Theorem C.1** (Finite-horizon a.s. injectivity under GD). *Fix a finite vocabulary $\mathcal{V}$, a context bound $K \in \mathbb{N}$, a time horizon $T \in \mathbb{N}$, and consider the causal Transformer Language Model (TLM) of Definition B.13 under Assumption C.1. Let $\left\{ \left( \mathrm{s}_t \in \mathcal{V}^{\leq K}, \mathbf{p}_t \in \Delta^{|\mathcal{V}|-1} \right) \right\}_{t=1}^{T}$ be any sequence of samples and let $\{\eta_t \in (0,1)\}_{t=1}^{T}$ be any sequence of step-sizes. Assume the parameters are randomly initialized and updated by gradient descent:*

$$\boldsymbol{\theta}_0 \sim \mu, \qquad \mu \ll \mathrm{Leb}_p,$$
$$\boldsymbol{\theta}_{t+1} = \boldsymbol{\theta}_t - \eta_t \nabla \mathcal{L}_{\mathrm{s}_t, \mathbf{p}_t}(\boldsymbol{\theta}_t),$$

*where $\mathrm{Leb}_p$ denotes Lebesgue measure on $\mathbb{R}^p$ and $\mathcal{L}_{\mathrm{s}, \mathbf{p}} : \mathbb{R}^p \to \mathbb{R}$ is the standard cross-entropy loss*

$$\mathcal{L}_{\mathrm{s}, \mathbf{p}}(\boldsymbol{\theta}) = \mathrm{CrossEntropy}\big( f(\mathrm{s} \,;\, \boldsymbol{\theta}), \, \mathbf{p} \big).$$

*Then, with probability one over the draw of $\boldsymbol{\theta}_0$, the last-token, last-layer representation map*

$$\mathcal{V}^{\leq K} \ni \mathrm{s} \longmapsto \mathbf{r}(\mathrm{s} \,;\, \boldsymbol{\theta}_T) \in \mathbb{R}^d$$

*is injective. Equivalently,*

$$\Pr \big[ \exists \, \mathrm{s} \neq \mathrm{t} \in \mathcal{V}^{\leq K} : \mathbf{r}(\mathrm{s} \,;\, \boldsymbol{\theta}_T) = \mathbf{r}(\mathrm{t} \,;\, \boldsymbol{\theta}_T) \big] = 0,$$

*where $\mathbf{r}(\cdot \,;\, \boldsymbol{\theta}_T)$ denotes the last-token representation defined in Equation (29).*

*Proof.*

Let $\boldsymbol{\theta}_0 \sim \mu$ with $\mu \ll \mathrm{Leb}_p$. For a fixed training horizon $T$, define the *GD update map*

$$\Phi : \mathbb{R}^p \to \mathbb{R}^p, \qquad \Phi(\boldsymbol{\theta}_0) = \boldsymbol{\theta}_T,$$

i.e. $\Phi$ is the composition of $T$ gradient-descent steps with step sizes $\{\eta_t\}_{t=1}^{T} \subset (0,1)$ on the loss $\mathcal{L}$.

**1) Absolute continuity after $T$ steps.** By Corollary C.5.1, since $\mu \ll \mathrm{Leb}_p$, the pushforward law $\Phi_\# \mu$ of $\boldsymbol{\theta}_T$ remains absolutely continuous:

$$\boldsymbol{\theta}_T \sim \Phi_\# \mu \ll \mathrm{Leb}_p.$$

**2) Global almost-sure distinctness.** Let $\mathcal{S} := \mathcal{V}^{\leq K}$, which is finite. By Corollary C.2.1, under any absolutely continuous parameter law,

$$\Pr \Big[ \mathbf{r}(\mathrm{s} \,;\, \boldsymbol{\theta}_T) \neq \mathbf{r}(\mathrm{t} \,;\, \boldsymbol{\theta}_T) \quad \forall \, \mathrm{s} \neq \mathrm{t} \in \mathcal{V}^{\leq K} \Big] = 1.$$

Thus the map $\mathrm{s} \mapsto \mathbf{r}(\mathrm{s} \,;\, \boldsymbol{\theta}_T)$ is injective almost surely, as claimed. $\qquad \square$

## C.1 ABSOLUTE CONTINUITY ENSURES ALMOST SURE INJECTIVITY

We begin by fixing two distinct sequences and asking when their last-token representations can coincide. As before, in this subsection we will consider a finite vocabulary $\mathcal{V}$ and a finite context window $K \in \mathbb{N}$. Additionally, recall that for $\boldsymbol{\theta} = (\boldsymbol{\theta}_1, \boldsymbol{\theta}_2, \boldsymbol{\theta}_3) \in \mathbb{R}^p$:

$$\mathbf{r}(\mathrm{u}\,;\,\boldsymbol{\theta}) := \left( \mathrm{Tr}_{|u|}\big(\mathrm{Emb}(\mathrm{u}\,;\,\boldsymbol{\theta}_1)\,;\,\boldsymbol{\theta}_2\big) \right)_{|u|} \in \mathbb{R}^d,$$

and for $\mathrm{s} \neq \mathrm{t}$, we define the discrepancy:

$$h(\boldsymbol{\theta}) := \big\| \mathbf{r}(\mathrm{s}\,;\,\boldsymbol{\theta}) - \mathbf{r}(\mathrm{t}\,;\,\boldsymbol{\theta}) \big\|_2^2.$$

By Proposition B.3, this map is real-analytic. To invoke the zero-set theorem, it suffices to show that $h \not\equiv 0$. We construct a parameter configuration $\boldsymbol{\theta}_\star$ such that $\mathbf{r}(\mathrm{s}\,;\,\boldsymbol{\theta}_\star) \neq \mathbf{r}(\mathrm{t}\,;\,\boldsymbol{\theta}_\star)$, treating two exhaustive cases:

- **Case A:** If the sequences differ at their final token or in length, we isolate this distinction via selective initialization of embeddings and positional encodings.
- **Case B:** If they differ earlier, we construct orthogonal embeddings and exploit attention heads to differentiate the contributions to the final representation.

In both cases, we demonstrate explicit parameter settings under which the discrepancy is nonzero. This confirms $h \not\equiv 0$, and the zero set $\{ \boldsymbol{\theta} : \mathbf{r}(\mathrm{s}\,;\,\boldsymbol{\theta}) = \mathbf{r}(\mathrm{t}\,;\,\boldsymbol{\theta}) \}$ has measure zero by Theorem A.1. Hence, if the parameter distribution is absolutely continuous, the probability of a collision is zero. A union bound extends this to any finite set of inputs.

**Theorem C.2** (Almost-sure pairwise distinctness of last-token representations)**.** *Let the parameter vector* $\boldsymbol{\theta} \in \mathbb{R}^p$ *be drawn from any distribution absolutely continuous with respect to Lebesgue measure. Then, for any fixed* $\mathrm{s} \neq \mathrm{t}$,

$$\Pr\left[\, \mathbf{r}(\mathrm{s}\,;\,\boldsymbol{\theta}) = \mathbf{r}(\mathrm{t}\,;\,\boldsymbol{\theta}) \,\right] = 0.$$

*Proof.* Let $T_\mathrm{s} = |\mathrm{s}|$ and $T_\mathrm{t} = |\mathrm{t}|$, and $h(\boldsymbol{\theta}) := \big\| \mathbf{r}(\mathrm{s}\,;\,\boldsymbol{\theta}) - \mathbf{r}(\mathrm{t}\,;\,\boldsymbol{\theta}) \big\|_2^2$. Since $h$ is real-analytic (Proposition B.3), it suffices to show that it is not the zero function on $\mathbb{R}^p$; then $h^{-1}(\{0\})$ has Lebesgue measure zero by Theorem A.1, and absolute continuity transfers this to probability zero.

We construct a parameter setting $\boldsymbol{\theta}_\star$ for which $h(\boldsymbol{\theta}_\star) > 0$, treating two exhaustive cases:

**Case A:** $T_\mathrm{s} \neq T_\mathrm{t}$ **or** $\mathrm{s}_{T_\mathrm{s}} \neq \mathrm{t}_{T_\mathrm{t}}$**.** Set all Transformer parameters to zero so that the network acts as the identity: $\mathrm{Tr}_T(\mathbf{X}) = \mathbf{X}$.

- If $\mathrm{s}_{T_\mathrm{s}} \neq \mathrm{t}_{T_\mathrm{t}}$, set $\mathbf{E}_{\mathrm{s}_{T_\mathrm{s}}} = \mathbf{e}_1$, $\mathbf{E}_{\mathrm{t}_{T_\mathrm{t}}} = \mathbf{e}_2 \neq \mathbf{e}_1$, and all other rows of $\mathbf{E}$ to zero. Set $\mathbf{P} = \mathbf{0}_{K \times d}$. Then $\mathbf{r}(\mathrm{s}\,;\,\boldsymbol{\theta}_\star) = \mathbf{e}_1$, $\mathbf{r}(\mathrm{t}\,;\,\boldsymbol{\theta}_\star) = \mathbf{e}_2$, so $h(\boldsymbol{\theta}_\star) = \| \mathbf{e}_1 - \mathbf{e}_2 \|_2^2 > 0$.

- If $T_\mathrm{s} \neq T_\mathrm{t}$, set $\mathbf{E} = \mathbf{0}_{|\mathcal{V}| \times d}$ and $\mathbf{P}_{T_\mathrm{s}} = \mathbf{e}_1$, $\mathbf{P}_{T_\mathrm{t}} = \mathbf{e}_2 \neq \mathbf{e}_1$ (all others zero). Then, again, $\mathbf{r}(\mathrm{s}\,;\,\boldsymbol{\theta}_\star) = \mathbf{e}_1$, $\mathbf{r}(\mathrm{t}\,;\,\boldsymbol{\theta}_\star) = \mathbf{e}_2$, so $h(\boldsymbol{\theta}_\star) > 0$.

**Case B:** $T := T_\mathrm{s} = T_\mathrm{t}$ **and** $\mathrm{s}_T = \mathrm{t}_T$**, but** $\mathrm{s}_i \neq \mathrm{t}_i$ **for some** $i \in [T-1]$**.** Let $i^\star$ be the smallest such index. Note $T \geq 2$.

We construct a model with (i) all blocks after the first set to identity (zero parameters), (ii) in the first block, all heads set to zero except head 1 and the MLP is zero.

We explicitly construct embeddings and head-1 parameters $(\mathbf{Q}, \mathbf{K}, \mathbf{V})$, as well as the output projection $\mathbf{W}^O$, so that $\mathbf{r}(\mathrm{s}\,;\,\boldsymbol{\theta}_\star) \neq \mathbf{r}(\mathrm{t}\,;\,\boldsymbol{\theta}_\star)$.

**1) Embedding Construction.** Choose orthogonal vectors $\mathbf{e}, \mathbf{p}, \mathbf{q} \in \mathbb{R}^d$ satisfying:

$$\langle \mathbf{e}, \mathbf{p} \rangle = \langle \mathbf{e}, \mathbf{q} \rangle = \langle \mathbf{p}, \mathbf{q} \rangle = 0, \quad \langle \mathbf{1}_d, \mathbf{e} \rangle = \langle \mathbf{1}_d, \mathbf{p} \rangle = \langle \mathbf{1}_d, \mathbf{q} \rangle = 0, \quad \| \mathbf{e} \|_2 = \| \mathbf{p} \|_2 = \| \mathbf{q} \|_2 = 1.$$

Such vectors exist due to Assumption C.1 (requires $d \geq 4$). Set embeddings:

$$\mathbf{E}_v = \begin{cases} \mathbf{e}, & v \in \{\mathrm{s}_{i^\star}, \mathrm{s}_T\} \\ \mathbf{0}_d, & \text{otherwise} \end{cases}, \qquad \mathbf{P}_j = \begin{cases} \mathbf{p}, & j = i^\star \\ \mathbf{q}, & j = T \\ \mathbf{0}_d, & \text{otherwise} \end{cases}.$$

Thus, the input rows before LayerNorm are:

$$
\left[\mathbf{H}(s\,;\boldsymbol{\theta}_\star)\right]_j = \begin{cases} \mathbf{e} + \mathbf{p}, & j = i^\star \\ \mathbf{e} + \mathbf{q}, & j = T \\ \in \{\mathbf{e}, \mathbf{0}_d\}, & \text{otherwise} \end{cases}, \qquad \left[\mathbf{H}(t\,;\boldsymbol{\theta}_\star)\right]_j = \begin{cases} \mathbf{p}, & j = i^\star \\ \mathbf{e} + \mathbf{q}, & j = T \\ \in \{\mathbf{e}, \mathbf{0}_d\}, & \text{otherwise} \end{cases}.
$$

**2) LayerNorm Output.** Use LayerNorm with $(\boldsymbol{\gamma}, \boldsymbol{\beta}) = (\mathbf{1}, \mathbf{0})$. Since all components have zero mean, the normalization is:

$$
\mathrm{LN}(\mathbf{x}) = \frac{\mathbf{x}}{\sqrt{\frac{1}{d}\|\mathbf{x}\|^2 + \varepsilon}} =: c(\mathbf{x})\mathbf{x}.
$$

Define:

$$
c_{ep} := \left(\tfrac{2}{d} + \varepsilon\right)^{-1/2}, \qquad c_e := \left(\tfrac{1}{d} + \varepsilon\right)^{-1/2}.
$$

Then:

$$
\left[\overline{\mathbf{H}}(s\,;\boldsymbol{\theta}_\star)\right]_j = \begin{cases} c_{ep}(\mathbf{e} + \mathbf{p}), & j = i^\star \\ c_{ep}(\mathbf{e} + \mathbf{q}), & j = T \\ \in \{\mathbf{0}_d, c_e\mathbf{e}\}, & \text{otherwise} \end{cases}, \qquad \left[\overline{\mathbf{H}}(t\,;\boldsymbol{\theta}_\star)\right]_j = \begin{cases} c_{ep}\mathbf{p}, & j = i^\star \\ c_{ep}(\mathbf{e} + \mathbf{q}), & j = T \\ \in \{\mathbf{0}_d, c_e\mathbf{e}\}, & \text{otherwise} \end{cases}.
$$

**3) Head Parameters.** Let $\mathbf{e}_1 \in \mathbb{R}^{d_\eta}$ be the first standard basis vector. Set:

$$
\mathbf{Q} = \alpha \mathbf{e}\mathbf{e}_1^\top, \qquad \mathbf{K} = \beta \mathbf{p}\mathbf{e}_1^\top, \qquad \mathbf{V} = \mathbf{e}\mathbf{e}_1^\top,
$$

where $\alpha, \beta > 0$ are scalars to be chosen.

Then for any $j$, attention vectors are:

$$
\mathbf{q}_j = \alpha \left\langle \left[\overline{\mathbf{H}}(\cdot\,;\boldsymbol{\theta}_\star)\right]_j, \mathbf{e} \right\rangle \mathbf{e}_1, \quad \mathbf{k}_j = \beta \left\langle \left[\overline{\mathbf{H}}(\cdot\,;\boldsymbol{\theta}_\star)\right]_j, \mathbf{p} \right\rangle \mathbf{e}_1, \quad \mathbf{v}_j = \left\langle \left[\overline{\mathbf{H}}(\cdot\,;\boldsymbol{\theta}_\star)\right]_j, \mathbf{e} \right\rangle \mathbf{e}_1.
$$

At row $T$, $\mathbf{q}_T^{(s)} = \mathbf{q}_T^{(t)} = \alpha c_{ep}\mathbf{e}_1$. Only the key at $i^\star$ is nonzero:

$$
\mathbf{k}_{i^\star}^{(s)} = \beta c_{ep}\mathbf{e}_1, \quad \mathbf{k}_{i^\star}^{(t)} = \beta c_e\mathbf{e}_1.
$$

Value vectors at $i^\star$ differ:

$$
\mathbf{v}_{i^\star}^{(s)} = c_{ep}\mathbf{e}_1, \quad \mathbf{v}_{i^\star}^{(t)} = \mathbf{0}_d.
$$

And $\mathbf{v}_T^{(s)} = \mathbf{v}_T^{(t)} = c_{ep}\mathbf{e}_1$.

**4) Attention Weights.** The only nonzero score is at $i^\star$:

$$
\mathbf{S}_{T,i^\star}^{(s)} = \frac{\alpha\beta}{\sqrt{d_\eta}} c_{ep}^2, \quad \mathbf{S}_{T,i^\star}^{(t)} = \frac{\alpha\beta}{\sqrt{d_\eta}} c_{ep}c_e, \quad \mathbf{S}_{T,j}^{(\cdot)} = 0 \text{ for } j \neq i^\star.
$$

Fix $\delta \in (0, \tfrac{1}{2})$ and define $L := \log\left(\frac{1-\delta}{\delta}(T-1)\right)$. Set $\alpha\beta = \sqrt{d_\eta}L/c_{ep}^2$, so $\mathbf{S}_{T,i^\star}^{(s)} = L$ and $\mathbf{S}_{T,i^\star}^{(t)} > L$. Then:

$$
\mathbf{A}_{T,i^\star}^{(s)} \geq 1 - \delta, \quad \mathbf{A}_{T,i^\star}^{(t)} > 1 - \delta, \quad \mathbf{A}_{T,j}^{(\cdot)} \leq \frac{\delta}{T-1} \text{ for } j \neq i^\star.
$$

**5) Self-Attention Output.**

$$
\mathbf{y}_T^{(s)} = (1-\delta)c_{ep}\mathbf{e}_1 + \sum_{j \neq i^\star} \mathbf{A}_{T,j}^{(s)}\mathbf{v}_j^{(s)}, \quad \mathbf{y}_T^{(t)} = \sum_{j \neq i^\star} \mathbf{A}_{T,j}^{(t)}\mathbf{v}_j^{(t)}.
$$

Tails are bounded by:

$$
\left\| \sum_{j \neq i^\star} \mathbf{A}_{T,j}^{(\cdot)}\mathbf{v}_j^{(\cdot)} \right\|_2 \leq \delta c_e.
$$

Since both outputs lie in $\mathrm{span}\{\mathbf{e}_1\}$, we compare:

$$\langle \mathbf{y}_T^{(\mathrm{s})} - \mathbf{y}_T^{(\mathrm{t})}, \mathbf{e}_1 \rangle \geq (1-\delta)c_{ep} - 2\delta c_e.$$

Choosing $\delta < \frac{c_{ep}}{c_{ep}+2c_e}$ makes this strictly positive.

**6) Output Projection and Propagation.** Let $\mathbf{W}^O$ be the matrix with $(\mathbf{W}^O)_{1,1} = 1$ and all other entries zero. Then the head output is projected into coordinate 1, making the last row of the first transformer block differ between s and t in the first coordinate. Since the original rows at $T$ were identical and the rest of the network is identity, this difference propagates to the final output, and we get $\mathbf{r}(\mathrm{s}\,;\boldsymbol{\theta}_\star) \neq \mathbf{r}(\mathrm{t}\,;\boldsymbol{\theta}_\star)$.

$\square$

**Remark 12** (Causal Self-Attention). *The same construction works for causal self-attention. In our setup, attention at position $T$ only needs to consider tokens at positions $j \leq T$, and we only rely on attention from $T$ to $i^\star < T$. All nonzero scores occur at these allowable indices, so causal masking does not affect the computation or the argument.*

**Corollary C.2.1** (Almost-sure global distinctness over a finite input family). *Let $\mathcal{S} \subseteq \mathcal{V}^{\leq K}$ be any finite collection of inputs. If $\boldsymbol{\theta}$ is drawn from a law absolutely continuous w.r.t. $\mathrm{Leb}_p$, then*

$$\Pr\big[\, \mathbf{r}(\mathrm{s}\,;\boldsymbol{\theta}) \neq \mathbf{r}(\mathrm{t}\,;\boldsymbol{\theta}) \ \text{for all distinct}\ \mathrm{s}, \mathrm{t} \in \mathcal{S} \,\big] \ = \ 1.$$

*In particular, the last-token representations are pairwise distinct almost surely across all inputs.*

*Proof.* For each unordered pair $\{\mathrm{s}, \mathrm{t}\} \subset \mathcal{S}$ with $\mathrm{s} \neq \mathrm{t}$, Theorem C.2 gives $\Pr[\,\mathbf{r}(\mathrm{s}\,;\boldsymbol{\theta}) = \mathbf{r}(\mathrm{t}\,;\boldsymbol{\theta})\,] = 0$. By the union bound over the finitely many pairs ($\binom{|\mathcal{S}|}{2}$ in total),

$$\Pr\Big[\exists\, \mathrm{s} \neq \mathrm{t} \in \mathcal{S} : \mathbf{r}(\mathrm{s}\,;\boldsymbol{\theta}) = \mathbf{r}(\mathrm{t}\,;\boldsymbol{\theta})\Big] \leq \sum_{\mathrm{s},\mathrm{t}} \Pr\big[\,\mathbf{r}(\mathrm{s}\,;\boldsymbol{\theta}) = \mathbf{r}(\mathrm{t}\,;\boldsymbol{\theta})\,\big] = 0.$$

Hence the complement event has probability 1. $\square$

**Remark 13** (Pointwise vs. last-token injectivity). *Sutter et al. (2025) establish a related but distinct guarantee. They analyze the mapping from a prompt to the* entire *sequence (matrix) of hidden states, which already rules out collisions for inputs of different lengths. Their result is* pointwise injectivity*: if two prompts differ at position $t$, then the $t$-th hidden state (row) differs. This does not, by itself, imply injectivity of the map to the final hidden state / last-token embedding that we study, so two different prompts could still coincide at the last token–our quantity of operational interest.*

## C.2 ABSOLUTE CONTINUITY OF THE PARAMETER DISTRIBUTION IS PRESERVED UNDER GD

Our goal in this subsection is to explain why absolute continuity of the parameter law at initialization survives any finite number of gradient–descent (GD) steps, thereby allowing the almost-sure injectivity argument from the previous subsection to persist throughout training. The argument proceeds in four steps.

**Step 1: Regularity of the GD map.** By Propositions A.6 and B.3, the loss $\mathcal{L}_{\mathrm{s},\mathbf{p}}$ is real-analytic, and real-analyticity is closed under differentiation and composition. Consequently the GD map $\phi(\boldsymbol{\theta}) = \boldsymbol{\theta} - \eta\nabla\mathcal{L}_{\mathrm{s},\mathbf{p}}(\boldsymbol{\theta})$ is real-analytic, its Jacobian $D\phi(\boldsymbol{\theta}) = \mathbf{I}_p - \eta\nabla^2\mathcal{L}_{\mathrm{s},\mathbf{p}}(\boldsymbol{\theta})$ is real-analytic, and so is $\boldsymbol{\theta} \mapsto \det D\phi(\boldsymbol{\theta})$ (the determinant is a polynomial in the matrix entries).

**Step 2: Witness and measure-zero critical set.** We rule out the degenerate case by a witness: at $\boldsymbol{\theta}_\star = \mathbf{0}_p$, our Hessian calculation (Lemma C.4) shows $\det D\phi(\boldsymbol{\theta}_\star) > 0$, hence $\det D\phi$ is not identically zero and its zero set $\mathcal{C} := \{\det D\phi = 0\}$ has Lebesgue measure zero by the real-analytic zero–set theorem (Theorem A.1; summarized in Theorem C.3).

**Step 3: Local-to-global via countable chart covers.** On the complement $\mathbb{R}^p \setminus \mathcal{C}$, the Inverse Function Theorem (Theorem A.2) provides, for every $\boldsymbol{\theta}$, a neighborhood on which $\phi$ is a $C^1$ diffeomorphism. Although these neighborhoods form an a priori uncountable cover, the second countability of $\mathbb{R}^p$ (and of its subspaces) ensures a *countable* subcover of such charts (Proposition A.15, Lemma C.5). This countability is crucial because it lets us pass from local statements to a global measure statement via countable unions. With this cover in hand, the change-of-variables formula on each chart (Theorem C.4) implies that the image under the local inverse of any null set remains null; piecing the charts together and adding the null set $\mathcal{C}$ shows that preimages of Lebesgue-null sets under $\phi$ are null (Lemma C.6).

**Step 4: Preservation of absolute continuity and conclusion.** Equivalently, $\phi$ pushes absolutely continuous laws to absolutely continuous laws (Theorem C.5); iterating across finitely many GD steps preserves absolute continuity (Corollary C.5.1). Finally, combining this preservation with the almost-sure pairwise distinctness of last-token representations over any finite input family (Corollary C.2.1) yields the main consequence we need for training: the last-token representation map remains injective almost surely after any finite GD horizon.

### C.2.1 WITNESS CONSTRUCTION

The goal of this subsubsection is to show that the GD Jacobian determinant is not identically zero by evaluating it at the all-zeros witness $\boldsymbol{\theta}_\star = \mathbf{0}_p$. The argument proceeds bottom-up: we first establish a "zero-gate" lemma that zeroes out most Hessian blocks when the gate matrix vanishes (Lemma C.1), derive the resulting spectrum (Lemma C.2, Lemma C.3), and then assemble these into the full Hessian at the witness (Lemma C.4).

**Lemma C.1** (Zero-gate through scalar loss). *Let $\mathcal{U} \subseteq \mathbb{R}^{m+q}$ be open and write points as $\mathbf{v} = (\boldsymbol{\xi}, \boldsymbol{\psi})$ with $\boldsymbol{\xi} \in \mathbb{R}^m$ and $\boldsymbol{\psi} \in \mathbb{R}^q$. Let $\pi : \mathbb{R}^{m+q} \to \mathbb{R}^m$ be the projection $\pi(\boldsymbol{\xi}, \boldsymbol{\psi}) = \boldsymbol{\xi}$. Consider*

$$g \in C^2(\mathbb{R}^m \, ; \, \mathbb{R}^{n \times r}), \qquad h \in C^2(\mathcal{U} \, ; \, \mathbb{R}^r),$$

*and define $f : \mathcal{U} \to \mathbb{R}^n$ by*

$$f(\boldsymbol{\xi}, \boldsymbol{\psi}) := g(\boldsymbol{\xi}) \, h(\boldsymbol{\xi}, \boldsymbol{\psi}) = g\big(\pi(\boldsymbol{\xi}, \boldsymbol{\psi})\big) \, h(\boldsymbol{\xi}, \boldsymbol{\psi}).$$

*Let $\mathcal{L} \in C^2(\mathbb{R}^n; \mathbb{R})$ and set*

$$R := \mathcal{L} \circ f : \mathcal{U} \to \mathbb{R}, \qquad R(\boldsymbol{\xi}, \boldsymbol{\psi}) = \mathcal{L}\big(g(\boldsymbol{\xi}) \, h(\boldsymbol{\xi}, \boldsymbol{\psi})\big).$$

*Fix $\mathbf{v}_0 = (\boldsymbol{\xi}_0, \boldsymbol{\psi}_0) \in \mathcal{U}$ and assume $g(\boldsymbol{\xi}_0) = \mathbf{0}_{n \times r}$. Then the Hessian of $R$ at $\mathbf{v}_0$ has block form*

$$\nabla^2 R(\mathbf{v}_0) = \begin{pmatrix} \nabla^2_{\boldsymbol{\xi}\boldsymbol{\xi}} R(\mathbf{v}_0) & \nabla^2_{\boldsymbol{\xi}\boldsymbol{\psi}} R(\mathbf{v}_0) \\ \nabla^2_{\boldsymbol{\psi}\boldsymbol{\xi}} R(\mathbf{v}_0) & \nabla^2_{\boldsymbol{\psi}\boldsymbol{\psi}} R(\mathbf{v}_0) \end{pmatrix} = \begin{pmatrix} \nabla^2_{\boldsymbol{\xi}\boldsymbol{\xi}} R(\mathbf{v}_0) & \mathbf{0}_{m \times q} \\ \mathbf{0}_{q \times m} & \mathbf{0}_{q \times q} \end{pmatrix}.$$

*i.e. all mixed and $\boldsymbol{\psi}$–only second partials vanish.*

*Proof.*

**1)** Introduce the bilinear multiplication map $\mu : \mathbb{R}^{n \times r} \times \mathbb{R}^r \to \mathbb{R}^n$, $\mu(\mathbf{M}, \mathbf{y}) = \mathbf{M}\mathbf{y}$, and the $C^2$ map $H : \mathcal{U} \to \mathbb{R}^{n \times r} \times \mathbb{R}^r$, $H(\boldsymbol{\xi}, \boldsymbol{\psi}) = (g(\boldsymbol{\xi}), h(\boldsymbol{\xi}, \boldsymbol{\psi}))$. Then $f = \mu \circ H$ and we write:

$$g_0 := g(\boldsymbol{\xi}_0) = \mathbf{0}_{n \times r} \qquad h_0 := h(\boldsymbol{\xi}_0, \boldsymbol{\psi}_0) \qquad H(\mathbf{v_0}) = (g_0, h_0).$$

Because $\mu$ is bilinear, $D\mu(\mathbf{M}, \mathbf{y})[(\Delta\mathbf{M}, \Delta\mathbf{y})] = \Delta\mathbf{M}\,\mathbf{y} + \mathbf{M}\,\Delta\mathbf{y}$. By the chain rule:

$$\begin{aligned} Df(\mathbf{v}_0)\big[(\mathbf{h}_{\boldsymbol{\xi}}, \mathbf{h}_{\boldsymbol{\psi}})\big] &= D\mu(g_0, h_0)\Big[ Dg(\boldsymbol{\xi}_0)[\mathbf{h}_{\boldsymbol{\xi}}], \; Dh(\mathbf{v}_0)[(\mathbf{h}_{\boldsymbol{\xi}}, \mathbf{h}_{\boldsymbol{\psi}})] \Big] \\ &= Dg(\boldsymbol{\xi}_0)[\mathbf{h}_{\boldsymbol{\xi}}]\, h_0 + \underbrace{g_0}_{\mathbf{0}_{n \times r}} Dh(\mathbf{v}_0)[(\mathbf{h}_{\boldsymbol{\xi}}, \mathbf{h}_{\boldsymbol{\psi}})] \\ &= Dg(\boldsymbol{\xi}_0)[\mathbf{h}_{\boldsymbol{\xi}}]\, h_0. \end{aligned}$$

In particular, $Df(\mathbf{v}_0)\big[(\mathbf{0}_m, \; \cdot \;)\big] = \mathbf{0}_n$. The second-order chain rule for Fréchet derivatives (e.g. Magnus & Neudecker 2019, Thm. 18.4) yields:

$$D^2 f(\mathbf{v}_0)[\mathbf{h}, \mathbf{k}] = D^2 \mu\big(H(\mathbf{v}_0)\big)\big[ DH(\mathbf{v}_0)[\mathbf{h}], \; DH(\mathbf{v}_0)[\mathbf{k}] \big] + D\mu\big(H(\mathbf{v}_0)\big)\big[ D^2 H(\mathbf{v}_0)[\mathbf{h}, \mathbf{k}] \big].$$

Because $\mu$ is bilinear, $D^2\mu \equiv \mathbf{0}$ and the first term is $0$. Furthermore,

$$D^2 H(\mathbf{v}_0)[\mathbf{h}, \mathbf{k}] = \Big( D^2 g(\boldsymbol{\xi}_0)[\mathbf{h}_{\boldsymbol{\xi}}, \mathbf{k}_{\boldsymbol{\xi}}], \ D^2 h(\mathbf{v}_0)\big[(\mathbf{h}_{\boldsymbol{\xi}}, \mathbf{h}_{\boldsymbol{\psi}}), (\mathbf{k}_{\boldsymbol{\xi}}, \mathbf{k}_{\boldsymbol{\psi}})\big]\Big),$$

and it holds that:

$$\begin{aligned}
D^2 f(\mathbf{v}_0)[\mathbf{h}, \mathbf{k}] &= D\mu(g_0, h_0)\Big[ D^2 g(\boldsymbol{\xi}_0)[\mathbf{h}_{\boldsymbol{\xi}}, \mathbf{k}_{\boldsymbol{\xi}}], \ D^2 h(\mathbf{v}_0)\big[(\mathbf{h}_{\boldsymbol{\xi}}, \mathbf{h}_{\boldsymbol{\psi}}), (\mathbf{k}_{\boldsymbol{\xi}}, \mathbf{k}_{\boldsymbol{\psi}})\big] \Big] \\
&= \Big( D^2 g(\boldsymbol{\xi}_0)[\mathbf{h}_{\boldsymbol{\xi}}, \mathbf{k}_{\boldsymbol{\xi}}] \Big) h_0 + \underbrace{g_0}_{\mathbf{0}_{n \times r}} \Big( D^2 h(\mathbf{v}_0)\big[(\mathbf{h}_{\boldsymbol{\xi}}, \mathbf{h}_{\boldsymbol{\psi}}), (\mathbf{k}_{\boldsymbol{\xi}}, \mathbf{k}_{\boldsymbol{\psi}})\big] \Big) \\
&= \Big( D^2 g(\boldsymbol{\xi}_0)[\mathbf{h}_{\boldsymbol{\xi}}, \mathbf{k}_{\boldsymbol{\xi}}] \Big) h_0 .
\end{aligned}$$

If at least one of the two directions has $\boldsymbol{\xi}$–component zero, then $D^2 g(\boldsymbol{\xi}_0)[\mathbf{h}_{\boldsymbol{\xi}}, \mathbf{k}_{\boldsymbol{\xi}}] = \mathbf{0}$, so the bilinear form vanishes.

**2)** Apply the second-order chain rule to $R = \mathcal{L} \circ f$ at $\mathbf{v}_0$:

$$D^2 R(\mathbf{v}_0)[\mathbf{h}, \mathbf{k}] = D^2 \mathcal{L}\big(f(\mathbf{v}_0)\big)\big[ Df(\mathbf{v}_0)[\mathbf{h}], \ Df(\mathbf{v}_0)[\mathbf{k}] \big] + D\mathcal{L}\big(f(\mathbf{v}_0)\big)\big[ D^2 f(\mathbf{v}_0)[\mathbf{h}, \mathbf{k}] \big]. \quad (\star)$$

By **(1)**, if at least one of the two directions is pure $\boldsymbol{\psi}$, both terms on the right-hand side of vanish. Therefore

$$D^2 R(\mathbf{v}_0)[\mathbf{h}, \mathbf{k}] = 0 \qquad \text{whenever at least one of } \mathbf{h}, \mathbf{k} \text{ is of the form } (\mathbf{0}_m, \ \cdot \ ).$$

Invoking Proposition A.14, this is exactly the statement that the $\boldsymbol{\xi}\boldsymbol{\psi}$, $\boldsymbol{\psi}\boldsymbol{\xi}$ and $\boldsymbol{\psi}\boldsymbol{\psi}$ Hessian blocks are $\mathbf{0}$. The remaining block $\nabla^2_{\boldsymbol{\xi}\boldsymbol{\xi}} R(\mathbf{v}_0)$ is whatever is induced by $(\star)$ for pairs

$$(\mathbf{h}, \mathbf{k}) = \big((\mathbf{h}_{\boldsymbol{\xi}}, \mathbf{0}_q), (\mathbf{k}_{\boldsymbol{\xi}}, \mathbf{0}_q)\big).$$

$\qquad\qquad\qquad\qquad\qquad\qquad\qquad\qquad\qquad\qquad\qquad\qquad\qquad\qquad\qquad\qquad\qquad\qquad\quad \square$

**Lemma C.2** (Spectrum under block-diagonal extension)**.** *Let $f \in C^2(\mathbb{R}^{m+q}; \mathbb{R})$, and fix $\mathbf{v} = (\boldsymbol{\xi}_0, \boldsymbol{\psi}_0) \in \mathbb{R}^{m+q}$. Assume the Hessian of $f$ at $\mathbf{v}$ has the block form*

$$\mathbf{H} := \nabla^2 f(\mathbf{v}) = \begin{pmatrix} \mathbf{B} & \mathbf{0}_{m \times q} \\ \mathbf{0}_{q \times m} & \mathbf{0}_{q \times q} \end{pmatrix}, \qquad \mathbf{B} \in \mathbb{R}^{m \times m}.$$

*Then the characteristic polynomial factorizes as*

$$\chi_{\mathbf{H}}(\lambda) := \det\big(\lambda \mathbf{I}_{m+q} - \mathbf{H}\big) = \det\big(\lambda \mathbf{I}_m - \mathbf{B}\big)\lambda^q.$$

*Consequently,*

$$\sigma(\mathbf{H}) = \sigma(\mathbf{B}) \cup \{0\}, \quad \text{and} \quad \mathrm{mult}_{\mathbf{H}}(0) = \mathrm{mult}_{\mathbf{B}}(0) + q,$$

*i.e., the spectrum of $H$ consists of the eigenvalues of $B$ together with $q$ additional zeros, and the algebraic multiplicity of the eigenvalue $0$ for $H$ equals that for $B$ plus $q$.*

*Proof.* Since $\mathbf{H}$ is block diagonal,

$$\lambda \mathbf{I}_{m+q} - \mathbf{H} = \begin{pmatrix} \lambda \mathbf{I}_m - \mathbf{B} & \mathbf{0}_{m \times q} \\ \mathbf{0}_{q \times m} & \lambda \mathbf{I}_q \end{pmatrix}.$$

The determinant of a block triangular (in particular block diagonal) matrix equals the product of the determinants of its diagonal blocks (e.g. Horn & Johnson 2013, Cor. 0.8.5). Hence

$$\chi_{\mathbf{H}}(\lambda) = \det(\lambda \mathbf{I}_m - \mathbf{B}) \cdot \det(\lambda \mathbf{I}_q) = \det(\lambda \mathbf{I}_m - \mathbf{B}) \cdot \lambda^q.$$

The zeros of $\chi_{\mathbf{H}}$ are the eigenvalues of $\mathbf{H}$ counted with algebraic multiplicity, which yields $\sigma(\mathbf{H}) = \sigma(\mathbf{B}) \cup \{0\}$ and $\mathrm{mult}_{\mathbf{H}}(0) = \mathrm{mult}_{\mathbf{B}}(0) + q$. $\qquad \square$

**Remark 14.** *If $0 \in \sigma(\mathbf{B})$, then $0$ appears in $\sigma(\mathbf{H})$ with multiplicity strictly larger than $q$; the statement above accounts for this by adding $q$ to the algebraic multiplicity of $0$ carried over from $\mathbf{B}$.*

**Lemma C.3** (Hessian of $\mathcal{L}$ w.r.t. $\mathbf{U}, \boldsymbol{\beta}$ at $\boldsymbol{\theta}_\star = \mathbf{0}$ and its spectrum). *Let $n := |\mathcal{V}|$ and $d$ be the embedding width. Fix $(\mathrm{s}, \mathbf{p}) \in \mathcal{V}^{\leq K} \times \Delta^{n-1}$, and consider the Transformer Language Model of Definition B.13. In the unembedding layer, set the LayerNorm scale to zero, $\boldsymbol{\gamma} = \mathbf{0}_d$. Let the parameter be ordered as*

$$\boldsymbol{\theta} = (\mathbf{u}, \boldsymbol{\beta}, \boldsymbol{\gamma}, \boldsymbol{\theta}'), \qquad \mathbf{u} := \mathrm{vec}_{n,d}(\mathbf{U}) \in \mathbb{R}^{nd}, \ \boldsymbol{\beta} \in \mathbb{R}^d.$$

*Restrict attention to the $(\mathbf{u}, \boldsymbol{\beta})$-coordinates and the base point*

$$\boldsymbol{\theta}_\star = \mathbf{0}_p \quad i.e. \quad \mathbf{U} = \mathbf{0}_{n \times d}, \ \boldsymbol{\beta} = \mathbf{0}_d, \ \boldsymbol{\gamma} = \mathbf{0}_d, \ \boldsymbol{\theta}' = 0.$$

*Write $\mathbf{b} := \frac{1}{n}\mathbf{1}_n$ and $\mathbf{w} := \mathbf{b} - \mathbf{p} \in \mathbb{R}^n$.*

*Then the Hessian of the cross-entropy loss*

$$\mathcal{L}(\boldsymbol{\theta}) = \mathrm{CrossEntropy}(f(\mathrm{s}\,; \boldsymbol{\theta}), \mathbf{p})$$

*with respect to $(\mathbf{u}, \boldsymbol{\beta})$ at $\boldsymbol{\theta}_\star$ is the symmetric block matrix*

$$\nabla^2_{(\mathbf{u}, \boldsymbol{\beta})} \mathcal{L}(\boldsymbol{\theta}_\star) = \begin{pmatrix} \mathbf{0}_{nd \times nd} & \mathbf{I}_d \otimes \mathbf{w} \\ \mathbf{I}_d \otimes \mathbf{w}^\top & \mathbf{0}_{d \times d} \end{pmatrix}.$$

*The spectrum of this Hessian is*

$$\mathrm{spec}(\nabla^2_{(\mathbf{u}, \boldsymbol{\beta})} \mathcal{L}(\boldsymbol{\theta}_\star)) = \{ \underbrace{+\|\mathbf{w}\|_2, \ldots, +\|\mathbf{w}\|_2}_{d}, \underbrace{-\|\mathbf{w}\|_2, \ldots, -\|\mathbf{w}\|_2}_{d}, \underbrace{0, \ldots, 0}_{d(n-1)} \}.$$

*Proof.*

**1) Logits in vectorized form.** With $\boldsymbol{\gamma} = \mathbf{0}_d$, the LayerNorm output at the unembedding is constant: $\mathrm{LN}(\mathbf{h}) \equiv \boldsymbol{\beta}$ (Definition B.9). Thus the logits before the final softmax are

$$\mathbf{Z} = \mathbf{U}\boldsymbol{\beta} \in \mathbb{R}^n.$$

Using $\mathrm{vec}(\mathbf{A}\mathbf{X}\mathbf{b}) = (\mathbf{b}^\top \otimes \mathbf{A})\mathrm{vec}(\mathbf{X})$ (standard identity for vectorization, cf. Henderson & Searle (1981)), with $\mathbf{A} = \mathbf{I}_n$ and $\mathbf{b} = \boldsymbol{\beta}$,

$$\mathbf{z} = \mathrm{vec}(\mathbf{Z}) = \mathrm{vec}(\mathbf{U}\boldsymbol{\beta}) = (\boldsymbol{\beta}^\top \otimes \mathbf{I}_n)\mathbf{u}.$$

Therefore, near $(\mathbf{u}, \boldsymbol{\beta}) = (\mathbf{0}_{nd}, \mathbf{0}_d)$, the logits map is the bilinear function

$$z(\mathbf{u}, \boldsymbol{\beta}) := (\boldsymbol{\beta}^\top \otimes \mathbf{I}_n)\mathbf{u} \in \mathbb{R}^n.$$

**2) First and second differentials.** Let $(\mathbf{h}, \boldsymbol{\eta})$ and $(\mathbf{k}, \boldsymbol{\xi})$ be directions in $\mathbb{R}^{nd} \times \mathbb{R}^d$. Differentiating $z(\mathbf{u}, \boldsymbol{\beta}) = (\boldsymbol{\beta}^\top \otimes \mathbf{I}_n)\mathbf{u}$ gives

$$Dz(\mathbf{u}, \boldsymbol{\beta})[\mathbf{h}, \boldsymbol{\eta}] = (\boldsymbol{\beta}^\top \otimes \mathbf{I}_n)\mathbf{h} + (\boldsymbol{\eta}^\top \otimes \mathbf{I}_n)\mathbf{u}.$$

At $(\mathbf{u}, \boldsymbol{\beta}) = (\mathbf{0}_{nd}, \mathbf{0}_d)$,

$$Dz(\mathbf{0}_{nd}, \mathbf{0}_d)[\mathbf{h}, \boldsymbol{\eta}] = \mathbf{0}_{n \times (nd+d)}$$

(since both terms are multiplied by $\mathbf{u}$ or $\boldsymbol{\beta}$). Differentiating once more (or, equivalently, using bilinearity of $z$) yields the constant symmetric bilinear form

$$D^2 z(\mathbf{0}_{nd}, \mathbf{0}_n)[(\mathbf{h}, \boldsymbol{\eta}), (\mathbf{k}, \boldsymbol{\xi})] = (\boldsymbol{\xi}^\top \otimes \mathbf{I}_n)\mathbf{h} + (\boldsymbol{\eta}^\top \otimes \mathbf{I}_n)\mathbf{k}.$$

**3) Gradient of the CE-in-softmax at the origin.** Let $F(\mathbf{z}) := \mathrm{CrossEntropy}(\mathrm{softmax}(\mathbf{z}), \mathbf{p})$. A standard computation (softmax Jacobian) gives

$$\nabla_{\mathbf{z}} F(\mathbf{z}) = \mathrm{softmax}(\mathbf{z}) - \mathbf{p}.$$

At $\mathbf{z} = \mathbf{0}_n$, $\mathrm{softmax}(\mathbf{0}_n) = \frac{1}{n}\mathbf{1}_n =: \mathbf{b}$, hence

$$\nabla_{\mathbf{z}} F(\mathbf{0}_n) = \mathbf{b} - \mathbf{p} =: \mathbf{w}.$$

**4) Second-order chain rule for $F \circ Z$ at $(\mathbf{0}, \mathbf{0})$.** Similarly to the proof of Lemma C.1, the second differential of a composition is

$$D^2(F \circ z)(\mathbf{v})[\mathbf{h}, \mathbf{k}] = D^2 F(z(\mathbf{v}))\big[Dz(\mathbf{v})\mathbf{h}, \ Dz(\mathbf{v})\mathbf{k}\big] + DF(z(\mathbf{v}))\big[D^2 z(\mathbf{v})[\mathbf{h}, \mathbf{k}]\big].$$

At $\mathbf{v} = (\mathbf{0}_{nd}, \mathbf{0}_d)$, $Dz(\mathbf{v}) = \mathbf{0}_{n \times (nd+d)}$ and $DF(z(\mathbf{v})) = \nabla_{\mathbf{z}} F(\mathbf{0}_n)^\top = \mathbf{w}^\top$, so

$$\begin{aligned}
D^2 \mathcal{L}(\mathbf{v})\big[(\mathbf{h}, \boldsymbol{\eta}), (\mathbf{k}, \boldsymbol{\xi})\big] &= \mathbf{w}^\top D^2 z(\mathbf{v})\big[(\mathbf{h}, \boldsymbol{\eta}), (\mathbf{k}, \boldsymbol{\xi})\big] \\
&= \mathbf{w}^\top \big((\boldsymbol{\xi}^\top \otimes \mathbf{I}_n)\mathbf{h} + (\boldsymbol{\eta}^\top \otimes \mathbf{I}_n)\mathbf{k}\big) \\
&= \mathbf{h}^\top (\mathbf{I}_d \otimes \mathbf{w})\,\boldsymbol{\xi} \ + \ \mathbf{k}^\top (\mathbf{I}_d \otimes \mathbf{w})\,\boldsymbol{\eta},
\end{aligned}$$

where we used the mixed-product rule for Kronecker products and the identity

$$\mathbf{w}^\top(\boldsymbol{\xi}^\top \otimes \mathbf{I}_n) = \boldsymbol{\xi}^\top \otimes \mathbf{w}^\top.$$

**5) Identification of the Hessian blocks.** By definition of the Hessian as a bilinear form,

$$D^2 \mathcal{L}(\mathbf{v})\big[(\mathbf{h}, \boldsymbol{\eta}), (\mathbf{k}, \boldsymbol{\xi})\big] = \begin{pmatrix} \mathbf{h}^\top & \boldsymbol{\eta}^\top \end{pmatrix} \begin{pmatrix} \mathbf{0}_{nd \times nd} & \frac{\partial^2 \mathcal{L}}{\partial \mathbf{u}\,\partial \boldsymbol{\beta}} \\ \frac{\partial^2 \mathcal{L}}{\partial \boldsymbol{\beta}\,\partial \mathbf{u}} & \mathbf{0}_{d \times d} \end{pmatrix} \begin{pmatrix} \mathbf{k} \\ \boldsymbol{\xi} \end{pmatrix}.$$

Comparing with the expression obtained in Step 4 for arbitrary $(\mathbf{h}, \boldsymbol{\eta})$ and $(\mathbf{k}, \boldsymbol{\xi})$ forces

$$\frac{\partial^2 \mathcal{L}}{\partial \mathbf{u}\,\partial \boldsymbol{\beta}}(\boldsymbol{\theta}_\star) = \mathbf{I}_d \otimes \mathbf{w}, \qquad \frac{\partial^2 \mathcal{L}}{\partial \boldsymbol{\beta}\,\partial \mathbf{u}}(\boldsymbol{\theta}_\star) = \big(\mathbf{I}_d \otimes \mathbf{w}\big)^\top = \mathbf{I}_d \otimes \mathbf{w}^\top,$$

and, because $Dz(\mathbf{v}) = \mathbf{0}_{n \times (nd+d)}$ (so no quadratic term survives in either $\mathbf{u}$ or $\boldsymbol{\beta}$ alone),

$$\frac{\partial^2 \mathcal{L}}{\partial \mathbf{u}\,\partial \mathbf{u}}(\boldsymbol{\theta}_\star) = \mathbf{0}_{nd \times nd}, \qquad \frac{\partial^2 \mathcal{L}}{\partial \boldsymbol{\beta}\,\partial \boldsymbol{\beta}}(\boldsymbol{\theta}_\star) = \mathbf{0}_{d \times d}.$$

This gives exactly the claimed block matrix.

**6) Spectrum.** Let

$$\mathbf{H} := \nabla^2_{(\mathbf{u}, \boldsymbol{\beta})} \mathcal{L}(\boldsymbol{\theta}_\star) = \begin{pmatrix} \mathbf{0}_{nd \times nd} & \mathbf{I}_d \otimes \mathbf{w} \\ \mathbf{I}_d \otimes \mathbf{w}^\top & \mathbf{0}_{d \times d} \end{pmatrix}.$$

Then

$$\mathbf{H}^2 = \begin{pmatrix} (\mathbf{I}_d \otimes \mathbf{w})(\mathbf{I}_d \otimes \mathbf{w}^\top) & \mathbf{0}_{nd \times d} \\ \mathbf{0}_{d \times nd} & (\mathbf{I}_d \otimes \mathbf{w}^\top)(\mathbf{I}_d \otimes \mathbf{w}) \end{pmatrix} = \begin{pmatrix} \mathbf{I}_d \otimes (\mathbf{w}\mathbf{w}^\top) & \mathbf{0}_{nd \times d} \\ \mathbf{0}_{d \times nd} & \mathbf{I}_d \otimes (\mathbf{w}^\top\mathbf{w}) \end{pmatrix}.$$

The eigenvalues of $\mathbf{w}\mathbf{w}^\top$ are $\|\mathbf{w}\|_2^2$ (multiplicity 1) and 0 (multiplicity $n - 1$); the eigenvalues of $\mathbf{w}^\top\mathbf{w}$ equal $\|\mathbf{w}\|_2^2$ (scalar). Therefore the eigenvalues of $\mathbf{H}^2$ are

$$\underbrace{\|\mathbf{w}\|_2^2, \ldots, \|\mathbf{w}\|_2^2}_{2d \text{ times}}, \quad \underbrace{0, \ldots, 0}_{d(n-1) \text{ times}} \ .$$

Because $\mathbf{H}$ is symmetric, its eigenvalues are the real square-roots of those of $\mathbf{H}^2$, namely $\pm\|\mathbf{w}\|_2$ (each with multiplicity $d$) and 0 (with multiplicity $d(n-1)$). This is exactly the set stated in the lemma. $\qquad \square$

**Lemma C.4** (Full Hessian at the witness: block form and spectrum). *Let $n := |\mathcal{V}|$ and $d$ be the embedding width. Write the parameter as*

$$\boldsymbol{\theta} = \big((\mathbf{u}, \boldsymbol{\beta}), (\boldsymbol{\gamma}, \boldsymbol{\theta}')\big), \qquad \mathbf{u} = \text{vec}_{n,d}(\mathbf{U}) \in \mathbb{R}^{nd}, \ \boldsymbol{\beta}, \boldsymbol{\gamma} \in \mathbb{R}^d, \ \boldsymbol{\theta}' \in \mathbb{R}^{p'},$$

*so $p = nd + 2d + p'$. Consider the witness point*

$$\boldsymbol{\theta}_\star = \mathbf{0}_p \quad (\mathbf{U} = \mathbf{0}_{n \times d}, \ \boldsymbol{\beta} = \mathbf{0}_d, \ \boldsymbol{\gamma} = \mathbf{0}_d, \ \boldsymbol{\theta}' = \mathbf{0}_d).$$

*Let $\mathbf{b} := \frac{1}{n}\mathbf{1}_n$ and $\mathbf{w} := \mathbf{b} - \mathbf{p} \in \mathbb{R}^n$. Then the Hessian of the cross-entropy loss $\mathcal{L}(\boldsymbol{\theta})$ at $\boldsymbol{\theta}_\star$ admits the block-diagonal decomposition*

$$\nabla^2 \mathcal{L}(\boldsymbol{\theta}_\star) = \begin{pmatrix} \mathbf{B} & \mathbf{0} \\ \mathbf{0} & \mathbf{0} \end{pmatrix}, \qquad \mathbf{B} = \begin{pmatrix} \mathbf{0}_{nd \times nd} & \mathbf{I}_d \otimes \mathbf{w} \\ \mathbf{I}_d \otimes \mathbf{w}^\top & \mathbf{0}_{d \times d} \end{pmatrix}.$$

*Consequently,*

$$\text{spec}\big(\nabla^2 \mathcal{L}(\boldsymbol{\theta}_\star)\big) = \Big\{ \underbrace{+\|\mathbf{w}\|_2, \ldots, +\|\mathbf{w}\|_2}_{d}, \ \underbrace{-\|\mathbf{w}\|_2, \ldots, -\|\mathbf{w}\|_2}_{d}, \ \underbrace{0, \ldots, 0}_{p-2d} \Big\}.$$

*Proof.* Set $\boldsymbol{\gamma} = \mathbf{0}_d$. Then the unembedding LayerNorm output is constant, $\mathrm{LN}(\mathbf{h}) \equiv \boldsymbol{\beta}$, so the logits equal $\mathbf{z} = \mathbf{U}\boldsymbol{\beta}$. Hence, in a neighborhood of $\boldsymbol{\theta}_\star$, the loss depends only on $(\mathbf{u}, \boldsymbol{\beta})$ and is independent of $(\boldsymbol{\gamma}, \boldsymbol{\theta}')$.

We will apply Lemma C.1 with the open set $\mathcal{U} = \mathbb{R}^{nd+2d+p'}$, coordinates $\boldsymbol{\xi} = (\mathbf{u}, \boldsymbol{\beta})$ and $\boldsymbol{\psi} = (\boldsymbol{\gamma}, \boldsymbol{\theta}')$ and with $n = |\mathcal{V}|$, $r = d$. Define

$$g(\boldsymbol{\xi}) := \mathrm{mat}_{n,d}(\mathbf{u}) \in \mathbb{R}^{n \times d}, \qquad h(\boldsymbol{\xi}, \boldsymbol{\psi}) := \boldsymbol{\beta} \in \mathbb{R}^d,$$

so that

$$f(\boldsymbol{\xi}, \boldsymbol{\psi}) := g(\boldsymbol{\xi})\, h(\boldsymbol{\xi}, \boldsymbol{\psi}) = \mathbf{U}\boldsymbol{\beta} \in \mathbb{R}^n,$$

and, with $\mathcal{L}(\mathbf{z}) := \mathrm{CrossEntropy}(\mathrm{softmax}(\mathbf{z}), \mathbf{p})$,

$$R(\boldsymbol{\xi}, \boldsymbol{\psi}) := \mathcal{L}(f(\boldsymbol{\xi}, \boldsymbol{\psi})) = \mathrm{CrossEntropy}(\mathrm{softmax}(\mathbf{U}\boldsymbol{\beta}), \mathbf{p}).$$

At the witness $\mathbf{v}_0 = (\boldsymbol{\xi}_0, \boldsymbol{\psi}_0)$ we have $g(\boldsymbol{\xi}_0) = \mathbf{0}_{n \times d}$, so by Lemma C.1 all mixed and $\boldsymbol{\psi}$–only second partials of $R$ vanish at $\mathbf{v}_0$, i.e.

$$\nabla^2 R(\mathbf{v}_0) = \begin{pmatrix} \nabla^2_{(\mathbf{u},\boldsymbol{\beta})} R(\mathbf{v}_0) & \mathbf{0} \\ \mathbf{0} & \mathbf{0} \end{pmatrix}.$$

Identifying $R(\boldsymbol{\xi}, \boldsymbol{\psi}) \equiv \mathcal{L}(\boldsymbol{\theta})$ under the correspondence above yields

$$\nabla^2 \mathcal{L}(\boldsymbol{\theta}_\star) = \begin{pmatrix} \nabla^2_{(\mathbf{u},\boldsymbol{\beta})} \mathcal{L}(\boldsymbol{\theta}_\star) & \mathbf{0} \\ \mathbf{0} & \mathbf{0} \end{pmatrix}.$$

Combining, Lemmas C.2 and C.3, we get that

$$\mathrm{spec}(\nabla^2 \mathcal{L}(\boldsymbol{\theta}_\star)) = \mathrm{spec}(\nabla^2_{(\mathbf{u},\boldsymbol{\beta})} \mathcal{L}(\boldsymbol{\theta}_\star)) \cup \{0\}^{d+p'}$$

$$= \left\{ \pm \|\mathbf{w}\|_2 \text{ (each mult. } d\text{)},\ 0 \text{ (mult. } d(n-1)+d+p'\text{)} \right\}.$$

Since $p = nd + 2d + p'$, the multiplicity of $0$ equals $p - 2d$, which yields the claimed spectrum. $\qquad\square$

**Theorem C.3** (GD Jacobian is nondegenerate a.e.). *Fix a finite vocabulary $\mathcal{V}$, a context bound $K \in \mathbb{N}$, and the Transformer language model $f$ of Definition B.13. For any sample $(\mathrm{s}, \mathbf{p}) \in \mathcal{V}^{\leq K} \times \Delta^{|\mathcal{V}|-1}$ and any learning rate $\eta \in (0,1)$, let $\phi : \mathbb{R}^p \to \mathbb{R}^p$ be the gradient-descent update, defined as:*

$$\phi(\boldsymbol{\theta}) = \boldsymbol{\theta} - \eta\, \nabla_{\boldsymbol{\theta}} \mathcal{L}_{\mathrm{s},\mathbf{p}}(\boldsymbol{\theta}),$$

*where $\mathcal{L}_{\mathrm{s},\mathbf{p}} : \mathbb{R}^p \to \mathbb{R}$ is the standard Cross Entropy loss:*

$$\mathcal{L}_{\mathrm{s},\mathbf{p}}(\boldsymbol{\theta}) = \mathrm{CrossEntropy}(f(\mathrm{s}\,;\boldsymbol{\theta}), \mathbf{p}).$$

*Then the critical set*

$$\mathcal{C} := \{\boldsymbol{\theta} \in \mathbb{R}^p : \det D\phi(\boldsymbol{\theta}) = 0\}$$

*has Lebesgue measure zero in $\mathbb{R}^p$.*

*Proof.* By Propositions A.6 and B.3 and the closure properties of real analyticity, $\mathcal{L}_{\mathrm{s},\mathbf{p}}$ is real-analytic; hence so are its gradient and Hessian. Therefore $\phi$ is real-analytic (Lewis, 2014, Thm. 1.1.15) and

$$D\phi(\boldsymbol{\theta}) = \mathbf{I}_p - \eta\, \nabla^2_{\boldsymbol{\theta}} \mathcal{L}_{\mathrm{s},\mathbf{p}}(\boldsymbol{\theta}).$$

Since the determinant is a polynomial in the entries, $\boldsymbol{\theta} \mapsto \det D\phi(\boldsymbol{\theta})$ is real-analytic.

It is not identically zero: at the witness $\boldsymbol{\theta}_\star = \mathbf{0}_p$, Lemma C.4 gives

$$\mathrm{spec}(\nabla^2 \mathcal{L}(\boldsymbol{\theta}_\star)) = \{\underbrace{+\|\mathbf{w}\|_2, \ldots, +\|\mathbf{w}\|_2}_{d}, \underbrace{-\|\mathbf{w}\|_2, \ldots, -\|\mathbf{w}\|_2}_{d}, \underbrace{0, \ldots, 0}_{p-2d}\}, \quad \mathbf{w} := \tfrac{1}{n}\mathbf{1} - \mathbf{p}.$$

Hence the eigenvalues of $D\phi(\boldsymbol{\theta}_\star) = \mathbf{I}_p - \eta\, \nabla^2 \mathcal{L}(\boldsymbol{\theta}_\star)$ are

$$\underbrace{1 - \eta\|\mathbf{w}\|_2}_{d \text{ times}}, \quad \underbrace{1 + \eta\|\mathbf{w}\|_2}_{d \text{ times}}, \quad \underbrace{1}_{p-2d \text{ times}},$$

so

$$\det D\phi(\boldsymbol{\theta}_\star) = (1 - \eta^2 \|\mathbf{w}\|_2^2)^d > 0.$$

Thus $\det D\phi$ is a nontrivial real-analytic function. By Theorem A.1, its zero set has Lebesgue measure 0. $\qquad\square$

### C.2.2 GRADIENT DESCENT PRESERVES ABSOLUTE CONTINUITY

With the witness in hand and the critical set shown to be measure-zero (Theorem C.3), we now carry out Steps 3–4 of the roadmap: cover the non-critical region by countably many diffeomorphic charts (Lemma C.5), use the change-of-variables formula to show preimages of null sets remain null (Lemma C.6), and conclude that one GD step preserves absolute continuity (Theorem C.5). Iterating yields the finite-horizon corollary (Corollary C.5.1).

**Lemma C.5** (Countable chart cover of $\mathbb{R}^p \setminus \mathcal{C}$). *Consider the setup of Theorem C.5. In particular, let $\phi : \mathbb{R}^p \to \mathbb{R}^p$ be the one-step GD map from that theorem:*

$$\phi(\boldsymbol{\theta}) = \boldsymbol{\theta} - \eta \, \nabla_{\boldsymbol{\theta}} \mathcal{L}_{\mathrm{s},\mathbf{p}}(\boldsymbol{\theta}), \tag{32}$$

*with stepsize $\eta \in (0, 1)$, and the measure-zero critical-set (Theorem C.3):*

$$\mathcal{C} := \{ \boldsymbol{\theta} \in \mathbb{R}^p : \det D\phi(\boldsymbol{\theta}) = 0 \}.$$

*Then there exist open sets $(\mathcal{U}_k)_{k \geq 1}$ covering $\mathcal{X} := \mathbb{R}^p \setminus \mathcal{C}$ such that, for each $k$, the restriction $\phi_k := \phi|_{\mathcal{U}_k} : \mathcal{U}_k \to \mathcal{V}_k := \phi(\mathcal{U}_k)$ is a $C^1$ diffeomorphism with $C^1$ inverse $\psi_k := \phi_k^{-1}$.*

*Proof.*

**1) $\mathcal{X}$ is open:** By Propositions A.6 and B.3 and the closure rules of real-analyticity, $\mathcal{L}_{\mathrm{s},\mathbf{p}}$ is $C^2$, hence $\phi$ is $C^1$. The map $\boldsymbol{\theta} \mapsto D\phi(\boldsymbol{\theta})$ is continuous, and the determinant is a continuous polynomial in the entries, so $g(\boldsymbol{\theta}) := \det D\phi(\boldsymbol{\theta})$ is continuous. Therefore $\mathcal{C} = g^{-1}(\{0\})$ is closed (Rudin, 1976, Thm. 4.8) and $\mathcal{X} = \mathbb{R}^p \setminus \mathcal{C}$ is open.

**2) Local diffeomorphisms by the Inverse Function Theorem:** Fix $\boldsymbol{\theta} \in \mathcal{X}$. Then $g(\boldsymbol{\theta}) \neq 0$, so by the Inverse Function Theorem (Theorem A.2) there exist open neighborhoods $\mathcal{U}_{\boldsymbol{\theta}} \ni \boldsymbol{\theta}$ and $\mathcal{V}_{\boldsymbol{\theta}} \ni \phi(\boldsymbol{\theta})$ such that

$$\phi_{\boldsymbol{\theta}} := \phi|_{\mathcal{U}_{\boldsymbol{\theta}}} : \mathcal{U}_{\boldsymbol{\theta}} \to \mathcal{V}_{\boldsymbol{\theta}}$$

is a $C^1$ diffeomorphism with $C^1$ inverse $\psi_{\boldsymbol{\theta}} := \phi_{\boldsymbol{\theta}}^{-1}$. Moreover,

$$D\psi_{\boldsymbol{\theta}}(\phi(\mathbf{x})) = \left( D\phi(\mathbf{x}) \right)^{-1} \qquad \forall \mathbf{x} \in \mathcal{U}_{\boldsymbol{\theta}}.$$

In particular $D\phi(\mathbf{x})$ is invertible for all $\mathbf{x} \in \mathcal{U}_{\boldsymbol{\theta}}$, whence $\mathcal{U}_{\boldsymbol{\theta}} \subset \mathcal{X}$. Thus $\{\mathcal{U}_{\boldsymbol{\theta}}\}_{\boldsymbol{\theta} \in \mathcal{X}}$ is an open cover of $\mathcal{X}$ by IFT charts.

**3) Select a countable subcover:** By Proposition A.15(3), $\mathbb{R}^p$ is second-countable; subspaces of second-countable spaces are second-countable, hence $\mathcal{X}$ is second-countable. By Proposition A.15(4), every open cover of a second-countable space admits a countable subcover. Therefore there exist points $\boldsymbol{\theta}_1, \boldsymbol{\theta}_2, \ldots \in \mathcal{X}$ such that $\mathcal{X} = \bigcup_{k=1}^{\infty} \mathcal{U}_{\boldsymbol{\theta}_k}$.

Set $\mathcal{U}_k := \mathcal{U}_{\boldsymbol{\theta}_k}$, $\mathcal{V}_k := \mathcal{V}_{\boldsymbol{\theta}_k}$, and $\phi_k := \phi|_{\mathcal{U}_k} = \phi_{\boldsymbol{\theta}_k}$, $\psi_k := \psi_{\boldsymbol{\theta}_k}$. Each $\phi_k$ is a $C^1$ diffeomorphism with $C^1$ inverse $\psi_k$ by Step 2. This yields the desired countable chart cover of $\mathcal{X}$. $\square$

**Theorem C.4** (Change of Variables Folland 1999, Thm. 2.47(b)). *Let $\mathcal{U}, \mathcal{V} \subseteq \mathbb{R}^p$ be open and $\psi : \mathcal{V} \to \mathcal{U}$ a $C^1$ diffeomorphism. If $\mathcal{E} \subseteq \mathcal{V}$ is Lebesgue measurable, then*

$$\mathrm{Leb}_p\big(\psi(\mathcal{E})\big) = \int_{\mathcal{E}} \left| \det D\psi(\mathbf{y}) \right| d\mathbf{y}.$$

**Lemma C.6** (Pre-images of null sets are null). *Consider the setup of Theorem C.5, in particular the $C^1$ gradient descent map:*

$$\phi(\boldsymbol{\theta}) = \boldsymbol{\theta} - \eta \nabla_{\boldsymbol{\theta}} \mathcal{L}_{\mathrm{s},\mathbf{p}}(\boldsymbol{\theta}), \qquad \eta \in (0, 1),$$

*and its critical set $\mathcal{C} := \{ \boldsymbol{\theta} \in \mathbb{R}^p : \det D\phi(\boldsymbol{\theta}) = 0 \}$. Then, for every measurable $\mathcal{A} \subseteq \mathbb{R}^p$,*

$$\mathrm{Leb}_p(\mathcal{A}) = 0 \implies \mathrm{Leb}_p\big(\phi^{-1}(\mathcal{A})\big) = 0.$$

*Proof.* Let $\mathcal{X} = \mathbb{R}^p \setminus \mathcal{C}$ and decompose the pre-image:

$$\phi^{-1}(\mathcal{A}) = \big(\phi^{-1}(\mathcal{A}) \cap \mathcal{C}\big) \cup \big(\phi^{-1}(\mathcal{A}) \cap \mathcal{X}\big).$$

The first set is contained in $\mathcal{C}$, a measure zero set (Theorem C.3), hence has $\mathrm{Leb}_p$–measure $0$. By Lemma C.5, cover $\mathcal{X}$ by countably many charts $\{\mathcal{U}_k\}$ on which $\phi_k := \phi|_{\mathcal{U}_k}$ is a $C^1$ diffeomorphism onto $\mathcal{V}_k := \phi(\mathcal{U}_k)$ with inverse $\psi_k \in C^1(\mathcal{V}_k \,;\, \mathcal{U}_k)$. Then, it holds that:

$$\phi^{-1}(\mathcal{A}) \cap \mathcal{U}_k = \psi_k(\mathcal{A} \cap \mathcal{V}_k).$$

Since $\mathrm{Leb}_p(\mathcal{A}) = 0$ and both $\mathcal{A}$ and $\mathcal{V}_k$ are measurable, $\mathcal{A} \cap \mathcal{V}_k$ is measurable and has measure $0$. By Theorem C.4 applied to $\psi_k$ with $\mathcal{E} = \mathcal{A} \cap \mathcal{V}_k$,

$$\mathrm{Leb}_p\big(\psi_k(\mathcal{A} \cap \mathcal{V}_k)\big) = \int_{\mathcal{A} \cap \mathcal{V}_k} \big| \det D\psi_k(\mathbf{y}) \big| \, d\mathbf{y} = 0.$$

Therefore, each $\phi^{-1}(\mathcal{A}) \cap \mathcal{U}_k$ is null and because a countable union of null sets is null, it holds that:

$$\mathrm{Leb}_p\big(\phi^{-1}(\mathcal{A})\big) = 0.$$

$\square$

**Theorem C.5** (Preservation of absolute continuity under one GD step). *Consider the setup of Theorem C.3. In particular, let $\phi : \mathbb{R}^p \to \mathbb{R}^p$ be the one-step GD map from that theorem:*

$$\phi(\boldsymbol{\theta}) = \boldsymbol{\theta} - \eta \, \nabla_{\boldsymbol{\theta}} \mathcal{L}_{\mathrm{s},\mathbf{p}}(\boldsymbol{\theta}), \tag{33}$$

*with stepsize $\eta \in (0,1)$. Then, gradient-descent preserves absolute continuity: for every absolutely continuous probability law $\mu$ on $\mathbb{R}^p$, its image under $\phi$ remains absolutely continuous:*

$$\phi_{\#}\mu \;\ll\; \mathrm{Leb}_p.$$

*Therefore, the updated parameters $\boldsymbol{\theta}' := \phi(\boldsymbol{\theta})$ are absolutely continuous.*

*Proof.* By Proposition B.3 and closure properties, $\mathcal{L}_{\mathrm{s},\mathbf{p}}$ is $C^2$, hence $\phi \in C^1$ and is Borel-measurable. From Theorem C.3 the critical set

$$\mathcal{C} \; := \; \{\boldsymbol{\theta} \in \mathbb{R}^p : \det D\phi(\boldsymbol{\theta}) = 0\}$$

has $\mathrm{Leb}_p$-measure $0$. Therefore, the hypothesis of Lemma C.6 holds, and we have the property:

$$\mathrm{Leb}_p(\mathcal{A}) = 0 \quad \Longrightarrow \quad \mathrm{Leb}_p\big(\phi^{-1}(\mathcal{A})\big) = 0 \qquad \text{for every measurable } \mathcal{A} \subseteq \mathbb{R}^p. \tag{$\dagger$}$$

Let $\mathcal{A}$ be any Borel set with $\mathrm{Leb}_p(\mathcal{A}) = 0$. Then

$$\phi_{\#}\mu(\mathcal{A}) \; = \; \mu\big(\phi^{-1}(\mathcal{A})\big) \; = \; 0,$$

because $\mu \ll \mathrm{Leb}_p$ and $\mathrm{Leb}_p\big(\phi^{-1}(\mathcal{A})\big) = 0$ by ($\dagger$). Since this holds for every $\mathrm{Leb}_p$-null set $\mathcal{A}$, we conclude $\phi_{\#}\mu \ll \mathrm{Leb}_p$. $\square$

**Corollary C.5.1** (Preservation of absolute continuity under finitely many GD steps). *Fix a finite vocabulary $\mathcal{V}$, a context bound $K \in \mathbb{N}$, and the Transformer language model $f$ of Definition B.13. For $t = 1, \ldots, T$, let $(\mathrm{s}_t, \mathbf{p}_t) \in \mathcal{V}^{\leq K} \times \Delta^{|\mathcal{V}|-1}$ and $\eta_t \in (0,1)$, and define the $t$-th GD update*

$$\phi_t(\boldsymbol{\theta}) \; = \; \boldsymbol{\theta} - \eta_t \, \nabla_{\boldsymbol{\theta}} \mathcal{L}_{\mathrm{s}_t,\mathbf{p}_t}(\boldsymbol{\theta}), \qquad \mathcal{L}_{\mathrm{s}_t,\mathbf{p}_t}(\boldsymbol{\theta}) = \mathrm{CrossEntropy}\big(f(\mathrm{s}_t \,;\, \boldsymbol{\theta}), \mathbf{p}_t\big).$$

*Let the $T$-step update map be the composition*

$$\Phi \; := \; \phi_T \circ \cdots \circ \phi_1 \; : \; \mathbb{R}^p \to \mathbb{R}^p.$$

*Then, for every absolutely continuous probability law $\mu$ on $\mathbb{R}^p$, its image under $\Phi$ remains absolutely continuous:*

$$\Phi_{\#}\mu \;\ll\; \mathrm{Leb}_p.$$

*Equivalently, if $\boldsymbol{\theta}^{(0)} \sim \mu$ with $\mu \ll \mathrm{Leb}_p$ and*

$$\boldsymbol{\theta}^{(t+1)} \; = \; \phi_t\big(\boldsymbol{\theta}^{(t)}\big), \quad t = 0, \ldots, T-1,$$

*then the $T$-step parameters $\boldsymbol{\theta}^{(T)} = \Phi\big(\boldsymbol{\theta}^{(0)}\big)$ are absolutely continuous.*

*Proof.* Since the result of Lemma C.6 holds for each $\phi_t$, for any null set $\mathcal{A}$, repeated preimages remain null:

$$\mathrm{Leb}_p\big((\phi_T \circ \cdots \circ \phi_1)^{-1}(\mathcal{A})\big) = 0.$$

The same argument as in the proof of Theorem C.5 then yields the claim. $\square$

# D LEFT-INVERTIBILITY VIA SIPIT

**Goal.**  We study when and how the hidden states of a causal decoder-only Transformer admit a *left inverse*: given the layer-$\ell$ representation at position $t$ and the true prefix $\pi = s_{1:t-1}$, can we recover the next token $s_t$?

**Main idea.**  Under mild randomness in the parameters and causal masking, the *one-step last-token map* that sends a candidate token $v$ to the layer-$\ell$ representation at position $t$ (conditioning on $\pi$) is almost-surely injective, and in fact has a positive separation margin. This yields a simple verifier: declare $v$ correct iff the observed hidden state lies in a small ball around $F(v; \pi, t)$.

**Algorithmic consequence.**  Because causality localizes the dependence to $(\pi, s_t)$, we can invert an entire sequence sequentially with a single pass over the vocabulary per position. We call this procedure SIPIT (Sequential Inverse Prompt via ITerative updates), and we show exact (and robust) recovery holds almost surely, with worst-case time $\Theta(T|\mathcal{V}|)$.

**Standing conventions for this section.**  Fix a layer index $\ell \in [L]$. For any input sequence $s = \langle s_1, \ldots, s_T \rangle$, define the layer outputs row-wise by

$$\mathbf{H}^{(0)}(s) := \mathrm{Emb}(s), \qquad \mathbf{H}^{(\ell)}(s) := \mathrm{TB}^{(\ell)}\big(\mathbf{H}^{(\ell-1)}(s)\big) \in \mathbb{R}^{T \times d},$$

and write $\mathbf{h}_t(s)$ to denote the row of $\mathbf{H}^{(\ell)}(s)$ at position $t$. Furthermore, we use $\oplus$ for sequence concatenation: if $s = \langle s_1, \ldots, s_{t-1} \rangle$ and $v \in \mathcal{V}$, then $s \oplus v = \langle s_1, \ldots, s_{t-1}, v \rangle$.

The parameters $\boldsymbol{\theta}$ and target layer $\ell$ are considered fixed and omitted for simplicity.

**Assumption D.1** (Causal self-attention throughout). *Every attention layer in every block is* causal *in the sense of Definitions B.6 and B.7. Consequently, for any $s$ and any $t \in [T]$,*

$$\mathbf{h}_t(s) \text{ depends only on the prefix } \langle s_1, \ldots, s_t \rangle. \tag{34}$$

**Assumption D.2** (Injectivity Assumption). SIPIT *is applied to models initialized with parameters drawn from an absolutely continuous distribution and trained via (mini-batch) gradient descent with step sizes in $(0, 1)$, as described in Appendix C. Under these conditions, any network considered in the sequel is almost-surely injective (Theorem C.1).*

## D.1 ONE-STEP LAST-TOKEN MAPS

We first isolate the positionwise map that drives inversion. Fix a position $t$ and prefix $\pi \in \mathcal{V}^{t-1}$. The *one-step map* $F(\cdot; \pi, t)$ sends a candidate token $v$ to the layer-$\ell$ hidden state at position $t$ obtained when the prefix is $\pi$ and the token at $t$ is $v$. Causality implies that $\mathbf{h}_t$ depends only on $(\pi, v)$ (not on any future tokens), and we show that, for almost all parameter settings, $F$ is injective with a strictly positive pairwise margin over $\mathcal{V}$.

**Definition D.1** (One-step map at time $t$ under prefix $\pi$). *Let $\pi \in \mathcal{V}^{t-1}$ be a fixed prefix (possibly $t = 1$, when $\pi$ is empty). Define*

$$F : \mathcal{V} \longrightarrow \mathbb{R}^d, \qquad F(v; \pi, t) := \mathbf{h}_t(\pi \oplus v).$$

**Remark 15.** *$F$ is simply a function that returns the hidden output of token $v$ at the $\ell$ transformer block given that $\pi$ is used a fixed prefix. This map allows us to have a convenient notation for introducing results about inversion. Furthermore, since $F$ is built using $\ell$ transformer blocks, it is parameterized by $\boldsymbol{\theta}$. Nevertheless, for the sake of simplicity, we will refer to $F_{\ell, \boldsymbol{\theta}}$ simply as $F$.*

Once the One-step map (Definition D.1) is introduced, one can present its a.s. injectivity through an application of the previously obtained result (Theorem C.1). Furthermore, one can deploy the common prefix to introduce a stronger notion of injectivity: margin separation (Lemma D.1).

**Theorem D.1** (A.s. one-step injectivity). *Fix $t$ and the prefix $\pi \in \mathcal{V}^{t-1}$. Under Assumptions D.1 and D.2, it holds that:*

$$\Pr\left[ \exists v \neq v' \in \mathcal{V} : F(v; \pi, t) = F(v'; \pi, t) \right] = 0.$$

*Equivalently, $F$ is injective almost-surely.*

*Proof.* Set the finite family $\mathcal{S}_{t,\pi} := \{\pi \oplus v : v \in \mathcal{V}\} \subseteq \mathcal{V}^t$ and view $\mathbf{h}_t(s)$ as the last-token representation of the *truncated* Transformer consisting of the first $\ell$ blocks. All assumptions used in Corollary C.2.1 remain valid for this truncated model. Applying the corollary with $\mathcal{S} = \mathcal{S}_{t,\pi}$ yields, almost-surely, $\mathbf{h}_t(\pi \oplus v) \neq \mathbf{h}_t(\pi \oplus v')$ whenever $v \neq v'$. This is exactly the injectivity of $F$. $\square$

**Lemma D.1** (Strict separation margin a.s.). *Under the conditions of Theorem D.1, define the (data-dependent) margin*

$$\Delta_{\pi,t} := \min_{v \neq v' \in \mathcal{V}} \left\| F(v \,;\, \pi, t) - F(v' \,;\, \pi, t) \right\|_2$$

*Then,*

$$\Pr[\Delta_{\pi,t} > 0] = 1.$$

*Proof.* By Theorem D.1, with probability 1 the set

$$\{F(v \,;\, \pi, t) : v \in \mathcal{V}\}$$

consists of $|\mathcal{V}|$ distinct points in $\mathbb{R}^d$. On this event of full probability, every pairwise distance among these finitely many points is strictly positive, so their minimum is strictly positive as well.

Thus, the event $\{\Delta_{\pi,t} > 0\}$ coincides with the event that $F$ is injective on $\mathcal{V}$. Since injectivity holds almost-surely by assumption, we conclude that $\Pr[\Delta_{\pi,t} > 0] = 1$. $\square$

## D.2 THE CORE ROUTINES: LOCAL VERIFIERS, ACCEPTANCE REGIONS, AND POLICIES

Given $F(\cdot \,;\, \pi, t)$, inversion reduces to a local hypothesis test: for an observed $\widehat{\mathbf{h}}_t$, which token's predicted representation is closest? We formalize this with *acceptance regions*–closed balls around $F(v \,;\, \pi, t)$–and a *verifier* that accepts $v$ iff $\widehat{\mathbf{h}}_t$ lies in its ball. Almost-sure injectivity yields uniqueness at radius 0, and a positive margin yields uniqueness for any $\varepsilon < \Delta_{\pi,t}/2$. To explore candidates efficiently, we couple the verifier with any *policy* that enumerates untried tokens (e.g., uniform without replacement or a gradient-guided ranking).

**Definition D.2** (Local *verifier* and acceptance tolerance). *Given a tolerance $\varepsilon \geq 0$, define the acceptance region for symbol $v$ as the closed ball (Definition A.8):*

$$\mathcal{A}_{\pi,t}(v \,;\, \varepsilon) := \overline{B}\big(F(v \,;\, \pi, t), \varepsilon\big).$$

*A candidate token $v \in \mathcal{V}$ is* verified *for observation $\widehat{\mathbf{h}}_t$ if and only if $\widehat{\mathbf{h}}_t \in \mathcal{A}_{\pi,t}(v \,;\, \varepsilon)$.*

**Remark 16** (Decoding via acceptance regions). *Given a prefix $\pi \in \mathcal{V}^{t-1}$ and the observation $\widehat{\mathbf{h}}_t$ at position $t$, we identify the next token by checking in which acceptance region $\widehat{\mathbf{h}}_t$ lies: declare $v$ verified iff $\widehat{\mathbf{h}}_t \in \mathcal{A}_{\pi,t}(v; \varepsilon)$. By Lemma D.1, for any $\varepsilon < \Delta_{\pi,t}/2$ the regions $\{\mathcal{A}_{\pi,t}(v; \varepsilon)\}_{v \in \mathcal{V}}$ are pairwise disjoint; hence there is at most one verified token (and in the noiseless case $\varepsilon = 0$, exactly one).*

Building on the intuition in Remark 16, we introduce two radii to define acceptance regions that avoid collisions:

**Proposition D.1** (Probabilistic soundness and uniqueness of the local verifier). *Fix position $t$ and prefix $\pi \in \mathcal{V}^{t-1}$. Under Assumptions D.1 and D.2, for all $v^\star \in \mathcal{V}$, the following hold with probability one:*

1. ***Noiseless soundness.*** *If $\varepsilon = 0$ and $\widehat{\mathbf{h}}_t = F(v^\star \,;\, \pi, t)$, then $v^\star$ is the unique verified symbol.*

2. ***Robust uniqueness.*** *If $\varepsilon < \Delta_{\pi,t}/2$ and $\widehat{\mathbf{h}}_t \in \mathcal{A}_{\pi,t}(v^* \,;\, \varepsilon)$, then $v^\star$ is the unique verified symbol.*

*Proof.* Recall that under Assumptions D.1 and D.2, $F$ is injective and $\Delta_{\pi,t} > 0$ almost-surely.

*(1) Noiseless soundness.* For any $v \in \mathcal{V}$, $\mathcal{A}_{\pi,t}(v \,;\, 0) = \{F(v \,;\, \pi, t)\}$. If $\widehat{\mathbf{h}}_t = F(v^\star \,;\, \pi, t)$ and some $v \neq v^\star$ were also verified at $\varepsilon = 0$, we would have $F(v \,;\, \pi, t) = F(v^\star \,;\, \pi, t)$, which is a probability zero event under the assumptions made. Hence $v^\star$ is uniquely verified almost-surely.

*(2) Robust uniqueness.* Assume $\varepsilon < {}^{\Delta_{\pi,t}}/2$ and $\|\widehat{\mathbf{h}}_t - F(v^\star\,;\,\pi,t)\|_2 < \varepsilon$. If some $v \neq v^\star$ were also verified, then $\|\widehat{\mathbf{h}}_t - F(v\,;\,\pi,t)\|_2 \leq \varepsilon$. By the triangle inequality,

$$\left\|F(v\,;\,\pi,t) - F(v^\star\,;\,\pi,t)\right\|_2 \;\leq\; \left\|\widehat{\mathbf{h}}_t - F(v\,;\,\pi,t)\right\|_2 + \left\|\widehat{\mathbf{h}}_t - F(v^\star\,;\,\pi,t)\right\|_2 \;<\; 2\varepsilon \;<\; \Delta_{\pi,t},$$

contradicting the definition of $\Delta_{\pi,t}$ (again, valid under the assumptions made). Thus $v^\star$ is uniquely verified almost-surely. $\qquad\square$

**Candidate enumeration.** Finally, we introduce the last conceptual block required to build the inversion algorithm: a *policy algorithm* that systematically enumerates candidate tokens so that the verifier is guaranteed to encounter the true one.

**Definition D.3** (Policy algorithm). *Let $\mathcal{V}$ be a finite vocabulary. A* policy algorithm *is a (possibly randomized) map*

$$\Pi : \{\mathcal{C} \subsetneq \mathcal{V}\} \longrightarrow \mathcal{V} \qquad \text{such that} \qquad \Pi(\mathcal{C}) \in \mathcal{V} \setminus \mathcal{C} \ \text{for all } \mathcal{C} \subsetneq \mathcal{V}.$$

*(When $\mathcal{C} = \mathcal{V}$ the map is undefined.)*

**Remark 17** (Enumeration property). *Intuitively, a policy chooses any token not tried yet. Starting from $\mathcal{C}_0 = \varnothing$ and iterating*

$$v_i := \Pi(\mathcal{C}_{i-1}), \qquad \mathcal{C}_i := \mathcal{C}_{i-1} \cup \{v_i\} \quad (i = 1, \ldots, |\mathcal{V}|),$$

*produces a sequence $(v_1, \ldots, v_{|\mathcal{V}|})$ that is a (possibly random) permutation of $\mathcal{V}$. Thus, in exactly $|\mathcal{V}|$ steps, every token is output once with no repetitions.*

**Two examples of policy algorithms.** We give (i) a *uniform-random without replacement* policy and (ii) a *gradient-guided* policy.

---

**Algorithm 2** Policy (Random)

---

**Require:** Vocabulary $\mathcal{V}$; visited set $\mathcal{C}$; embedding matrix $\mathbf{E} \in \mathbb{R}^{|\mathcal{V}| \times d}$
**Ensure:** Next token ID and embedding
1: Sample a permutation $L = (v_1, \ldots, v_{|\mathcal{V}|})$ uniformly from $\mathcal{V}$
2: Define $\rho(v\,;\,\pi)$ as the rank of $v$ in $L$
3: $v^\star = \arg\min_{v \in \mathcal{V} \setminus C} \rho(v\,;\,\pi)$
4: **return** $v^\star$, $\mathbf{E}_{v^\star}$

---

**Algorithm 3** Policy (Gradient-based)

---

**Require:** Vocabulary $\mathcal{V}$; visited set $\mathcal{C}$; embedding matrix $\mathbf{E} \in \mathbb{R}^{|\mathcal{V}| \times d}$ ; prefix $\pi \in \mathcal{V}^{t-1}$; layer $\ell$;
    previous continuous embedding $\mathbf{e}^{(j-1)}$ ; step size $\gamma > 0$; gradient-based update rule $\mathcal{G}$
**Ensure:** Next token ID and embedding
1: $\mathbf{g} \leftarrow \nabla_{\mathbf{e}^{(j-1)}} \frac{1}{2} \left\| F\left(\mathbf{e}^{(j-1)}\,;\,\pi,t\right) - \widehat{\mathbf{h}}_t \right\|_2^2$
2: $\mathbf{e}^{(j)} \leftarrow \mathcal{G}(\mathbf{e}^{(j-1)}, \mathbf{g}, \gamma)$
3: Get $L = (v_1, \ldots, v_{|\mathcal{V}|})$ by ordering $v_i$ based on $\ell_2(\mathbf{E}_{v_i}, \mathbf{e}^{(j)})$
4: Define $\rho(v\,;\,\pi)$ as the rank of $v$ in $L$
5: $v^\star = \arg\min_{v \in \mathcal{V} \setminus C} \rho(v\,;\,\pi)$
6: **return** $v^\star$, $\mathbf{e}^{(j)}$

---

**Remark 18** (Bypassing the embedding layer). *We slightly overload notation and write $F(\mathbf{e}; \pi, t)$. Here we bypass the token embedding lookup and inject a continuous vector at the current position: the first $t{-}1$ rows of $\mathbf{H}^{(0)}$ are set to $\mathrm{Emb}(\pi)$ and the $t$-th row is set to $\mathbf{e}$. This extension is used only to guide the search (e.g., in **Policy-Gradient**). All theoretical guarantees are stated for $F(v; \pi, t)$ with $v \in \mathcal{V}$ and are unaffected by allowing $F$ to accept a continuous proxy during candidate scoring. Any extra inputs/side outputs used by a policy (such as the updated proxy) are orthogonal to the correctness statements.*

**Remark 19** (Practical choice of policy)**.** *Both Algorithms 2 and 3 satisfy Definition D.3. In practice we use the **gradient-guided** policy with standard gradient descent updates, as it tends to find the verified token with far fewer proposals: the next token is chosen by ranking $\mathcal{V}$ by the distance $\|\mathbf{E}_v - \mathbf{e}^{(j)}\|_2$ to the updated proxy $\mathbf{e}^{(j)}$. This preserves the same worst-case guarantees (single pass over $\mathcal{V}$) while improving empirical efficiency.*

### D.3 GLOBAL INVERSION VIA SIPIT

We now compose the local verifier into a sequential decoder. At step $t$, causality ensures $\mathbf{h}_t(\mathrm{s}) = F(\mathrm{s}_t; \pi, t)$ for the true prefix $\pi = \mathrm{s}_{1:t-1}$. Since the verifier uniquely accepts $\mathrm{s}_t$ (noiselessly, and robustly under perturbations below half the margin), any covering policy must encounter and accept the true token within a single pass over $\mathcal{V}$. Iterating from $t = 1$ to $T$ yields exact recovery almost surely; we also quantify robustness and the worst-case runtime.

We are now ready to introduce our inversion algorithm: SIPIT (Algorithm 1). The algorithms applies to decoder-only transformers with *causal* self-attention (Assumption D.1), and assumes injectivity, which occurs with almost-surely (Assumption D.2). We assume access to the layer-$\ell$ hidden states per position $\left\{\widehat{\mathbf{h}}_t\right\}_{t=1}^{T}$ and to the parameters needed to evaluate the local verifier from Definition D.2 for arbitrary $(t, \pi, j)$, as well as the gradient (when needed), namely to the model up to layer $\ell$. A policy algorithm is fixed (e.g., Algorithm 3).

We begin by recording the following standard lemma and omitting the proof, as it is immediate from causal masking: under causal self-attention, the representation at position $t$ is independent of future tokens.

**Lemma D.2** (Causal factorization and prefixwise identifiability)**.** *Under Assumptions D.1 and D.2, fix position $t \in [T]$. For any $\mathrm{s} = \langle \mathrm{s}_1, \ldots, \mathrm{s}_T \rangle$ with $\pi = \langle \mathrm{s}_1, \ldots, \mathrm{s}_{t-1} \rangle$,*

$$\mathbf{h}_t(\mathrm{s}) \;=\; F(\mathrm{s}_t \,;\, \pi, t),$$

*where $F$ is the one-step map from Definition D.1.*

*Proof.* With causal masking, position $t$ attends only to positions $\leq t$. Evaluating the network up to layer $\ell$ therefore yields a representation at $t$ that is a function of the prefix $\pi$ and the current token $\mathrm{s}_t$ only, i.e. $F(\mathrm{s}_t \,;\, \pi, t)$, as claimed. $\square$

**Proposition D.2** (The verifier is the right primitive)**.** *Fix $t$ and a true prefix $\pi = \langle \mathrm{s}_1, \ldots, \mathrm{s}_{t-1} \rangle$. Under Assumption D.1, the observed hidden state at step $t$ satisfies $\mathbf{h}_t(\mathrm{s}) = F(\mathrm{s}_t \,;\, \pi, t)$ (Lemma D.2). In addition, under Assumption D.2, $F$ is injective and has positive margin $\Delta_{\pi,t} > 0$ almost-surely (Theorem D.1 and lemma D.1). Consequently, for the local verifier of Definition D.2, the following hold with probability one:*

*1. (Noiseless) With $\varepsilon = 0$ and observation $\widehat{\mathbf{h}}_t = \mathbf{h}_t(\mathrm{s})$, the unique verified token is $\mathrm{s}_t$.*

*2. (Robust) If $\widehat{\mathbf{h}}_t = \mathbf{h}_t(\mathrm{s}) + \mathbf{e}_t$ with $\|\mathbf{e}_t\|_2 < \varepsilon < {}^{\Delta_{\pi,t}}\!/2$, then $\mathrm{s}_t$ is the unique verified token.*

*Proof.* Immediate from Lemma D.2 and Proposition D.1 applied with $v^{\star} = \mathrm{s}_t$, which holds almost-surely by Theorem D.1 and Lemma D.1. $\square$

**Proposition D.3** (Eventual acceptance under increasing enumeration)**.** *Fix a position $t$ and the true prefix $\pi = \langle \mathrm{s}_1, \ldots, \mathrm{s}_{t-1} \rangle$. Under Assumption D.1 and Assumption D.2, let $\varepsilon \geq 0$ and work on the probability-one event where the local verifier uniquely accepts the true token $\mathrm{s}_t$ (e.g., $\varepsilon = 0$ or $\varepsilon < \Delta_{\pi,t}/2$; see Proposition D.2).*

*Let $\Pi$ be any policy algorithm (Definition D.3). Define the increasing visited sets by $\mathcal{C}_0 = \varnothing$, $v_i := \Pi(\mathcal{C}_{i-1})$, and $\mathcal{C}_i := \mathcal{C}_{i-1} \cup \{v_i\}$ for $i \geq 1$, and stop at the first index*

$$\tau := \min \left\{ i \geq 1 : \; \widehat{\mathbf{h}}_t \in \mathcal{A}_{\pi,t}(v_i \,;\, \varepsilon) \right\}.$$

*Then $(v_i)_{i \geq 1}$ enumerates $\mathcal{V}$ without replacement and $\tau \leq |\mathcal{V}|$ almost surely. In particular, for the fixed prefix $\pi$, the policy's increasingly expanding search over $\mathcal{V}$ eventually proposes the unique verified token $\mathrm{s}_t$ and accepts it with probability 1.*

*Proof.* Work on the probability-one event of Proposition D.2 (under Assumptions D.1 and D.2 with the stated $\varepsilon$), on which the local verifier at step $t$ uniquely accepts the true token $s_t$. Equivalently,

$$\widehat{\mathbf{h}}_t \in \mathcal{A}_{\pi,t}(v\,;\,\varepsilon) \iff v = s_t. \tag{35}$$

**Enumeration without replacement.** By the definition of a policy algorithm (Definition D.3), $v_i = \Pi(\mathcal{C}_{i-1}) \in \mathcal{V} \setminus \mathcal{C}_{i-1}$ and $\mathcal{C}_i = \mathcal{C}_{i-1} \cup \{v_i\}$. Hence $v_i \notin \mathcal{C}_{i-1}$ and $|\mathcal{C}_i| = |\mathcal{C}_{i-1}| + 1$. Inducting on $i$ yields that $(v_i)_{i \geq 1}$ has no repetitions and $\mathcal{C}_i$ contains exactly $i$ distinct tokens. Since $\mathcal{V}$ is finite, after $|\mathcal{V}|$ steps we have $\mathcal{C}_{|\mathcal{V}|} = \mathcal{V}$, i.e., $(v_i)_{i=1}^{|\mathcal{V}|}$ is a permutation of $\mathcal{V}$ (this holds pathwise, for any realization of the policy's internal randomness).

**Eventual acceptance.** Because $(v_i)$ is a permutation of $\mathcal{V}$, there exists a unique index $j \in \{1, \ldots, |\mathcal{V}|\}$ with $v_j = s_t$. By equation 35,

$$\tau = \min\{\,i \geq 1 : \widehat{\mathbf{h}}_t \in \mathcal{A}_{\pi,t}(v_i\,;\,\varepsilon)\,\} = \min\{\,i \geq 1 : v_i = s_t\,\} = j,$$

so $\tau \leq |\mathcal{V}|$ and the process accepts $s_t$.

Since the event on which equation 35 holds has probability 1, the conclusion (eventual acceptance at finite $\tau$) holds almost surely. □

**Theorem D.2** (Correctness of SIPIT (noiseless & robust)). *For each $t \in \{1, \ldots, T\}$ let $\pi_t = \langle s_1, \ldots, s_{t-1} \rangle$ and let $\Delta_{\pi_t,t} > 0$ be the margin of the one-step map $F(\cdot\,;\,\pi_t, t)$ from Lemma D.1. Under Assumptions D.1 and D.2, run SIPIT (Algorithm 1) with a tolerance $\varepsilon \geq 0$ and observations*

$$\widehat{\mathbf{h}}_t = \mathbf{h}_t(s) + \mathbf{e}_t \qquad (t = 1, \ldots, T),$$

*where the perturbations satisfy $\|\mathbf{e}_t\|_2 \leq \varepsilon$ for all $t$ and*

$$\varepsilon < \tfrac{1}{2}\Delta_{\pi_t,t} \qquad \text{for all } t.$$

*Then, with probability 1 over the model parameters: (i) for every $t$, the inner for-loop over $j$ (the loop over vocabulary candidates) terminates within $|\mathcal{V}|$ iterations by accepting the true token $s_t$; and (ii) after the outer for-loop over $t$ (the loop over positions) finishes, the algorithm outputs the exact sequence $\widehat{s} = s$.*

*In particular, this covers the noiseless case by taking $\varepsilon = 0$ and $\widehat{\mathbf{h}}_t = \mathbf{h}_t(s)$, and the robust case with any uniform $\varepsilon$ such that $\max_t \|\mathbf{e}_t\|_2 \leq \varepsilon < \tfrac{1}{2}\min_t \Delta_{\pi_t,t}$.*

*Proof.* By Assumption D.2 and theorem D.1, and Lemma D.1, there is a probability-one event on which, for all $t$, $F(\cdot\,;\,\pi_t, t)$ is injective with strictly positive margin $\Delta_{\pi_t,t}$. Intersecting across finitely many $t$ preserves probability 1. Work on this event.

By Assumption D.1 and lemma D.2, $\mathbf{h}_t(s) = F(s_t; \pi_t, t)$. Since $\|\mathbf{e}_t\|_2 \leq \varepsilon$,

$$\widehat{\mathbf{h}}_t = F(s_t; \pi_t, t) + \mathbf{e}_t \in \overline{B}\big(F(s_t; \pi_t, t), \varepsilon\big) = \mathcal{A}_{\pi_t,t}(s_t; \varepsilon),$$

so the local verifier *accepts* $s_t$. Moreover, because $\varepsilon < \tfrac{1}{2}\Delta_{\pi_t,t}$, Proposition D.1(2) implies *robust uniqueness*:

$$\widehat{\mathbf{h}}_t \in \mathcal{A}_{\pi_t,t}(v; \varepsilon) \iff v = s_t. \tag{36}$$

When $\varepsilon = 0$, equation 36 also holds by Proposition D.1(1). We now analyze SIPIT and proceed by induction on $t$.

*Base case ($t = 1$).* The *outer for-loop over $t$* begins with $\widehat{s} = \langle\rangle = \pi_1$. Inside the *inner for-loop over $j$* (the loop over vocabulary candidates), the policy (Definition D.3) enumerates $\mathcal{V}$ without replacement. By Proposition D.3, there exists $j^\star \leq |\mathcal{V}|$ such that $v_{j^\star} = s_1$, which is accepted and triggers the **break**; the algorithm appends $s_1$.

*Inductive step.* Suppose after completing the inner loop at step $t - 1$ the algorithm has appended $s_{t-1}$, so the prefix entering step $t$ is $\widehat{s} = \pi_t$. By equation 36, within the inner loop the verifier accepts exactly when $v_j = s_t$. Because the policy enumerates $\mathcal{V}$ without replacement, some $j \leq |\mathcal{V}|$ satisfies $v_j = s_t$, which is accepted, appended, and the inner loop **breaks**.

Thus for every $t$, the inner loop terminates by accepting $s_t$ within $|\mathcal{V}|$ iterations, and after the outer loop finishes we have appended $(s_1, \ldots, s_T)$, i.e., $\widehat{s} = s$. Since the reasoning holds on a probability-one event (independent of the policy's internal randomness), the conclusion is almost sure. □

**Termination and complexity.** Having established correctness, we record the worst-case iteration count and discuss its relation to wall-clock time.

**Proposition D.4** (Termination and linear step bound). *Run SIPIT (Algorithm 1) on a length-T sequence with any policy that enumerates $\mathcal{V}$ without replacement. Then the algorithm halts after a finite number of iterations. Moreover, in the worst case the* inner for-loop over $j$ executes at most $|\mathcal{V}|$ iterations at each position $t$, so the total number of verifier tests across the entire run is at most $T|\mathcal{V}|$. In particular, the number of loop iterations grows linearly with $T \cdot |\mathcal{V}|$.*

*Proof.* Fix a position $t$. The *inner for-loop over $j$* proposes unvisited tokens and stops when a candidate verifies, or after exhausting $\mathcal{V}$. Because the policy enumerates without replacement, the loop can execute at most $|\mathcal{V}|$ iterations at step $t$. The *outer for-loop over $t$* runs for exactly $T$ positions, hence the total number of inner-loop iterations (i.e., verifier tests) is at most $\sum_{t=1}^{T} |\mathcal{V}| = T|\mathcal{V}| < \infty$. Therefore the algorithm halts and the total number of tests is linear in $T \cdot |\mathcal{V}|$. □

**Remark 20** (Iterations vs. wall–clock time). *Proposition D.4 bounds the number of iterations/tests: the inner loop performs at most $|\mathcal{V}|$ verifier tests per position, so the total is $\Theta(T|\mathcal{V}|)$. This is an iteration complexity statement that holds for any policy satisfying the "enumerate $\mathcal{V}$ without replacement" property. Actual wall–clock time also depends on the per–test cost (one call to $F(v; \pi, t)$ plus a distance) and on any policy overhead (e.g., forward/backward proxy updates, scoring, sorting). A generic decomposition is*

$$time = \Theta(T|\mathcal{V}| \cdot C_{test}) + \sum_{t=1}^{T} C_{policy}(t),$$

*where $C_{test}$ is the cost of one membership test and $C_{policy}(t)$ captures policy-specific work at step $t$. Thus, if $|\mathcal{V}|$ is treated as fixed and $C_{test}$, $C_{policy}(t)$ are bounded (e.g., a constant number of proxy updates and at most one ranking per update), wall–clock time is $O(T)$. If $|\mathcal{V}|$ grows or the policy sorts per update, additional factors like $|\mathcal{V}|$ or $\log |\mathcal{V}|$ may appear in the time, but the termination and the $\Theta(T|\mathcal{V}|)$ iteration bound remain unchanged.*

**Remark 21** (Choosing the tolerance $\varepsilon$). *Theory guarantees uniqueness whenever $\varepsilon < \frac{1}{2}\Delta_{\pi,t}$ (Proposition D.1). Since $\Delta_{\pi,t}$ is unknown, two practical choices work well: (i)* backoff*: start with a small $\varepsilon$ and increase only if no token verifies; (ii)* calibration*: set $\varepsilon$ from held-out hidden states at layer $\ell$. In all cases the decision rule remains a simple yes/no membership test.*

**Remark 22** (Why SIPIT is sequential). *The algorithm never solves a global assignment. At position $t$ it conditions on the current prefix $\pi$ and queries the local verifier for a single token. Causality (Assumption D.1) ensures $\mathbf{h}_t$ depends only on $(\pi, s_t)$, so these local, prefixwise decisions compose to recover the full sequence.*

# E IMPLEMENTATION DETAILS AND ADDITIONAL EXPERIMENTS

This section collects implementation details (§E.1), additional ablations on collision experiments and SIPIT (§E.2), controlled experiments with identical next-token predictions (§E.3), qualitative inspection of the closest hidden-state pairs (§E.4), and connections to anisotropy and intrinsic dimension (§E.5).

## E.1 IMPLEMENTATION DETAILS

**What is a collision in practice.** In the theoretical parts of the paper we use "collision" in the usual functional sense: two distinct prompts $s \neq s'$ such that their last-token representations coincide exactly,

$$\mathbf{r}(s; \boldsymbol{\theta}_T) = \mathbf{r}(s'; \boldsymbol{\theta}_T).$$

This is the event whose probability is controlled in theorems 2.2 and 2.3 and in Section C, and all proofs are carried out at the level of exact equality (no numerical threshold is required).

In the experiments, however, representations are stored in floating-point format, so exact equality of $\mathbf{r}(s; \boldsymbol{\theta}_T)$ and $\mathbf{r}(s'; \boldsymbol{\theta}_T)$ may not be a meaningful or robust criterion. We therefore adopt a numerical proxy: given two prompts $s, s'$ and their embeddings $\mathbf{r}(s; \boldsymbol{\theta}_T), \mathbf{r}(s'; \boldsymbol{\theta}_T) \in \mathbb{R}^d$, we declare a

*practical collision* only if

$$\texttt{torch.allclose}\big(\mathbf{r}(\mathrm{s}\,;\boldsymbol{\theta}_T),\mathbf{r}(\mathrm{s}'\,;\boldsymbol{\theta}_T)\big) = \texttt{True},$$

i.e., every coordinate falls within PyTorch's prescribed tolerances, namely $10^{-5}$ and $10^{-8}$ for relevant and absolute tolerance respectively. Across all of the billions to trillions of empirical checks, every pair of distinct prompts $\mathrm{s} \neq \mathrm{s}'$ failed this criterion: `torch.allclose` always returned `False`, and the observed $\ell_2$ distances were consistently bounded away from zero. Thus, at the resolution of our numerical precision, we did not observe any collisions in practice.

**SIPIT implementation.** We implement SIPIT exactly as in Alg. 1 with the gradient-guided policy. To stabilize the continuous proxy used for ranking, we apply gradient clipping (capping the gradient norm at 1) and we periodically project it back to the nearest token embedding every $K = 50$ candidate proposals:

$$\mathbf{e}^{(j)} \leftarrow \mathbf{E}_{v^\dagger}, \qquad v^\dagger = \arg\min_{v \in \mathcal{V}\backslash\mathcal{C}}\big\|\mathbf{E}_v - \mathbf{e}^{(j)}\big\|_2,$$

without taking gradients through this projection. These heuristics affect efficiency only; the verifier and all correctness guarantees remain unchanged.

**HARDPROMPTS implementation.** The original HARDPROMPTS method Wen et al. (2023) targets multimodal vision-language models and optimizes prompts via a CLIP-based similarity objective. In our text-only setting we lack the vision branch and CLIP loss, so we adapt Algorithm 1 of Wen et al. (2023) to language models by replacing the objective with the same $\ell_2$ loss used in SIPIT's gradient calculation, and setting the optimization steps $T = \frac{1}{4}\#\,\text{tokens}\cdot|\mathcal{V}|$. All other details (step sizes, stopping rules) mirror our SIPIT setup to ensure a fair comparison.

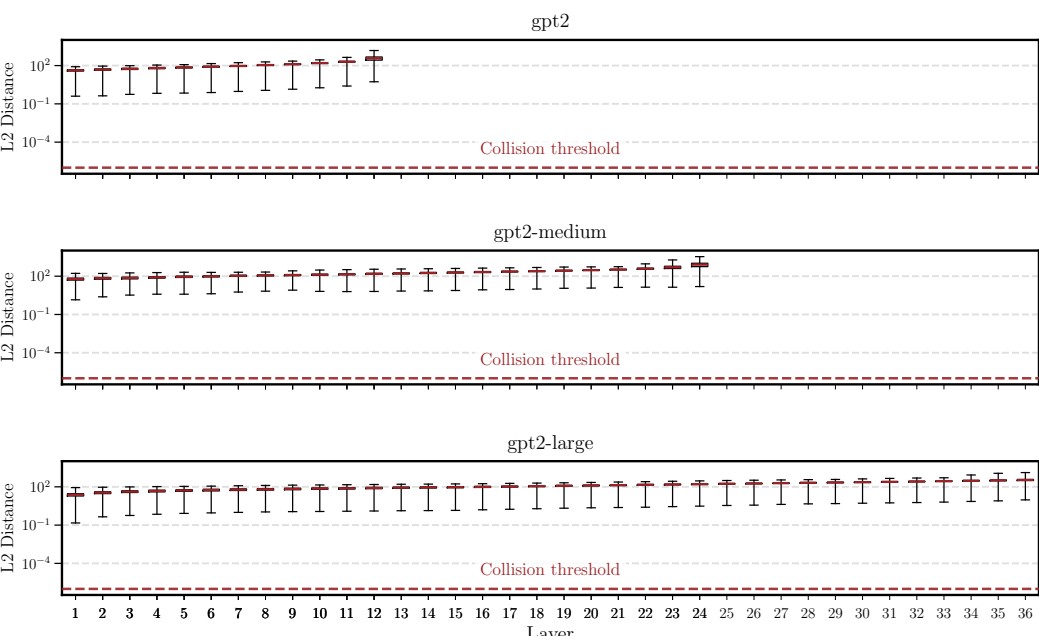

Figure 7: Seeking collisions in a large-scale prompt set (§4.1). For each layer, boxplots show the distribution (log scale) of the *minimum pairwise* $\ell_2$ distances between last-token states across prompts for the `GPT-2` model family (`Small`, `Medium`, and `Large`); red bars mark medians and the dashed line indicates the collision threshold $10^{-6}$.

### E.2 ADDITIONAL ABLATIONS

#### E.2.1 COLLISION EXPERIMENTS

We report three complementary ablations that probe how separation behaves across depth, length, and model family.

**GPT-2 family across depth.** For `GPT-2 Small`, `GPT-2 Medium`, and `GPT-2 Large`, the per-layer boxplots (log scale) of the *minimum pairwise* $\ell_2$ distances between last-token states in Figure 7 show that all minima sit orders of magnitude above the collision threshold $10^{-6}$ at every depth, and the typical separation *increases with depth* (median red bars drift upward). This rules out collisions in practice and indicates that deeper blocks monotonically sharpen last-token distinctions in these models.

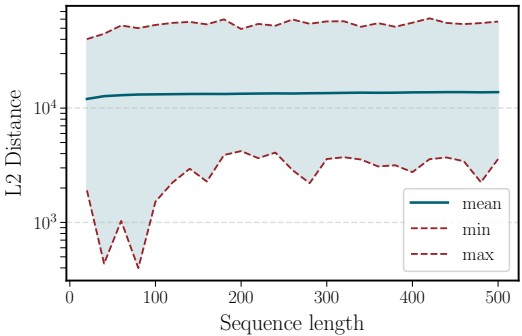

Figure 9: Sequence length versus distance over all pairs of distinct prompts for `Gemma-1B`.

**Gemma-3 family across depth and scale.** Across `Gemma3-1B`, `Gemma3-4B`, and `Gemma3-12B`, the layerwise boxplots (log scale) in Figure 8 again show minima far above $10^{-6}$ at all depths. Both depth *and* model size trend positively with separation: medians and lower whiskers move upward in deeper layers and larger models, indicating progressively stronger margins and no observed collisions.

**Effect of sequence length (Gemma-1B).** Varying the prompt length reveals that min/mean/max pairwise distances rise quickly for short sequences and then plateau, with the minimum never approaching zero (see Figure 9). This suggests that beyond a modest context size, additional tokens do not erode separability; margins stabilize rather than collapse, making collisions unlikely for any prompt length explored.

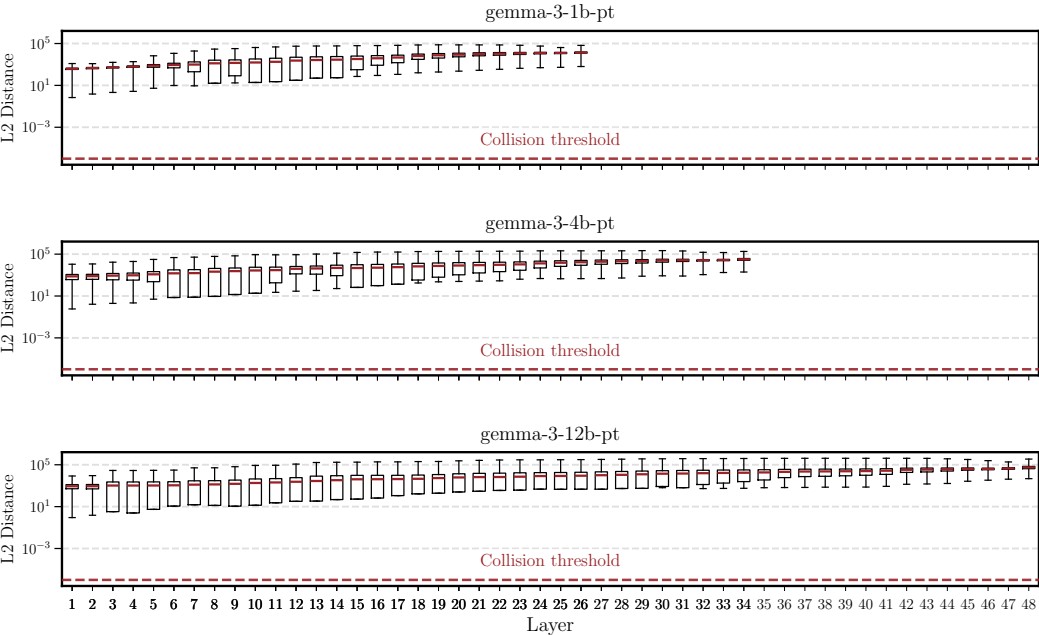

Figure 8: Seeking collisions in a large-scale prompt set (§4.1). For each layer, boxplots (log scale) show the distribution of *minimum pairwise* $\ell_2$ distances between last-token states across prompts for the `Gemma-3` model family (1B, 4B, 12B); red bars denote medians and the dashed line marks the collision threshold $10^{-6}$.

Overall, these ablations corroborate the main text: last-token states remain well-separated across architectures and depths, separation typically grows with depth (and scale for Gemma), and margins stabilize with sequence length, aligning with our almost-sure injectivity guarantees and with SIPIT's exact recovery behavior.

### E.2.2  SIPIT

| Model | Vocab size | Inversion performance | | |
|---|---|---|---|---|
| | | Accuracy | Time (s) | Vocab explored (%) |
| Mistral-7B-v0.1 | 32000 | 100% | $72.99 \pm 37.57$ | $0.21 \pm 0.11$ % |
| Llama-3.1-8B | 128255 | 100% | $345.35 \pm 181.30$ | $0.22 \pm 0.12$ % |

Table 6: Performance of SIPIT on different vocabulary sizes

**Vocabulary Size.**  To further validate our findings (as presented in Section 4) regarding the scaling of SIPIT with vocabulary size, we conducted additional experiments on the two models with substantially different vocabulary sizes, Mistral-7B-v0.1 ($\approx 32K$ vocabulary) and Llama-3.1-8B ($\approx 128K$). For a fair comparison, we construct sentences that tokenize to exactly the same sequence of tokens across both models. The results are reported in the Table 6. We observe that, in practice, the inversion time grows linearly with vocabulary size, as expected, reflected by the nearly constant percentage of tokens explored between the small-vocabulary model (Mistral) and the larger-vocabulary model (Llama). Importantly, for both models, the fraction of tokens explored remains below $0.25\%$, indicating that the gradient-based heuristic is both robust and highly efficient.

| Dataset | Inversion Time (s) | Accuracy |
|---|---|---|
| Train Data | $146.48 \pm 91.52$ | 100% |
| Test Data | $128.62 \pm 83.40$ | 100% |
| OOD | $106.87 \pm 39.10$ | 100% |

Table 7: Performance of SIPIT on in-distribution vs. out-of-distribution data

**Robustness of SIPIT on unseen and random sequences.**  Based on GPT-2, we constructed three datasets, which we refer to as **Train**, **Test**, and **OOD** (Out-of-Distribution). The **Train** set is formed by sampling sentences from WebText (the dataset used to train GPT-2 Radford et al. (2019)); the **Test** set contains sentences sampled from Wikipedia (not in the training set); and the **OOD** set consists of random token sequences. Each dataset contains 50 prompts of length 100 tokens. We report the findings in Table 7. Interestingly, the **OOD** samples are significantly faster to invert than the **Train** and **Test** samples. We hypothesize that this difference stems from the geometry of the hidden representations: natural language sentences (**Train** and **Test**) tend to lie on a structured, clustered manifold, which can make the inversion landscape locally flatter and less well-conditioned. In contrast, random token sequences produce more dispersed and isolated hidden states, yielding clearer descent directions and effectively stronger gradient signals, which accelerates convergence. Across all three datasets, we obtain exact recovery for every sequence, further supporting the theoretical guarantees of SIPIT.

### E.3  IDENTICAL NEXT-TOKEN

The collision and ablation experiments above use generic prompt sets. A natural stress test is to ask: what happens when we *deliberately* construct prompt pairs that produce the same next-token prediction? To answer this question we designed a set of new experiments where two different prompt are specifically constructed to yield the exact same target answer. First, we focused on word-to-word machine translation (google/smol) and math tasks (ProCreations/SimpleMath)

| Model | $\ell_2$ **Distance (min)** | | |
|---|---|---|---|
| | **layer 1** | **layer $L/2$** | **layer $L$** |
| Llama-3.1-8B | 0.694 | 1.632 | 4.202 |
| Mistral-7B-v0.1 | 0.207 | 1.056 | 2.348 |
| Phi-4-mini-instruct | 4.375 | 6.974 | 17.328 |

Table 8: Distances for Translation (En–Fr) separator-token embeddings across layers.

| Model | $\ell_2$ **Distance (min)** | | |
|---|---|---|---|
| | **layer 1** | **layer $L/2$** | **layer $L$** |
| Llama-3.1-8B | 0.789 | 2.126 | 8.245 |
| Mistral-7B-v0.1 | 0.222 | 1.664 | 4.362 |
| Phi-4-mini-instruct | 4.447 | 8.497 | 37.262 |

Table 9: Distances for Math separator-token embeddings across layers.

on `Llama-3.1-8B`, `Mistral-7B`, and `Phi-4-mini-instruct`. From these datasets, we built few shot prompts that differed only in their delimiters (e.g. `->` vs `:`) while preserving identical translations or arithmetic solutions. Some qualitative examples are shown below:

```
Delimiter: ->

Translate into French.

Hello -> Bonjour
Goodbye -> Au revoir
House ->
```

```
Delimiter: :

Translate into French.

Hello : Bonjour
Goodbye : Au revoir
House :
```

```
Delimiter: ->

Do the additions.

2790 + 6698 -> 9488
8262 + 3848 -> 12110
1628 + 132 ->
```

```
Delimiter: =

Do the additions.

2790 + 6698 = 9488
8262 + 3848 = 12110
1628 + 132 =
```

We then assessed collisions involving four different separator token embeddings across all dataset pairs, specifically `->`, `:`, `=`, and `-`. Despite producing the exact same answer the corresponding embeddings remain clearly distinct (no "collision") since the minimum $\ell_2$ distance is well above the collision threshold over the $\approx 140K$ possible pairs, as seen in tables 8 and 9.

Furthermore, we constructed a dataset of random prefixes sampled from internet text, each followed by the fixed suffix "`Complete this: The quick brown fox jumps over the lazy`". To build the dataset, we sampled 10K prefix sequences of length 50 tokens from Wikipedia and appended the tokenized suffix to each. The minimum $\ell_2$ distances obtained are reported in Table 10. Even in this setting, although the next token prediction is exactly "dog", all last-token embeddings remain far above the tolerance threshold.

| Model | $\ell_2$ **Distance (min)** | | |
|---|---|---|---|
| | **layer 1** | **layer $L/2$** | **layer $L$** |
| Mistral-7B-v0.1 | 0.012 | 0.265 | 3.494 |
| Llama-3.1-8B | 0.046 | 0.733 | 6.227 |
| Phi-4-mini-instruct | 0.087 | 2.302 | 18.913 |

Table 10: Distances for random-prefix dataset with fixed "quick brown fox" suffix.

### E.4 PROMPTS WITH SIMILAR REPRESENTATIONS

To complement the quantitative injectivity results in the main text, we inspected qualitative examples of sequences whose last-token hidden states are among the closest we observed. For a given model,

we computed the Euclidean distance between last-layer representations $h_L(s)$ and $h_L(t)$ of the final token in two sequences $s$ and $t$, and manually examined pairs with the smallest $\ell_2$ distances.

For both `Llama-3.1-8B` and `Mistral-7B-v0.1`, the closest pairs correspond to Python code snippets that are almost identical, typically differing only by a small shift such as one or more trailing newline tokens. In most of the close pairs we examined, the two sequences satisfy

$$s \; = \; t \circ \langle \text{new line token} \rangle^k$$

for some small $k \geq 1$. Even in these extremal cases, however, the last-token representations remain clearly separated in $\ell_2$ distance.

**Llama-3.1-8B.**  One of the closest pairs we found for `Llama-3.1-8B` is shown below. The only difference between the two sequences is the presence of several trailing newline characters at the end of the second snippet. The last-token $\ell_2$ distance at the final layer for this pair is 1.274, which is small relative to typical distances but still far from zero, and thus consistent with the absence of collisions observed in our exhaustive tests.

```
Llama-3.1-8B: Sequence s

...
# Theme options are theme-specific and customize the ...
#html_theme_options = {}

# Add any paths that contain custom themes here ...
#html
```

```
Llama-3.1-8B: Sequence t

...
# Theme options are theme-specific and customize the ...
#html_theme_options = {}

# Add any paths that contain custom themes here ...
#html
\n
\n
\n
```

**Mistral-7B-v0.1.**  A similar phenomenon occurs for `Mistral-7B-v0.1`. Again, one of the closest pairs consists of two almost identical code snippets, where the second sequence appends a single newline token: For this pair, the last-token $\ell_2$ distance at the last layer is 1.146. As in the Llama example, the nearest neighbors arise from almost identical contexts differing only in trailing whitespace tokens, and even these extremal cases exhibit a non-negligible separation in representation space.

```
Mistral-7B-v0.1: Sequence s

...
# The reST default role to use for all documents.
#default_role = None

# If true, '()' will be appended to :func: ...
#add_function_parentheses = True

# If true, the current module ...
```

```
Mistral-7B-v0.1: Sequence t

# The reST default role to use for all documents.
#default_role = None

# If true, '()' will be appended to :func: ...
#add_function_parentheses = True

# If true, the current module ...
\n
```

**Summary.** Across all models and pairs we inspected, we did not observe qualitatively different prompts whose last-layer, last-token embeddings were comparably close. Instead, the nearest neighbors consistently involved near-duplicate snippets (often code or documentation) differing only by whitespace or other minor formatting tokens. These qualitative observations align with the injectivity margins reported in the main text and support the view that small perturbations in formatting do not lead to collisions in the representations used by SIPIT.

### E.5 RELATION WITH ANISOTROPY AND INTRINSIC DIMENSION

As part of our broader investigation, we also examined connections to the analyses presented in the works of Razzhigaev et al. (2025) (LLM-Microscope) and Razzhigaev et al. (2024), and ran a targeted experiment in this spirit.

**Experimental setup.** We performed a proof-of-concept study using GPT-2 Small. We sampled 100 natural-language prompts of fixed length $K$ and, for each prompt, generated 1000 single-token continuations by appending each token from a fixed vocabulary subset of size 1000. For every layer $\ell$, we extracted the hidden representation of the last token for all 1000 continuations, producing a $1000 \times d$ matrix for each (layer, prompt) pair. On each matrix we computed (i) anisotropy and intrinsic dimension as in LLM-Microscope, and (ii) simple "injectivity margin" statistics: the minimum pairwise Euclidean distance between continuation embeddings, averaged over prompts. Aggregating over the 100 prompts yields, for each layer, a triple consisting of anisotropy, intrinsic dimension, and injectivity margin.

**Experiment 1: anisotropy vs. injectivity margin.** Across layers, we correlated mean anisotropy with the mean injectivity margin. The resulting Pearson correlation is **0.72**, and the Spearman corre-

| Layer | Anisotropy (mean) | ID (mean) | Margin (min) |
|-------|-------------------|-----------|--------------|
| 1     | 0.089579          | 20.754620 | 1.850306     |
| 2     | 0.076049          | 17.565538 | 1.956753     |
| 3     | 0.071429          | 16.765265 | 2.064488     |
| 4     | 0.075067          | 16.679382 | 2.241199     |
| 5     | 0.083282          | 17.183246 | 2.382355     |
| 6     | 0.089542          | 17.697870 | 2.499817     |
| 7     | 0.088463          | 17.018419 | 2.704958     |
| 8     | 0.083261          | 16.296431 | 2.886434     |
| 9     | 0.081803          | 16.040713 | 3.025268     |
| 10    | 0.083083          | 15.730601 | 3.330774     |
| 11    | 0.090206          | 15.635035 | 3.918343     |
| 12    | 0.288352          | 16.434897 | 4.640457     |

Table 11: Layer-wise anisotropy, intrinsic dimension, and injectivity margin.

lation is **0.45**. In this setting, layers with higher anisotropy tend to exhibit larger injectivity margins: continuation clouds become both more structured (anisotropic) and farther from collisions. This suggests that anisotropy is compatible with, and may even reinforce, numerically robust injectivity.

**Experiment 2: intrinsic dimension vs. injectivity margin.** Repeating the analysis with intrinsic dimension, we observe a Pearson correlation of **-0.60** and a Spearman correlation of **-0.79** between intrinsic dimension and injectivity margin. Thus, layers with lower intrinsic dimensionality tend to have larger margins: compressed-looking manifolds are, if anything, more separated. This aligns with our theorem that injectivity rules out information-destroying collapses.

**Discussion.** This line of analysis is highly complementary to our injectivity framework. Whereas our results establish that internal representations are almost surely lossless, LLM-Microscope offers fine-grained geometric diagnostics of how these representations evolve across depth and training. Particularly notable is the observation that anisotropy and intrinsic dimension follow a reverse-U profile: representations become more anisotropic and lower-dimensional in intermediate layers, then partially re-expand near the output, offering a concrete geometric picture of how structure is carved into aligned directions and low-dimensional manifolds.

This is especially relevant given that our paper challenges classic accounts of learning via bottleneck compression (e.g. Shwartz-Ziv & Tishby (2017)). If information is preserved along the residual stream, learning cannot proceed layer by layer purely through compression. Our preliminary experiments suggest a different picture: as depth increases, margins grow, intrinsic dimension decreases, and anisotropy follows a concave trajectory with a late spike. Early layers expand and reorganize, intermediate layers carve information into low-dimensional directional manifolds, and upper layers sharpen this structure. Overall, this is consistent with a network that preserves injectivity while funneling information into increasingly structured, well-separated representations.

| Model (HF example) | Activation in FFN | Real-analytic? |
|---|---|---|
| Llama-2 | SwiGLU | Yes |
| Llama-3 | SwiGLU | Yes |
| Mistral-7B-v0.1 | SiLU | Yes |
| Mixtral-8x7B-v0.1 | SiLU | Yes |
| Gemma | GeGLU | Yes |
| Gemma-2 | GELU | Yes |
| Qwen2MoE | SwiGLU | Yes |
| Qwen-2 | SiLU | Yes |
| Qwen3MoE | SiLU | Yes |
| Qwen-3 | SiLU | Yes |
| Phi | GELU | Yes |
| Phi-3 | SiLU | Yes |
| GPT-2 | GELU | Yes |
| GPT-J | GELU | Yes |
| GptOss | SiLU | Yes |
| Grok-1 | GELU | Yes |
| DeepSeek-V2 | SiLU | Yes |
| DeepSeek-V3 | SiLU | Yes |

Table 12: Activation functions used in the feed-forward networks of representative modern LLMs.

## F  REAL-ANALYTIC ACTIVATION FUNCTIONS IN MODERN LLMS

A natural question raised by our analysis is to what extent modern large language models actually use real-analytic activation functions in their feed-forward networks. Since our results apply most directly when the non-linearities are real-analytic, it is important to check whether this assumption holds in practice.

To get a concrete picture, we surveyed a set of widely used open-source and proprietary-style architectures and recorded the activation function used in their feed-forward blocks. The models and their reported activations are summarized in Table 12. For each model, we also indicate whether the activation is real-analytic. Activations such as SiLU/Swish, SwiGLU, GeGLU, and GELU are all real-analytic, being compositions and products of elementary analytic functions (e.g., linear maps, exponentials, and the error function).

Across this representative sample, we find that *all* models (18 out of 18) use real-analytic activations in their feed-forward blocks. In other words, the analyticity assumption is not merely a technical convenience but accurately reflects common design practice. This supports the relevance of our theoretical results for real-world large language models: the vast majority of modern transformers already operate in a regime where the non-linearities are real-analytic, and hence fall directly within the scope of our analysis. We now formally prove that SiLU and GELU are real-analytic scalar functions, and that the corresponding gated constructions SwiGLU and GeGLU define real-analytic vector-valued maps. The proofs build from elementary ingredients upward: sigmoid (Proposition F.1) → SiLU (Proposition F.2); error function (Proposition F.3) → GELU (Proposition F.4); then coordinatewise lifting (Proposition F.5) and GLU gating (Proposition F.6).

**Proposition F.1** (Logistic sigmoid is real-analytic). *The logistic sigmoid*

$$\sigma(x) \;:=\; \frac{1}{1+e^{-x}}, \qquad x \in \mathbb{R},$$

*is real-analytic on $\mathbb{R}$.*

*Proof.* By Proposition A.5, the map $x \mapsto e^{-x}$ is real-analytic on $\mathbb{R}$. By Proposition A.1, the sum $x \mapsto 1 + e^{-x}$ is real-analytic; moreover $1 + e^{-x} > 0$ for all $x \in \mathbb{R}$, so it never vanishes. By the quotient rule in Proposition A.1, the reciprocal

$$x \mapsto \frac{1}{1+e^{-x}}$$

is therefore real-analytic on $\mathbb{R}$. □

**Proposition F.2** (SiLU / Swish is real-analytic). *The SiLU (or Swish) activation*

$$\mathrm{SiLU}(x) \;:=\; x\,\sigma(x) \;=\; \frac{x}{1+e^{-x}}, \qquad x \in \mathbb{R},$$

*is real-analytic on $\mathbb{R}$.*

*Proof.* The identity map $x \mapsto x$ is a polynomial, hence real-analytic by Proposition A.4. By Proposition F.1, $\sigma$ is real-analytic. The product of two real-analytic functions is real-analytic by Proposition A.1, so $x \mapsto x\,\sigma(x)$ is real-analytic on $\mathbb{R}$. □

**Proposition F.3** (Error function is real-analytic). *The error function*

$$\mathrm{erf}(x) \;:=\; \frac{2}{\sqrt{\pi}} \int_0^x e^{-t^2}\, dt, \qquad x \in \mathbb{R},$$

*is real-analytic on $\mathbb{R}$.*

*Proof.* By Proposition A.5, exp is real-analytic on $\mathbb{R}$ with power series $e^z = \sum_{k=0}^{\infty} \frac{z^k}{k!}$ and infinite radius of convergence. Substituting $z = -t^2$ yields

$$e^{-t^2} = \sum_{k=0}^{\infty} \frac{(-1)^k}{k!} t^{2k}, \qquad t \in \mathbb{R}.$$

This series has infinite radius of convergence, so it converges uniformly on every bounded interval. By standard results on termwise integration of power series (e.g. Rudin 1976), we may integrate termwise:

$$\int_0^x e^{-t^2}\, dt = \sum_{k=0}^\infty \frac{(-1)^k}{k!} \int_0^x t^{2k}\, dt = \sum_{k=0}^\infty \frac{(-1)^k}{k!(2k+1)}\, x^{2k+1}.$$

Multiplying by $2/\sqrt{\pi}$ we obtain

$$\mathrm{erf}(x) = \frac{2}{\sqrt{\pi}} \sum_{k=0}^\infty \frac{(-1)^k}{k!(2k+1)}\, x^{2k+1},$$

a power series with infinite radius of convergence. Hence $\mathrm{erf}$ is real-analytic on $\mathbb{R}$ by Definition A.1. $\qquad\square$

**Proposition F.4** (GELU is real-analytic)**.** *Let*

$$\Phi(x) \;:=\; \frac{1}{2}\Big(1 + \mathrm{erf}\big(\tfrac{x}{\sqrt{2}}\big)\Big)$$

*be the CDF of a standard normal random variable. The (exact) GELU activation*

$$\mathrm{GELU}(x) \;:=\; x\,\Phi(x)$$

*is real-analytic on $\mathbb{R}$.*

*Proof.* By Proposition F.3, $\mathrm{erf}$ is real-analytic. The map $x \mapsto \frac{x}{\sqrt{2}}$ is linear, hence real-analytic; by Proposition A.2, the composition $x \mapsto \mathrm{erf}\big(\frac{x}{\sqrt{2}}\big)$ is real-analytic. Adding the constant $1$ and scaling by $\frac{1}{2}$ preserves real-analyticity by Proposition A.1, so $\Phi$ is real-analytic. The identity map $x \mapsto x$ is a polynomial (Proposition A.4), hence real-analytic; their product $x \mapsto x\,\Phi(x)$ is therefore real-analytic by Proposition A.1. $\qquad\square$

Having established that SiLU and GELU are real-analytic as scalar functions, we now lift them to the vector-valued setting and show that the GLU-style gating used in modern architectures preserves real-analyticity.

**Proposition F.5** (Vector-valued SiLU and GELU are real-analytic)**.** *Let $m \in \mathbb{N}$. Define the coordinatewise maps*

$$\mathrm{SiLU}_m(\mathbf{x}) := \big(\mathrm{SiLU}(\mathbf{x}_1), \ldots, \mathrm{SiLU}(\mathbf{x}_m)\big)^\top, \quad \mathrm{GELU}_m(\mathbf{x}) := \big(\mathrm{GELU}(\mathbf{x}_1), \ldots, \mathrm{GELU}(\mathbf{x}_m)\big)^\top,$$

*for $\mathbf{x} \in \mathbb{R}^m$, where SiLU and GELU are as in Proposition F.2 and Proposition F.4. Then both $\mathrm{SiLU}_m$ and $\mathrm{GELU}_m$ are real-analytic maps $\mathbb{R}^m \to \mathbb{R}^m$.*

*Proof.* Each scalar component $\mathbf{x} \mapsto \mathrm{SiLU}(\mathbf{x}_i)$ (resp. $\mathrm{GELU}(\mathbf{x}_i)$) is the composition of the projection onto coordinate $i$ (a linear map) with the real-analytic scalar function SiLU (resp. GELU). By Proposition A.2, each component is real-analytic. Therefore, by Definition A.1, the vector-valued maps $\mathrm{SiLU}_m$ and $\mathrm{GELU}_m$ are real-analytic. $\qquad\square$

**Proposition F.6** (GLU-style blocks are real-analytic)**.** *Let $d_{\mathrm{in}}, d_{\mathrm{hid}} \in \mathbb{N}$ and consider affine maps*

$$A_1(\mathbf{x}) = \mathbf{W}_1 \mathbf{x} + \mathbf{b}_1, \qquad A_2(\mathbf{x}) = \mathbf{W}_2 \mathbf{x} + \mathbf{b}_2,$$

*with $\mathbf{W}_1, \mathbf{W}_2 \in \mathbb{R}^{d_{\mathrm{hid}} \times d_{\mathrm{in}}}$ and $\mathbf{b}_1, \mathbf{b}_2 \in \mathbb{R}^{d_{\mathrm{hid}}}$. Let $\phi : \mathbb{R}^{d_{\mathrm{hid}}} \to \mathbb{R}^{d_{\mathrm{hid}}}$ be either $\mathrm{SiLU}_{d_{\mathrm{hid}}}$ or $\mathrm{GELU}_{d_{\mathrm{hid}}}$ from Proposition F.5. Define the GLU-style block*

$$\mathrm{GLU}_\phi(\mathbf{x}) \;:=\; A_1(\mathbf{x}) \odot \phi\big(A_2(\mathbf{x})\big), \qquad \mathbf{x} \in \mathbb{R}^{d_{\mathrm{in}}},$$

*where $\odot$ denotes the Hadamard product.*

*Then $\mathrm{GLU}_\phi : \mathbb{R}^{d_{\mathrm{in}}} \to \mathbb{R}^{d_{\mathrm{hid}}}$ is real-analytic. In particular:*

- *Taking $\phi = \mathrm{SiLU}_{d_{\mathrm{hid}}}$ recovers SwiGLU, which is real-analytic.*

- *Taking $\phi = \mathrm{GELU}_{d_{\mathrm{hid}}}$ recovers GeGLU, which is real-analytic.*

*Proof.* Each affine map $A_j$ is real-analytic as a matrix product plus addition (Proposition A.10, Proposition A.1). By Proposition F.5, $\phi$ is real-analytic, so $\mathbf{x} \mapsto \phi(A_2(\mathbf{x}))$ is a composition of real-analytic maps (Proposition A.2), hence real-analytic. The map $\mathbf{x} \mapsto A_1(\mathbf{x}) \odot \phi(A_2(\mathbf{x}))$ is a Hadamard product of two real-analytic vector-valued functions; componentwise this is just the product of real-analytic scalars, so it is real-analytic by Proposition A.1 (equivalently, by Proposition A.11). Thus $\mathrm{GLU}_\phi$ is real-analytic. The SwiGLU and GeGLU cases follow by choosing $\phi$ accordingly. $\square$

**Relation to universal-approximation and expressivity results.** The material above concerns only the analyticity of the non-linearities used in our analysis. For completeness, we also record here how our injectivity theorem fits alongside existing expressivity results for Transformers; this discussion is logically independent of the real-analyticity assumptions.

Classical expressivity results for Transformers are primarily *existential*. Universal-approximation theorems (e.g. Yun et al. (2020); Sun & Yang (2020)) show that for any continuous sequence-to-sequence function $f$ on a compact domain and any $\varepsilon > 0$, there exists a Transformer with suitable depth and width whose outputs are within $\varepsilon$ of those of $f$. Turing-completeness results for encoder–decoder Transformers (e.g. Pérez et al., 2019) similarly establish the existence of parameter settings that simulate any Turing machine. Taken together, these works characterise what the architecture can represent *in principle*: they do not model random initialization or gradient-based training, and they are not formulated in our discrete setting with finite context length, fixed decoder-only architecture, and real-analytic activations.

Our results are complementary and instead concern what happens *typically* under standard training. We fix a concrete decoder-only architecture and a finite prompt set, and study the map from prompts to last-token representations. In this setting we prove that (i) for any fixed architecture, the set of parameters for which this map is non-injective has Lebesgue measure zero, and (ii) gradient-based training from standard random initializations preserves absolute continuity of the parameter distribution and therefore almost surely avoids this "collision set". Non-injective Transformers certainly exist (we explicitly construct such failure cases in Section 2), but our results show that they form a thin subset that typical optimization trajectories do not reach.

Our contribution is thus orthogonal to prior expressivity theory. We do *not* claim that Transformers can only represent injective functions. Rather, within the specific regime we study (decoder-only, real-analytic activations, cross-entropy loss, GD-type training from standard initialization), we show that the resulting last-token map is injective with probability one over initialization and training. In short, classical expressivity results describe what is mathematically *possible* for the Transformer function class, while our analysis characterizes what is *almost surely implemented* when that class is explored via standard training procedures.

