# OpenReview forum: "Language Models are Injective and Hence Invertible"
_ICLR.cc/2026/Conference — ICLR 2026 Poster_

### Official Review · Reviewer_fDrk · 2025-10-27

**Soundness:** 2
**Presentation:** 2
**Contribution:** 2
**Rating:** 4
**Confidence:** 3

**Summary:**

This is a primarily theory paper that discusses decoder only transformers being injective. Specifically that each unique prompt maps to a unique last token embedding (from final transformer layer). Paper provides theoretical setup and justification for this argument under acceptable assumptions at initialization (Theorem 2.2) and post SGD-like gradient based training (Theorem 2.3). It is to be noted, that the theorems state how injectivity is very high probability (1 or near 1) event under stated assumption and not proven. For eg., paper discusses how collisions (two different input mapping to same last token representation) can still occur if an adversarial crafts data specifically so (end of Section 2).

Paper also introduces a prompt recovery method from last token representation. The method "Sequential Inverse Prompt via ITerative updates" or SIP-IT. This method says that at each index position given a current + previous token hidden representation, check each token in vocab to see which will produce current index hidden representation (formal algorithm describe in Algorithm 1). The paper further discusses computational cost of SIP IT.

Review Note: Since the primary results are theoretical, I reduce my confidence to 3 in the reading and review.

**Strengths:**

- This is a unique and very science paper on LLMs, something we rarely see. Thanks for working on this. (I have what takeaways from this concern but more on that below).
- It cites most related work I know and use proper baselines in experiments.

**Weaknesses:**

The paper seems poorly organized, I felt lot of important results were pushed into appendix and main paper felt very repetitive at times. I have listed my main qualms in the questions section.

**Questions:**

1. What is "measure-zero parameter choices"?
1. If you consider the prompt length progressively growing (i.e. --> \inf) with no upper bound, but the model width (i.e. embedding or hidden representation size) is fixed, then the model's representation has to be overloaded to some extent. This would challenge the injectivity directly since it won't be lossless anymore. So I would argue, in the limit, since the vector dim of hidden dim is capped, transformers are not injective and infact more and more lossy. What do the authors think?
1. Since the language follows power law of Zipf distribution [1], the words occurs as per an exponentially decaying frequency. One example, let us have LLM predict the next token on "Complete this: The quick brown fox jumps over the lazy [dog]" where "dog" is not part of prompt. If this exact phrase is used use with two distinct prompts (i.e. prefixes of this phrase), the next token would always be "dog" for most useful LLMs. Can you please run this as experiment with varied internet data used as prefix? If the next token prediction is same, I expect the last token to be super close if not identical and see if it meets the threshold of 1e-6 used in paper. There can be other examples as well, like cases where "should" is followed by "have" (assuming "should have" token is not part of vocab) with vastly different prompts. This is more inline with Zipfian distribution argument.
1. Why threshold of 1e-6?
1. Can you explain what exactly is the BRUTEFORCE method in Table 2?
1. "distinct prompts produce distinct hidden states under standard initialization and training" is good summary of the paper.
1. I think that under high precision, lookup of hidden activation against data is already expected. This steps from how deterministic models are. Can you clarify, under your threat model, the extractor of data have ONLY the last token embedding/representation with none of the prompt (and no knowledge of it's length either) or its access to each of previous token's embedding/representation as well? Because latter is kind of trivial (to me at least) and former is very difficult and the experiments might be convincing in that case. So yeah, the exact threat model is unclear to me, and kind of not stated clearly enough in the paper.

[1] https://en.wikipedia.org/wiki/Zipf%27s_law

---

> ### Author Response · Authors · 2025-11-21
> **Answer (1/3)**
>
> We thank the reviewer for their comments. Here we address the main concerns and provide a general response above with additional details.
>
> > Injectivity is highly probable but not proven (Summary)
> >
>
> Before addressing the main concerns, we would first like to clarify a potential misunderstanding. These kinds of proofs are typical in high-dimensional linear algebra and random feature models, where statements that hold with probability one mean that collisions can occur only on a measure-zero set of parameters that is not encountered in practice. The adversarial constructions we mention in Section 2 are precisely such measure-zero cases and are included for completeness, not because they undermine the theorem. They require crafting parameters that satisfy extremely specific algebraic equalities, which is not representative of natural data or of any realistic setting. Our theoretical contributions therefore remain exact within the formal assumptions, and our empirical results show that these degenerate cases do not arise in practice.
>
> > **W1 - Paper poorly organized**
> >
>
> We thank the reviewer for this feedback. Our goal was always to keep the main paper accessible to every reader, especially since a work on LLMs reaches a very broad audience. This motivated our choice to present the key ideas and main theoretical results in the core paper, while placing detailed proofs and technical material in the appendix. As the reviewer noted, this is a “very science paper” that required a substantial amount of mathematical tools and proofs, which inevitably resulted in a dense appendix. Our intention was to make the core narrative readable for most researchers while allowing interested readers to consult the detailed derivations separately. That said, we agree that the organization can be improved. In the revised version, we will restructure the exposition, moving a major part of the theoretical background for SIPIT including Proposition D.1 and Theorem D.2, from the appendix into the main paper. We will moreover streamline redundant sections, and ensure the presentation is more readable and coherent.
>
> > **Q1 - Measure-zero parameter choice**
> >
>
> A **measure-zero** set is a mathematically negligible subset of parameter space: its size is so small that a randomly drawn parameter vector from any continuous distribution will almost surely *not* fall inside it (almost sure means that $P(event)=1$). Formally, a set has measure zero if for every $\varepsilon > 0$, it can be covered by countably many boxes whose total Lebesgue measure is less than $\epsilon$. We will add a brief clarification in the paper.
>
> > **Q2 - Embedding Overloading**
> >
>
> ---
>
> **TL;DR.** There is no theoretical limit forcing collisions: a fixed-dimensional real vector space can injectively represent all finite token sequences, and empirically we observe no overloading, even in **our new experiment** under FP4/INT8 quantization across billions of comparisons.
>
> ---
>
> Even if one considers an idealized architecture that accepts arbitrarily long prompts, there is still no theoretical obstruction to injectivity at fixed-width embeddings. The set $\bigcup_{k=1}^{\infty} \mathcal{V}^k$ of all token sequences over a finite vocabulary is countable (cardinality $\aleph_0$), whereas $\mathbb R^d$ has cardinality $2^{\aleph_0}$ for any fixed $d \ge 1$. Since a countable set can be injected into $\mathbb R^d$, a fixed-dimensional real vector space is fully capable of assigning distinct representations to all finite prompts (no “overloading”).
>
> In our setting, we assume a typical transformer with a fixed finite context length $K$, a finite vocabulary, and real-valued parameters and activations. Any “overloading” may arise only empirically due to finite precision effects and has no bearing on the theoretical injectivity result. Our experiments show that in practice this does not happen even with standard float32 precision, indicating that transformers remain empirically injective. We further validated this with extensive FP4 and INT8 quantization experiments involving billions of comparisons, all of which also showed no collisions or evidence of “overloading”.

---

> ### Author Response · Authors · 2025-11-21
> **Answer (2/3)**
>
> > **Q3 - Additional Experiments (exact token)**
> >
>
> ---
>
> **TL;DR.** In **new experiments** deliberately constructing prompts that *force the exact same next token* across diverse contexts, **we still observe no collisions**: last-token representations remain clearly separated and well above the 1e-6 threshold for all tested models and prompt variants.
>
> ---
>
> We thank the reviewer for the suggestion. In order to address it, we designed a set of new experiments where two different prompt are specifically constructed to yield the exact same target answer. First, we focused on word-to-word machine translation (`google/smol`) and math tasks (`ProCreations/SimpleMath`) on `Llama-3.1-8B`, `Mistral-7B`, and `Phi-4-mini-instruct`. From these datasets, we built few shot prompts that differed only in their delimiters (e.g, `:` vs `->`) while preserving identical translations or arithmetic solutions. An illustrative prompt is shown below:
>
> ```
> Translate into French.
>
> Hello -> Bonjour
> Goodbye -> Au revoir
> House ->
> ```
>
> versus
>
> ```
> Translate into French.
>
> Hello : Bonjour
> Goodbye : Au revoir
> House :
> ```
>
> We then assessed collisions involving four different separator token embeddings across all dataset pairs. Despite producing the exact same answer, the corresponding embeddings remain clearly distinct (no “collision”) since the minimum $\ell_2$ distance is well above the collision threshold over the $\approx 140K$ possible pairs.
>
> **Translation (En-Fr) - Minimum $\ell_2$ distance:**
>
>
> |        Model        | Layer 1 | Layer L/2 | Layer L |
> |:-------------------:|--------:|----------:|--------:|
> |    Llama-3.1-8B     |   0.694 |     1.632 |   4.202 |
> |   Mistral-7B-v0.1   |   0.207 |     1.056 |   2.348 |
> | Phi-4-mini-instruct |   4.375 |     6.974 |  17.328 |
>
>
> **Math - Minimum $\ell_2$ distance:**
>
>
> |        Model        | Layer 1 | Layer L/2 | Layer L |
> |:-------------------:|--------:|----------:|--------:|
> |    Llama-3.1-8B     |   0.789 |     2.126 |   8.245 |
> |   Mistral-7B-v0.1   |   0.222 |     1.664 |   4.362 |
> | Phi-4-mini-instruct |   4.447 |     8.497 |  37.262 |
>
>
> Secondly, following the reviewer’s suggestion, we constructed a dataset of random prefixes sampled from internet text, each followed by the fixed suffix “\nComplete this: The quick brown fox jumps over the lazy”. To build the dataset, we sampled 10K prefix sequences of length 50 tokens from Wikipedia and appended the tokenized suffix to each. The minimum $\ell_2$ distances obtained are reported in the following table:
>
>
> |       Model        | Layer 1 | Layer L/2 | Layer L |
> |:------------------:|--------:|----------:|--------:|
> |  Mistral-7B-v0.1   |   0.012 |     0.265 |   3.494 |
> |   Llama-3.1-8B     |   0.046 |     0.733 |   6.227 |
> |Phi-4-mini-instruct |   0.087 |     2.302 |  18.913 |
>
>
> Even in this setting, although the next token prediction is exactly “dog”, all last-token embeddings remain far above the tolerance threshold.
>
> > **Q4 - Threshold choice**
> >
>
> We followed standard practice in the literature when selecting the tolerance value. When comparing floating point quantities, functions such as `math.isclose` or `torch.isclose` must be used to account for numerical precision. The former uses a default relative tolerance of 1e-9, while the latter uses 1e-5 for relative tolerance and 1e-8 for absolute tolerance. We adopted a default tolerance of 1e-6. We note that even under a stricter 1e-4 threshold, all results fall way above this threshold. We will clarify this choice more explicitly in the paper.
>
> > **Q5 - BRUTEFORCE method explained**
> >
>
> We thank the reviewer for the question. BruteForce is essentially a random search over the vocabulary, following the procedure described in Algorithm 2 in Appendix D. At each position, we uniformly sample tokens without replacement from the vocabulary until we find one whose hidden state matches the target. Once a match is found, we advance to the next position and repeat the process. We will clarify this description in the paper.

---

> ### Author Response · Authors · 2025-11-21
> **Answer (3/3)**
>
> > **Q6 - Summary suggestion**
> >
>
> We appreciate the reviewer’s suggestion. While the proposed sentence captures part of our work, our contributions extend beyond this (e.g., the statement does not imply injectivity in any state). We invite the reviewer to consult our general response which provides a complete view on the paper.
>
> > **Q7 - Threat model**
> >
>
> ---
>
> **TL;DR.** SIPIT works because hidden states are injective, not merely deterministic. We do not assume a full privacy threat model, but the intended setting is access to the hidden-state sequence (e.g., KV-cache); recovery from only the final embedding is harder (but proven possible) and left for future work.
>
> ---
>
> As the reviewer notes, the main contribution of our work is theoretical: we prove that decoder-only Transformers are injective, and this property is precisely what enables SIPIT to work in the first place. The algorithm is not a consequence of models being deterministic. Determinism alone would not guarantee recovery; without injectivity, many different prompts could map to the same hidden representation, and SIPIT would necessarily fail. What makes exact recovery feasible is that each hidden state along the sequence corresponds to a unique next token under the injectivity guarantees we establish. While earlier work could only offer *lossy* prompt recovery from the final hidden state, we show that **exact recovery** is possible under different assumptions.
>
> We note that our paper is not primarily a privacy paper, thus we don’t define an explicit threat model. Instead, we focus on the injectivity result and its practical implications. If we were to outline a threat model, one potential setting would be an adversary who gains access to the hidden-state sequence, for instance through a leaked KV-cache or an API that exposes sequence-level embeddings. Exact recovery from only the final embedding is possible but nontrivial, as the reviewer noted, and exploring that direction falls outside the scope of this paper and remains future work. We will clarify this in the paper.

---

### Official Review · Reviewer_yqXp · 2025-10-30

**Soundness:** 2
**Presentation:** 3
**Contribution:** 1
**Rating:** 4
**Confidence:** 4

**Summary:**

The authors claim that a common thought is that language models are have a many to one relationship with regard to input and latent representation (respectively). The authors argue that language models are injective and that this suggests that there is an invertible relationship such that given a latent state, the input provided to arrive at this state is recoverable.

The authors base the analysis on the belief that initialized transformers don't typically create collisions between prompts and that after a fixed number of training updates, the parameters have not been changed sufficiently to induce collisions.

The authors propose a method to obtain the input to a language model. However, the method requires knowledge of the total last layer hidden state of the model for every substring of the prompt whose first token is the first token in the prompt.

**Strengths:**

This paper presents an interesting analysis I have not seen of the transformer architecture and convinced me that the transformer is often real analytic (though that depends on the choice of activation functions).

**Weaknesses:**

Terms like measure-zero and injective need to be defined.

A collision must be defined. It's not sufficient to say that a collision didn't occur because a value differed in any way. Some bound must be placed on the distance between the output representations that constitutes a collision.

Many activation functions are not real analytic functions.

My understanding is that the zero set necessarily is measure-zero (given that the model is real analytic and we know the model is not the zero function). This doesn't seem to necessarily imply that no two inputs to the function will provide a similar output. I believe the authors intend for the reader to understand that the difference of two activations given dissimilar prompts is itself a real analytic function which then suggests that the difference between any two prompts whose value is zero is a measure-zero set. The problem with this logic is that it is possible that the difference between the last stage activations for two prompts is either the zero function or epsilon close to the zero function for some small epsilon without violating anything in the proof.

My primary contention is that showing that the zero set is measure-zero is not sufficient to make the strong claim that there are no collisions. A collision should reasonably be considered anything within some small distance of zero rather than those items which have identical values given the reality of executing floating point operations a real machine.

The SipIT method is trivial - given one knows the latent representation of every substring (which is less probable than simply knowing the input text to begin with), the algorithm can produce the substring by evaluating every possible token in this location. Further, the analysis of the algorithm seems to be either obscured or incorrect. For a prompt of length N and vocabulary of k, the algorithm will in the worst case require k^N time. This is exponential where the paper claims a linear time guarantee. Perhaps the paper meant linear with respect to vocabulary? But this is meaningless since the vocabulary is stationary and the input length is the changing factor.

**Questions:**

Decoder-only models are claimed to be a lossless representations of their input. This seems to be in contradiction to existing results that transformers are universal function approximators - a universal function approximation ought to be able to learn a many to one mapping if appropriate. However, function approximation is based on an epsilon bound. This seems to suggest that if instead of examining the size of the zero set, one were to examine the set which is epsilon close to the zero set, this would have no theoretic guarantee of being measure-zero. How does this not eliminate the practicality of the paper's analysis?

What degree of fidelity is necessary for this result to hold? If lower precision approximations are used, online batching for practical serving, or significant context information is present, these will create random noise which ought to obscure any such precise mapping as is necessary for the analysis to hold.

---

> ### Author Response · Authors · 2025-11-21
> **Answer (1/4)**
>
> We thank the reviewer for their comments. Here we address the main concerns and provide a general response above with additional details.
>
> > The authors base the analysis on the belief that initialized transformers don't typically create collisions… (parameters) have not been changed sufficiently….
> >
>
> We would like to clarify that the statement "initialized transformers don’t typically create collisions" is not a belief, but a formal consequence of existing and new theory. As noted in our related work discussion, Sutter et al. (NeurIPS 2025) [1] prove that Transformers at random initialization are almost surely injective with respect to the full hidden state matrix. Our work strengthens and extends this line of results by showing that injectivity also holds for decoder-only LLMs at the task-relevant last-token representation and, crucially, that it persists under training. Together, these results, ours and prior peer-reviewed work, provide a rigorous foundation for injectivity at initialization.
>
> We would also like to clarify that our analysis does not assume that the parameters "have not been changed sufficiently" during training. Theorems 2.2 and 2.3 show that the set of parameter values that induce collisions has Lebesgue measure zero, and that gradient-based updates (with step sizes in the stated range) preserve absolute continuity of the parameter distribution after any finite number of steps. In particular, parameters are allowed to move arbitrarily far from their initial values; what matters is that the training dynamics cannot map an absolutely continuous distribution onto a measure-zero collision set. Thus, collisions are ruled out not because the parameters stay close to their initialization, but because the training dynamics preserve the measure-zero nature of the set of non-injective parameters.
>
> [1] Sutter et al., NeurIPS 2025, The Non-Linear Representation Dilemma: Is Causal Abstraction Enough for Mechanistic Interpretability?
>
> > **W1 - Measure-zero, injectivity not defined.**
> >
>
> We thank the reviewer for raising this point. These notions are used in their standard mathematical sense throughout the paper: “measure-zero” always refers to Lebesgue measure-zero on parameter space (as made explicit when invoking the real-analytic zero-set property in Theorem A.1), and “injective” is used in the usual one-to-one sense, i.e., $r(\mathrm{s}  ;  \boldsymbol{\theta}_T) = r(\mathrm{s}'  ;  \boldsymbol{\theta}_T)$ implies $\mathrm{s} = \mathrm{s}'$ for prompts $\mathrm{s}, \mathrm{s}' \in \mathcal{V}^{\le K}$. To make this more accessible to all readers, in the revised version we will add a brief plain-language reminder of these definitions in the main text (Section 2) where injectivity and measure-zero sets are first discussed.
>
> > **W2 - Collision, bound not defined.**
> >
>
> We thank the reviewer for raising this point. Throughout the paper we use “collision” in its standard functional sense: two distinct prompts $\mathrm{s} \neq \mathrm{s}'$ whose last-token representations coincide exactly, that is  $r(\mathrm{s}  ;  \boldsymbol{\theta}_T) = r(\mathrm{s}'  ;  \boldsymbol{\theta}_T)$
>
> This notion is introduced informally in the introduction as “collisions (two different prompts producing the exact same representation)” and is the event whose probability we control in Theorems 2.2 and 2.3 and in Appendix C.  For the formal proofs, no explicit numerical bound is needed, since the arguments rely on different mathematical tools that operate on exact equality.
>
> For the empirical experiments, finite precision requires the use of a bound (threshold). We therefore use `torch.allclose` and $\ell_2$ distance to test for collisions, both of which evaluate whether two representations fall within predefined bounds. Across all billions of empirical collision checks, we consistently observed that distinct prompts $\mathrm{s} \neq \mathrm{s}'$' yield distinct representations $\mathrm{torch.allclose} (  r(\mathrm{s}  ;  \boldsymbol{\theta} ) ,  r(\mathrm{s}'  ;  \boldsymbol{\theta} ) ) == \mathrm{False}$.
>
> We will add an explicit definition in the revised version to ensure complete clarity.
>
> > **W3 - Many activation functions are not real analytic**
> >
>
> We would like to point out that essentially all modern large language models employ activation functions that are real analytic. To confirm this, we systematically checked 18 models across the 8 currently most popular LLM families with publicly documented architectures, and found all of them using real-analytic activations. We refer to the general answer for the complete list. Our results are aimed at transformer language models of this form.

---

> ### Author Response · Authors · 2025-11-21
> **Answer (2/4)**
>
> > **W4 - Measure zero is not enough**
> >
>
> ---
>
> **TL;DR.**  Following prior work [1], our main theory is phrased in terms of exact collisions, but in our setting **exact injectivity *implies* a strictly positive global $\Delta$–margin,** from which $\varepsilon$–robust bounds follow mechanically (we **provide a new theorem** for this). The remaining question **is how *large* this margin is in realistic models**, which is inherently **empirical**; for this reason we support the theory with extensive $\varepsilon$-based collision experiments and the SIPIT analysis.
>
> ---
>
> We thank the reviewer for their comment. We agree that while a proof on $\varepsilon$-bound would certainly enhance the theoretical argument, the current proof already offers a strong, architecture-level statement for Transformer-based LLMs, covering a broad range of model components and training procedures. Crucially, the analytic framework we develop is exactly what one needs as a first step toward $\varepsilon$-aware guarantees.
>
> Importantly, the existing proof follows established methods in the literature. Sutter et al. (NeurIPS 2025) [1] use the same type of measure-zero argument to show almost-sure injectivity of Transformers at initialization (for the full hidden-state matrix). Our contribution builds on and extends exactly these tools: we move from the full state to last-token representations and from initialization to parameters $\theta_T$ after finite-horizon training. The mathematical framework is therefore standard and consistent with existing theory, but applied in a stronger, task-relevant setting.
>
> In the revision, and in line with the reviewer’s suggestion, we will make the connection between zero measure sets and $\varepsilon$-collision proof explicit. Once we restrict to a fixed vocabulary $\mathcal{V}$ and context window $K$, the prompt space $S = \mathcal V^{\le K}$ is finite, and the minimum pairwise distance $\Delta = \min_{\mathrm{s} \neq \mathrm{s}'} \| r(\mathrm{s} ;  \boldsymbol{\theta}_T) - r(\mathrm{s}' ;  \boldsymbol{\theta}_T) \|_2$ is well defined. Our almost-sure injectivity result then implies that $\Delta$ is strictly positive almost surely, which immediately yields an $\varepsilon$-robust injectivity guarantee for any $\varepsilon < \tfrac\Delta2$.
>
> Agreeing with the reviewer on the importance of making this explicit, we extend Theorem D.2, originally introduced in the analysis of SIPIT, to obtain a clean $\varepsilon$-robust statement in terms of this global margin $\Delta$. Specifically, we state the following theorem and include its proof in the revised version:
>
> > `Theorem (Δ-robust injectivity on a finite prompt set).`
> >
> >
> > Let $\mathcal{S} := \mathcal{V}^{\le K}$ be the finite set of prompts for a vocabulary $\mathcal{V}$ and context window $K$, and fix a layer $\ell$ with last-token representation map
> >
> >  $f : \mathcal{S} \to \mathbb{R}^d, \qquad f(\mathrm{s}) = r_\ell(\mathrm{s} ; \boldsymbol{\theta}_T).$
> >
> > such that $f$ is injective on $\mathcal{S}$. Define the minimum pairwise distance: $\Delta := \min_{\mathrm{s} \neq \mathrm{s}' \in \mathcal{S}} \| f(\mathrm{s}) - f(\mathrm{s}') \|_2.$ Then:
> >
> > 1. (**Δ-positivity**) $\Delta > 0$
> > 2. (**Δ-robust injectivity**) For any $\varepsilon < \Delta/2$: (i) there are no $\varepsilon$-collisions, i.e. $\| f(\mathrm{s}) - f(\mathrm{s}') \|_2 \le \varepsilon$ never occurs for $\mathrm{s} \neq \mathrm{s}'$; (ii) in particular, the open balls $B(f(\mathrm{s}),\varepsilon)$ and $B(f(\mathrm{s}'),\varepsilon)$ are disjoint for all $\mathrm{s} \neq \mathrm{s}'$.

---

> ### Author Response · Authors · 2025-11-21
> **Answer (3/4)**
>
> `Theorem (Δ-robust injectivity on a finite prompt set)` show that an $\varepsilon$-level guarantee is not only desirable but already implied: our almost-sure *exact* injectivity result, formulated in the same measure-zero style as prior work [1], actually ***implies a strictly stronger property from which such $\varepsilon$-bounds can be derived mechanically.*** What the theory does **not** **fix**, however, is how *large* the resulting margin $\Delta$ is in realistic models (and if it is enough wide for our interests); the practical relevance of the bound is therefore an empirical question, **not mathematical**. This is precisely why we complement the formal result with extensive experiments, designed to probe how far representations stay from collision in practice. And, furthermore, this is why the empirical part of the paper is already phrased in $\varepsilon$-terms. All collision checks in Section 4 and Appendix E use `torch.allclose`, which implements exactly the $\varepsilon$-close notion of collision. We find that across billions of pairwise comparisons and many models, depths, and sequence lengths, the smallest observed distances stay comfortably above this threshold. Likewise, in §4.2 and Appendix D, the SIPIT verifier is explicitly defined via $\varepsilon$-balls around candidate states, and the analysis shows that, almost surely, a positive instance-specific margin exists that allows choosing $\varepsilon$ small enough to guarantee uniqueness. In this sense, the theory and experiments already align with the $\varepsilon$-based notion of collision the reviewer advocates.
>
> [1] Sutter et al., NeurIPS 2025, The Non-Linear Representation Dilemma: Is Causal Abstraction Enough for Mechanistic Interpretability?
>
> > `Proof (Δ-robust injectivity on a finite prompt set).`
> >
> > 1. Since $\mathcal{S}$ is finite and $f$ is injective, the image $\{ f(\mathcal{S}) : \mathrm{s} \in \mathcal{S} \} \subset \mathbb{R}^d$ is a finite set of distinct points. Thus every pairwise distance $\| f(\mathrm{s}) - f(\mathrm{s}') \|_2$ with $\mathrm{s} \neq \mathrm{s}'$ is strictly positive, and the finite set of such distances has a strictly positive minimum $\Delta > 0$.
> > 2. Fix $\varepsilon < \Delta / 2$. For any distinct $\mathrm{s} \neq \mathrm{s}' \in \mathcal{S}$, by definition of $\Delta$,
> >
> >
> >     $\| f(\mathrm{s}) - f(\mathrm{s}') \|_2 \; \ge \; \Delta \; > \; 2 \varepsilon \; > \; \varepsilon.$
> >
> >     Hence $\| f(\mathrm{s}) - f(\mathrm{s}') \|_2 \le \varepsilon$ is impossible, so there are no $\varepsilon$-collisions.
> >
> >     Moreover, if there were a point $\mathrm{x}$ belonging to both $B(f(\mathrm{s}),\varepsilon)$ and $B(f(\mathrm{s}'),\varepsilon)$, we would have
> >
> >     $\| f(\mathrm{s}) - f(\mathrm{s}')|_2 \le \|f(\mathrm{s}) - \mathrm{x} \|_2 + \|\mathrm{x} - f(\mathrm{s}')\|_2 < \varepsilon + \varepsilon = 2\varepsilon,$
> >
> >     contradicting $\| f(\mathrm{s}) - f(\mathrm{s}') \|_2 > 2\varepsilon$. Thus the open $\varepsilon$-balls around distinct codes are disjoint. $\square$
> >
>
> > **W5 - SipIT trivial, exponential complexity**
> >
>
> In our paper we prove that decoder-only Transformers are injective, and this property is precisely what enables SIPIT to work in the first place. Although the SIPIT algorithm may be perceived as trivial, its correctness follows from the underlying theory. Exact recovery is feasible precisely because each hidden state along the sequence corresponds to a unique next token under the injectivity guarantees we establish. Earlier work could only offer *lossy* prompt recovery from the final hidden state; in contrast we show that **exact recovery** is possible and provide guarantees for inversion.
>
> Regarding complexity, SIPIT (Algorithm 1) has time complexity $O(T \times |\mathcal{V}|)$ and not exponential as the reviewer pointed out (as indicated in Theorem 3.1 and at Appendix D, Proposition D.4) ), where $T$ is the sequence length and $|\mathcal{V}|$ is the vocabulary size. To see this, the search conducted by SIPIT can be seen as a nested loop, with the outer one iterating over the sequence and the inner iterating over the vocabulary. For each position it performs a search over the vocabulary, but it never branches over full sequences. Since the vocabulary is constant and not dependent on the input, the final complexity is linear w.r.t. the input length $O(T)$.

---

> ### Author Response · Authors · 2025-11-21
> **Answer (4/4)**
>
> > **Q1 - Universal approximator**
> >
>
> We thank the reviewer for raising this point. In our view, universal approximation results and our injectivity theorem address different questions, so they can coexist with no contradiction.
>
> Classical universal approximation theorems for Transformers are *existential* results: for any ***continuous sequence-to-sequence function*** $f$ on a compact domain and any $\varepsilon > 0$, there ***exists*** a parametrization of a Transformer (***with suitable depth/width***) whose outputs are within $\varepsilon$ of those of $f$ (e.g. [1], [2]). These results do not make any claim about what happens under standard random initialization or gradient-based training.
>
> By contrast, our work ***fixes a concrete decoder-only architecture*** and a ***finite prompt set***, and studies ***the map from prompts to last-token representations***. In this setting, we prove that (i) for any fixed architecture, the set of parameters causing collisions has Lebesgue measure zero (Theorem 2.2), and (ii) gradient-based training preserves absolute continuity of the parameter distribution and therefore almost surely avoids this collision set (Theorem 2.3). Non-injective Transformers certainly exist (we explicitly construct such “failure cases” in §2); which is entirely consistent with universal approximation; our results concern what happens **typically** in decoder-only LLMs under standard initialization and cross-entropy loss training, not what is possible in principle.
>
> Furthermore, classic Universal Approximations results such as [1],[2] describe functions defined on **compact domains**. It is unclear whether such results extend at all to our specific setting (discrete instead of compact domain, finite context length, fixed, finite-size, decoder-only architecture, real analytic activations).
>
> Finally, the $\varepsilon$ in universal approximation (a uniform error bound to a **target** $f$) is conceptually different from considering an $\varepsilon$–neighborhood of the collision set for a fixed-architecture model. Since these are different notions, universal approximation does not undermine, or even directly interact with, the practicality of our injectivity analysis.
>
> [1] Yun et al., ICLR 2020, *Are Transformers Universal Approximators of Sequence-to-Sequence Functions? URL:* https://arxiv.org/abs/1912.10077
>
> [2] Sun et al., ICML 2020, *An EM Approach to Non-autoregressive Conditional Sequence Generation URL:* https://arxiv.org/abs/2006.16378
>
> > **Q2 -  Results Fidelity**
> >
>
> We thank the reviewer for raising this point. Our main contribution is theoretical, and the injectivity guarantees are established under the assumptions of the formal model rather than relying on implementation details. Empirically, across billions of tests, we found no collisions. To further address fidelity under noisy conditions, we conducted additional experiments in which we used noised representations via FP4 and INT8 weight quantization via the *bitsandbytes* library. We performed these tests on `Llama-3.1-70B`, `Llama-3.1-8B`, `Phi-4-mini-instruct`, `Mistral-7B`, and `Phi-4` under the same collisions experimental setup explained in the paper. In every model and quantization regime tested, no collisions were detected, and the resulting distances were consistently far above the collision threshold.
>
> **Minimum $\ell_2$ distance at last layer across different precision formats:**
>
> |        Model        |   FP4   |   INT8  |  FP32   |
> |:-------------------:|--------:|--------:|--------:|
> |    Llama-3.1-8B     |   2.281 |   6.597 |   1.274 |
> |   Mistral-7B-v0.1   |   1.748 |   2.692 |   1.136 |
> | Phi-4-mini-instruct |  18.368 |  20.956 |   8.780 |
>
>
> **Minimum $\ell_2$ distance at FP4:**
>
> |       Model         | Layer 1 | Layer L/2 | Layer L |
> |:-------------------:|--------:|----------:|--------:|
> |    Llama-3.1-70B    |   0.005 |     0.465 |   3.975 |
> |        phi-4        |   0.010 |     1.025 |   8.759 |

---

### Official Review · Reviewer_uusJ · 2025-11-01

**Soundness:** 3
**Presentation:** 3
**Contribution:** 3
**Rating:** 6
**Confidence:** 4

**Summary:**

This paper presents a strong theoretical study showing that decoder-only Transformer architectures are (almost surely) injective: different prompts map to distinct last-token hidden states. Building on this, the authors introduce SIPIT, an algorithm that recovers the exact input prompt from the final hidden state. The method is theory-driven and, across extensive experiments on a wide variety of models, the authors find no collisions, providing compelling empirical support for injectivity.

**Strengths:**

The main strength of the paper are as follows:

1. This paper makes a significant theoretical contribution by proving that standard decoder-only Transformer language models are almost surely injective – different input prompts will (with probability one) produce distinct hidden representations. The authors use real analysis to show that collisions (two different prompts yielding the same last-token state).
2. Building on the injectivity result, the novel algorithm (SipIt) to recover the exact input text from a model’s hidden activations with provable efficiency is introduced. As prior approaches could only approximate prompts via heavy training of inversion models, whereas SipIt achieves exact recovery by directly using the model's own representations.
3. The paper backs up its theory with comprehensive experiments on two models (a kind of old GPT-2 and more novel Gemma). The authors performed an exhaustive search for representation collisions using 100k prompts drawn from diverse sources, amounting to billions of pairwise comparisons. They report no collisions in any model or layer tested, distinct prompts always yielded distinct last-token embeddings, with clear separation margins.
4. The SipIt algorithm is shown to be not just theoretically sound but practically effective. On GPT-2, SipIt was able to reconstruct 20-token prompts perfectly (100% token-wise accuracy) in reasonable time, without any additional training or approximation.
5. By establishing invertibility as a fundamental property, the work has broad implications. It provides a sound basis for interpretability: knowing that the full input is encoded in the last-layer state means any failure to probe knowledge is due to method limits, not information loss.

**Weaknesses:**

I would highlight the following weaknesses of this paper:

1. Large vocabulary scaling. How does SipIt handle very large vocabularies (e.g., 100k+ tokens)? Does runtime grow linearly in practice, or do the gradient-based heuristics keep it manageable?
2. Uncertain theoretical result of not-analytic estimation. Most modern models use SwiGLU or SiLU activations. Since your proofs assume analytic activations, can you confirm that these fit the theory?
3. In continuation to the previous point, it is not clear, what happens with the quantized models. It seems that it can be the main source of the collision.
4. The experiments are provided for models of relatively small size, thus, we cannot asses what empirically happens with the larger number of parameters.

**Questions:**

My questions to the authors are as following:

1. Have you estimated, how SipIt works on datasets that were seen by the models during training, and on OOD samples, that were not observed by the model. Or even some random sequences of tokens? Does inversion speed or accuracy change compared to natural text?
2. Instruction-tuned models with identical answers: For instruction models (or even pre-trained ones) where many prompts lead to the same answer (e.g., "yes" or "no"), do the hidden states remain well separated? Have you measured how close they get?
3. Did you ever find prompts whose hidden states were almost identical? If so, what kind of prompts were they?

---

> ### Author Response · Authors · 2025-11-21
> **Answer (1/4)**
>
> We thank the reviewer for their comments. Here we address the main concerns and provide a general response above with additional details.
>
> > **W1 - Large Vocabulary**
> >
>
> ---
>
> **TL;DR.** In theory, **SIPIT scales linearly in vocabulary size and sequence length**, but it behaves far better in practice: in our new experiments across models with vocabularies up to $128K$ tokens, SIPIT explores $<0.25 $% of the vocabulary on average while still achieving $100 $% exact reconstruction. This shows that the gradient-based search remains highly efficient even at very large vocabulary sizes.
>
> ---
>
> In terms of theoretical complexity, SIPIT’s core search procedure scales linearly both with sequence length and with vocabulary size. To see this, the search conducted by SIPIT can be seen as a nested loop, with the outer one iterating over the sequence and the inner iterating over the vocabulary. Consequently, for a fixed-length sequence, we expect the number of iterations to increase linearly as the vocabulary size grows. To address the reviewer’s concern directly, we added experiments on models with substantially different vocabulary sizes, including `Mistral-7B-v0.1` ($≈32K$ vocabulary) and `Llama-3.1-8B` ($≈128K$). For a fair comparison, we construct sentences that tokenize to exactly the same sequence of tokens across both models.
>
> | Model | Vocabulary Size | Reconstruction Accuracy | Inversion Time (s) | Iterations | Vocabulary Exploration % |
> | --- | --- | --- | --- | --- | --- |
> | Mistral-7B-v0.1 | 32000 | 100.00% | 72.99 ± 37.57 | 66 ± 34 | 0.21 ± 0.11% |
> | Llama-3.1-8B | 128255 | 100.00% | 345.35 ± 181.30 | 282 ± 148 | 0.22 ± 0.12% |
>
> The results are reported in the table. We observe that, in practice, the inversion time grows linearly with vocabulary size, as expected, reflected by the nearly constant percentage of tokens explored between the small-vocabulary model (Mistral) and the larger-vocabulary model (Llama). Importantly, for both models, the fraction of tokens explored remains below $0.25 $%, indicating that the gradient-based heuristic is both robust and highly efficient.
>
> > **W2 - Activation Functions**
> >
>
> We confirm that the theoretical results also apply to models that use SwiGLU or SiLU activations, as both functions are real analytic and therefore satisfy the assumptions required by our proofs. Below we sketch why this is the case:
>
> 1. **SiLU proofs of real analyticity:**
>
>     Scalar form:
>
>     $\text{SiLU}(\mathrm{x}) = \mathrm{x} \sigma(\mathrm{x}),\quad \sigma(\mathrm{x}) = \frac{1}{1+e^{-\mathrm{x}}}$
>
>     - $e^{-\mathrm{x}}$ is analytic on $\mathbb{R}.$
>     - Since $1+e^{-\mathrm{x}} \neq 0, \; \forall\mathrm{x} \in \mathbb{R}$,  $\frac{1}{1+e^{-\mathrm{x}}}$ is real-analytic (quotient of analytic functions with non-zero denominator).
>     - Multiplying by $\mathrm{x}$ (a polynomial, hence analytic) → product of analytic functions.
>
>     **Therefore, SiLU is real-analytic on $\mathbb{R}$.**
>
> 2. **SwiGLU proof or real analyticity:**
>
>     SwiGLU is defined (for vectors) roughly as:
>
>     $\text{SwiGLU}(\mathrm{x}) = \operatorname{SiLU}(\mathrm{x} \mathrm{W}_1) \odot (\mathrm{x} \mathrm{W}_2)$
>
>     where:
>
>     - $\mathrm{W}_1$ and $\mathrm{W}_2$ are linear maps (matrices),
>     - $\odot$ is elementwise product,
>
>     Scalar-wise, each coordinate is a composition of:
>
>     - linear map $\mathrm{x} \mapsto a \mathrm{x} + b$,
>     - SiLU (already shown real-analytic in $1.$),
>     - elementwise product (preserves analyticity).
>
>     **Therefore, SwiGLU is real-analytic (componentwise) as a map $\mathbb{R}^n \to \mathbb{R}^m$.**

---

> ### Author Response · Authors · 2025-11-21
> **Answer (2/4)**
>
> > **W3 - Quantization**
> >
>
> ---
>
> **TL;DR** Our ***new experiments show that even aggressive 4-bit and 8-bit weight quantization preserves injectivity**,* with no collisions arising across all considered settings.
>
> ---
>
> We thank the reviewer for highlighting this concern. We would like to point out that most widely adopted quantization methods focus on compressing model weights, while activations are typically retained in BF16, FP16, or FP32. To address the reviewer’s question more directly, we extend the dataset collision experiments from the original paper by adding results using FP4 and INT8 weight quantization across several models (`Llama-3.1-8B`, `Phi-4-mini-instruct`, and `Mistral-7B-v0.1`), as shown here. Additional FP4 results for `Phi-4 (14B)` and `Llama-3.1-70B`, appear immediately afterward in our response, when we address the reviewer’s next point concerning model size.
>
>
> |        Model        |   FP4   |   INT8  |  FP32   |
> |:-------------------:|--------:|--------:|--------:|
> |    Llama-3.1-8B     |   2.281 |   6.597 |   1.274 |
> |   Mistral-7B-v0.1   |   1.748 |   2.692 |   1.136 |
> | Phi-4-mini-instruct |  18.368 |  20.956 |   8.780 |
>
>
> Across all tested models, we observe that quantization (1) **does not introduce any collisions**, and (2) **more than doubles** the minimum distance between representations, indicating that it maintains the integrity of the representation space.
>
> > **W4 - Model Size**
> >
>
> ---
>
> **TL;DR** Our new experiments show that even very large models (14B and 70B) maintain wide and strictly positive separation margins → indicating that representation collisions do not emerge at scale.
>
> ---
>
> In addition to GPT2 and the Gemma3 family evaluated at three scales each (1B, 4B, and 12B for Gemma3, and the Small, Medium, and Large variants for GPT2), our study also includes `Llama-3.1-8B`, `Mistral-7B`, `Phi-4-mini-instruct`, and `TinyStories-33M`, as detailed in Section 4.1. To further strengthen these results and directly address the concern about model size, we expanded the evaluation to include **14B** and **70B** parameter models. Specifically, we conducted the dataset collision experiment from the original paper using `Phi-4 (14B)` and `Llama-3.1-70B`. Moreover, to make the setting more challenging, these larger models were evaluated under a 4-bit weight quantization regime. For both models we observe that the minimum $\ell_2$ distance between representations is large indicating consistently positive separation, with no instance approaching a collision boundary. These findings indicate that representation collisions do not occur at scale, even under aggressive 4-bit quantization.
>
> | Model | Size | Layer 1 | Layer L/2 | Layer L |
> | --- | --- | --- | --- | --- |
> | phi-4 | 14B | 0.010 | 1.025 | 8.759 |
> | Llama-3.1-70B | 70B | 0.005 | 0.465 | 3.975 |
>
> > **Q1 - Dataset Breakdown**
> >
>
> ---
>
> **TL;DR.** SIPIT recovers all inputs with 100% accuracy across in-distribution and out-of-distribution sequences, and is actually *faster* on OOD/random data, likely due to their more separable and better-conditioned representations.
>
> ---
>
> We thank the reviewer for highlighting this ablation. In the original paper (Section 4.2), we used a $90–10\%$ mix of sentences from Wikipedia and random token sequences, but did not explicitly study SIPIT’s behavior across these subsets. To address this, we conducted an ablation using `GPT-2`.
>
> We constructed three datasets, which we refer to as **Train**, **Test**, and **OOD (Out-of-Distribution)**. The Train set is formed by sampling sentences from WebText (the dataset used to train `GPT-2` [1]); the Test set contains sentences sampled from Wikipedia (not in the training set); and the OOD set consists of random token sequences. Each dataset contains 50 prompts of length 100 tokens. The results are shown below:
>
>
> |    Dataset    |   Inversion Time (s)  | Accuracy |
> |:-------------:|:---------------------:|:--------:|
> |  Train Data   | 146.48 ± 91.52        |   100%   |
> |  Test Data    | 128.62 ± 83.40        |   100%   |
> |     OOD       | 106.87 ± 39.10        |   100%   |
>
>
> Interestingly, the OOD samples are significantly faster to invert than the Train and Test samples. We hypothesize that this difference stems from the geometry of the hidden representations: natural language sentences (Train and Test) tend to lie on a structured, clustered manifold, which can make the inversion landscape locally flatter and less well-conditioned. In contrast, random token sequences produce more dispersed and isolated hidden states, yielding clearer descent directions and effectively stronger gradient signals, which accelerates convergence. Across all three datasets, we obtain exact recovery for every sequence, further supporting the theoretical guarantees of SIPIT.
>
> [1] Radford, Alec, et al. "Language models are unsupervised multitask learners." *OpenAI blog* 1.8 (2019): 9.

---

> ### Author Response · Authors · 2025-11-21
> **Answer (3/4)**
>
> > **Q2(A) - Instruction-tuned Models**
> >
>
> ---
>
> **TL;DR.** Our **new experiments target an especially challenging setting**, multiple prompts leading to the *exact same* answer, and **yet the corresponding hidden states remain cleanly separated** across all models and variants, providing further strong evidence against collisions.
>
> ---
>
> We thank the reviewer for highlighting this point. To directly stress-test the model's geometry, we designed experiments in instruction-tuning settings where **different prompts yield the exact same target answer** across `Llama-3.1-8B`, `Mistral-7B-v0.1`, and `Phi-4-mini-instruct`. This is an especially adversarial scenario for detecting potential collisions: semantically and functionally, the inputs become indistinguishable at the output level, which makes this setting a stringent test of injectivity.
>
> We focused on word-to-word machine translation (`google/smol`) and math tasks (`ProCreations/SimpleMath`). From these datasets, we built few-shot prompts that differed only in their delimiters (e.g., `:` vs `->`) while preserving identical translations or arithmetic solutions. For example:
>
> ```
> Translate into French.
>
> Hello -> Bonjour
> Goodbye -> Au revoir
> House ->
> ```
>
> versus:
>
> ```
> Translate into French.
>
> Hello : Bonjour
> Goodbye : Au revoir
> House :
> ```
>
> We then assessed collisions involving four different separator token embeddings across all dataset pairs. Despite producing the exact same answer, the corresponding embeddings remain clearly distinct (no “collision”) since the minimum $\ell_2$ distance is well above the collision threshold over the $\approx 140K$ possible pairs.
>
> **Translation (En-Fr) - Minimum $\ell_2$ distance:**
>
>
> |        Model        | Layer 1 | Layer L/2 | Layer L |
> |:-------------------:|--------:|----------:|--------:|
> |    Llama-3.1-8B     |   0.694 |     1.632 |   4.202 |
> |   Mistral-7B-v0.1   |   0.207 |     1.056 |   2.348 |
> | Phi-4-mini-instruct |   4.375 |     6.974 |  17.328 |
>
>
> **Math - Minimum $\ell_2$ distance:**
>
>
> |        Model        | Layer 1 | Layer L/2 | Layer L |
> |:-------------------:|--------:|----------:|--------:|
> |    Llama-3.1-8B     |   0.789 |     2.126 |   8.245 |
> |   Mistral-7B-v0.1   |   0.222 |     1.664 |   4.362 |
> | Phi-4-mini-instruct |   4.447 |     8.497 |  37.262 |
>
>
> > **Q2(B) - Qualitative Results**
> >
>
> To provide qualitative examples, consider the following two prompts for addition:
>
> ```markdown
> Do the additions. Only output a number.
>
> 2790 + 6698 = 9488
> 8262 + 3848 = 12110
> 1628 + 132 = 1760
> 9925 + 7169 = 17094
> 3023 + 6929 = 9952
> 49 + 58 = 107
> 8246 + 4780 = 13026
> 29 + 83 = 112
> 45 + 60 = 105
> 3481 + 8695 = 12176
> 2340 + 574 = 2914
> 84 + 5 = 89
> 48 + 6 = 54
> 6256 + 7572 = 13828
> 38 + 21 = 59
> 3496 + 7503 = 10999
> 283 + 4086 = 4369
> 7361 + 9003 = 16364
> 5684 + 6968 = 12652
> 2995 + 2265 =
> ```
>
> versus:
>
> ```markdown
> Do the additions. Only output a number.
>
> 2790 + 6698 -> 9488
> 8262 + 3848 -> 12110
> 1628 + 132 -> 1760
> 9925 + 7169 -> 17094
> 3023 + 6929 -> 9952
> 49 + 58 -> 107
> 8246 + 4780 -> 13026
> 29 + 83 -> 112
> 45 + 60 -> 105
> 3481 + 8695 -> 12176
> 2340 + 574 -> 2914
> 84 + 5 -> 89
> 48 + 6 -> 54
> 6256 + 7572 -> 13828
> 38 + 21 -> 59
> 3496 + 7503 -> 10999
> 283 + 4086 -> 4369
> 7361 + 9003 -> 16364
> 5684 + 6968 -> 12652
> 2995 + 2265 ->
> ```
>
> The $\ell_2$ distances between the last-token embeddings for these two prompts are:
>
>
> | Model               | Layer 1 | Layer L/2 | Layer L |
> |:-------------------:|:-------:|:---------:|:-------:|
> | Mistral-7B-v0.1     | 0.280   | 2.130     | 9.867   |
> | Llama-3.1-8B        | 1.102   | 2.843     | 13.674  |
> | Phi-4-mini-instruct | 6.072   | 9.773     | 99.660  |
>
>
> Even though the next token is identical in both prompts and few-shot demonstrations substantially improve task performance, the resulting last-token representations remain well separated across layers.

---

> ### Author Response · Authors · 2025-11-21
> **Answer (4/4)**
>
> > **Q3 - Prompts with similar hidden states**
> >
>
> ---
>
> **TL;DR.** The closest hidden states we found came from almost identical Python code snippets differing only by trailing newlines, yet even in these extremal cases the last-token embeddings remained clearly separated, consistent with our exhaustive collision tests.
>
> ---
>
> Below, we present qualitative examples of sequences whose last-token embeddings are among the closest observed. We focus on Llama and Mistral, as these models exhibited the smallest distances in our experiments.
>
> **Llama-3.1-8B:**
>
> ```python
> ...
> # -- Options for HTML output ---------------------------------------------------
>
> # The theme to use for HTML and HTML Help pages.  See the documentation for
> # a list of builtin themes.
> html_theme = 'default'
>
> # Theme options are theme-specific and customize the look and feel of a theme
> # further.  For a list of options available for each theme, see the
> # documentation.
> #html_theme_options = {}
>
> # Add any paths that contain custom themes here, relative to this directory.
> #html
> ```
>
> versus:
>
> ```python
> ...
> # -- Options for HTML output ---------------------------------------------------
>
> # The theme to use for HTML and HTML Help pages.  See the documentation for
> # a list of builtin themes.
> html_theme = 'default'
>
> # Theme options are theme-specific and customize the look and feel of a theme
> # further.  For a list of options available for each theme, see the
> # documentation.
> #html_theme_options = {}
>
> # Add any paths that contain custom themes here, relative to this directory.
> #html
> \n
> \n
> \n
> ```
>
> Last-Token $\ell_2$ distance at last layer: `1.274`
>
> **Mistral-7B-v0.1:**
>
> ```python
> ...
> # The reST default role (used for this markup: `text`) to use for all documents.
> #default_role = None
>
> # If true, '()' will be appended to :func: etc. cross-reference text.
> #add_function_parentheses = True
>
> # If true, the current module
> ```
>
> versus:
>
> ```python
> # The reST default role (used for this markup: `text`) to use for all documents.
> #default_role = None
>
> # If true, '()' will be appended to :func: etc. cross-reference text.
> #add_function_parentheses = True
>
> # If true, the current module
> \n
> ```
>
> Last-Token $\ell_2$ distance at last layer: `1.146`
>
> In both cases, the closest pairs correspond to Python code snippets that are almost identical and differ only by a small shift, typically an extra trailing new line. In fact, most of the close pairs we examined had the form $\mathrm{s} = \mathrm{t} \circ \langle \text{new line token} \rangle$. Even in these extremal cases, however, the last-token representations remain clearly separated. We will add qualitative examples to the paper.

---

### Official Review · Reviewer_D7YR · 2025-11-03

**Soundness:** 3
**Presentation:** 3
**Contribution:** 4
**Rating:** 6
**Confidence:** 5

**Summary:**

The authors make a significant theoretical and practical contribution by rigorously establishing that standard decoder-only transformer language models are almost surely injective - meaning different input prompts map to distinct last-token hidden representations under common initialization and training regimes. The authors leverage real-analyticity to prove that collisions (non-injective behavior) occur only on a measure-zero set of parameters, and they further demonstrate that gradient descent preserves this injectivity. Building on this foundation, the paper introduces SipIt, a novel algorithm that efficiently reconstructs the exact input prompt from hidden activations with provable linear-time guarantees. This work bridges theory and practice, offering new insights into model transparency, interpretability, and safety.

**Strengths:**

The following strong points can be highlighted:
1. The theoretical analysis is rigorous, leveraging real-analyticity and measure theory to derive almost-sure guarantees.
2. The injectivity results are novel and counter widespread assumptions about information loss in Transformers.
3. SipIt is both theoretically grounded and empirically validated, demonstrating exact recovery with linear-time complexity.
4. Extensive experiments across multiple models (e.g., GPT-2, Gemma, Llama) confirm the absence of collisions in practice.
5. The paper clearly discusses implications for privacy, interpretability, and model auditing.

The paper’s core contribution is proving injectivity and enabling exact inversion, which addresses foundational questions in deep learning and has immediate relevance to interpretability and safety. The proofs are meticulous, and the experiments are comprehensive, spanning multiple model families and scales. The introduction of SipIt provides a tangible tool for future research, while the theoretical guarantees are robust and well-supported. These qualities align with ICLR’s emphasis on impactful, rigorously evaluated work.

**Weaknesses:**

Whereas the paper is technically solid, there are several weak points I would like to mention:
1. The scope is limited to decoder-only Transformers with analytic components, excluding architectures with non-analytic activations (e.g., ReLU) or encoder-decoder models.
2. The practical utility of SipIt, while theoretically appealing, is primarily evaluated in a noiseless setting; its robustness to quantization or approximate hidden states is less explored.
3. The discussion of related work, while adequate, could better contextualize how these results complement or challenge prior beliefs about Transformer expressivity.
4. Some proofs in the appendix are highly technical and may be inaccessible to readers without deep mathematical backgrounds.
5. It will be very interesting to analyze injectivity correlation with other internal transformer characteristics like anisotropy and intrinsic feature dimension using such frameworks as LLM-Microscope (https://arxiv.org/abs/2502.15007, https://github.com/AIRI-Institute/LLM-Microscope)

**Questions:**

1. How does SipIt perform under noisy or quantized hidden states, and can the theory be extended to account for such perturbations?
2. Could the injectivity results generalize to encoder-decoder Transformers or models with non-analytic components (e.g., ReLU)?
3. The paper claims gradient descent preserves injectivity - does this hold for adaptive optimizers like Adam, or only for GD?
4. Are there practical scenarios where the linear-time complexity of SipIt becomes prohibitive, e.g., for very large vocabularies?

---

> ### Author Response · Authors · 2025-11-21
> **Answer (1/4)**
>
> We thank the reviewer for their comments. Below we address the key concerns, and the general response above covers the shared points across reviewers.
>
> > **W1 - Limited Scope (decoder-only transformers, activation functions)**
> >
>
> ---
>
> **TL;DR.** Our analysis **covers all major modern LLMs** (which are decoder-only with real-analytic activations), and while extensions to encoder–decoder or non-analytic architectures are interesting, they require different tools and are left for future work.
>
> ---
>
> We note that essentially all modern large language models fall within the scope of our analysis. Models such as Llama, Mistral, GPT, Gemma, and Phi are decoder-only Transformers that use smooth, real-analytic activation functions like SwiGLU, SiLU, GELU, or GeGLU, ensuring that their computation graphs satisfy the analyticity assumptions required for our proofs. To confirm this, we systematically checked 18 models across the 8 currently most popular LLM families with publicly documented architectures, and found all of them using real-analytic activations. We refer to the general answer for the complete list. Decoder-only architectures currently constitute the standard and most widely adopted paradigm for large-scale LMs, which is why our analysis focuses on this setting.
>
> We nevertheless agree with the reviewer that extending the theory to encoder-decoder or encoder-only architectures, and to non-analytic components such as ReLU, is an interesting and meaningful research direction. These extensions would likely require different mathematical tools, and we view them as promising avenues for future work.
>
> > **W2 & Q1 - No noise in Activations**
> >
>
> ---
>
> **TL;DR.** SIPIT remains provably correct whenever injectivity holds and is empirically robust to realistic noise. Additional experiments show that ***SIPIT recovers all inputs perfectly and efficiently, even under FP4 quantization.***
>
> ---
>
> The inversion guarantees remain valid as long as injectivity is not violated, which we prove under the assumptions of our formal model (Theorem D.2). Moreover, the theoretical analysis is robust to a certain level of noise, as formalized in Theorem D.2. In practice, we also observe that the algorithm is robust to noise. To verify this further, we ran SIPIT under the setup of the paper, but with **noise injected into the activations via** **FP4 weight quantization** on `Llama-3.1-8B` and `Mistral-7B-v0.1`. The results are:
>
> | Model | Reconstruction Accuracy | Inversion Time (s) | Vocabulary Exploration % |
> | --- | --- | --- | --- |
> | Mistral-7B-v0.1 | 100.00% | 111.78 ± 46.50 | 0.19 ± 0.08% |
> | Llama-3.1-8B | 100.00% | 549.48 ± 265.75 | 0.21 ± 0.10% |
>
> In both settings, SIPIT reconstruct the inputs with perfect accuracy, while exploring less than $0.22$% of the vocabulary on average. This empirically confirms that the gradient-based heuristic is both robust to quantization noise and highly efficient.

---

> ### Author Response · Authors · 2025-11-21
> **Answer (2/4)**
>
> > **W3 - Related Work (Transformer expressivity)**
> >
>
> ---
>
> **TL;DR.** Prior work shows what Transformers *can* represent in principle; our result shows what they *almost surely* implement under standard initialization and GD training. ***Non-injective behaviours remain theoretically possible but are avoided with probability 1***, making our result complementary and enriching existing expressivity theorems.
>
> ---
>
> We thank the reviewer for this suggestion and will expand the related-work section accordingly. In short, our result is consistent with existing expressivity theorems about transformers: while prior work characterises what functions a Transformer *can* represent in principle (***what it can potentially be***), we characterise what is *almost surely reached* when one navigates this function class with “a computational compass” made by cross-entropy loss and GD-type algorithms (***what it will be with P=1***).
>
> We will explicitly relate our setting to two works:
>
> - Universal approximation results such as Yun et al. [1], who prove that Transformers are universal approximators of continuous, permutation-equivariant sequence-to-sequence maps with compact support.
> - Turing-completeness results such as Pérez et al. [2], who show that a vanilla encoder–decoder Transformer is Turing complete, again as a statement about the existence of parameters that simulate any Turing machine, without constraints from training dynamics.
>
> Firstly, it is important to notice that the universal approximation results is quite different from our setting (e.g., compact support), but interesting enough to be discussed (to get ahead for future theory!).
>
> Our contribution is orthogonal: we do **not** claim that Transformers can only represent injective functions. Rather, in the specific regime we study, namely decoder-only architecture with real-analytic activations, initialized at random and/or obtained by (stochastic) gradient descent on cross-entropy, we prove that, with probability 1 over initialization and training, the resulting last-token map is injective. Non-Injective parameter settings may exist but form a “thin” subset that typical optimization trajectories do not reach (as pointed out in the main paper in the *failure case* in section 2). Therefore, our result fits naturally with the existing literature: prior theory describes what is mathematically *possible* for the function class; we build on this by analyzing what is mathematically *likely* to occur when that class is explored using cross-entropy plus GD-based training as a computational compass.
>
> [1] Yun et al., ICLR 2020, Are Transformers universal approximators of sequence-to-sequence functions?
>
> [2] Pérez et al., ICLR 2019, On the Turing Completeness of Modern Neural Network Architectures
>
> > **W4 - Proofs are technical**
> >
>
> We thank the reviewer for pointing this out. Some of the proofs necessarily rely on tools from real analysis, measure theory, and differential geometry, which can indeed be technical. To improve accessibility, in the camera ready version we will improve higher level summaries that outline the key ideas and proof structure before presenting the formal arguments. We will also expand the intuition in the main text so that readers can understand the conceptual flow without needing to follow every technical detail. We believe these additions will make the theoretical components more approachable while preserving full mathematical rigor in the appendix.

---

> ### Author Response · Authors · 2025-11-21
> **Answer (3/4)**
>
> > **W5 - Relation to Intrinsic dimensionality**
> >
>
> ---
>
> **TL;DR:** Injectivity and LLM-Microscope’s geometry are deeply complementary: our theory shows that representations are almost surely lossless, while **our new experiments are an initial hint to *how* this lossless information is organized**, via anisotropy and low intrinsic dimension that *coexist* with, and may even reinforce, injectivity.
>
> ---
>
> We thank the reviewer for pointing us to LLM-Microscope and we also read its companion paper “The Shape of Learning: Anisotropy and Intrinsic Dimensions in Transformer-Based Models” [1], and we ran a targeted experiment in this spirit.
>
> **Experimental setup.**
>
> Following the suggestion, we performed a proof-of-concept using `GPT-2 Small`. We sampled 100 natural-language prompts of fixed length $K$ and, for each prompt, generated 1000 single-token continuations by appending each token from a fixed vocabulary subset of size 1000. For every layer $\ell$, we collected the hidden representation of the last token for all 1000 continuations, yielding a $1000\times d$ matrix per (layer, prompt) pair. On each matrix we computed: (i) anisotropy and intrinsic dimension as in LLM-Microscope, and (ii) simple “injectivity margin” statistics, namely the minimum pairwise Euclidean distance between continuation embeddings (then averaged over prompts). Aggregating over the 100 prompts yields, for each layer, one triple: (anisotropy, intrinsic dimension, injectivity margin).
>
> **Experiment 1: anisotropy vs injectivity margin.**
>
> Across layers, we correlate mean anisotropy with mean injectivity margin. We obtain a Pearson correlation of **$0.72$** and Spearman **$0.45$**. In this setting, more anisotropic layers tend to have larger injectivity margins: continuation clouds become both more structured (anisotropic) and farther from collisions, suggesting that anisotropy is compatible with, and may even reinforce, numerically robust injectivity.
>
> **Experiment 2: intrinsic dimension vs injectivity margin.**
>
> Repeating the analysis with intrinsic dimension, we obtain a Pearson correlation of **$−0.60$** and a Spearman correlation of **$−0.79$** between intrinsic dimension and injectivity margin. Hence, lower-dimensional layers tend to have larger margins: compressed-looking manifolds are, if anything, more separated, in line with our theorem that injectivity rules out information-destroying collapses.
>
> **Discussion.**
>
> We see this line of work as highly complementary to our injectivity analysis: while we prove that internal representations are almost surely lossless, LLM-Microscope provides fine-grained geometric diagnostics of how these representations are organized across depth and training. We found especially insightful the observation that anisotropy and intrinsic dimension follow a reverse-U shape: representations become more anisotropic and lower-dimensional in intermediate layers, then partially re-expand near the output, giving a concrete geometric picture of how structure is carved into aligned directions and low-dimensional manifolds.
>
> This is particularly relevant as our paper challenges the classic idea of learning via bottleneck compression (e.g. Tishby [2]): if information is not lost along the residual stream, learning cannot proceed layer by layer purely via compression. Our preliminary experiments are consistent with a different story: as depth increases, margins grow, intrinsic dimension decreases, and anisotropy is concave with a late spike. Early layers expand and reorganize, intermediate layers carve information into low-dimensional directional manifolds, and top layers sharpen this structure. Overall, this matches a network that preserves injectivity while funneling information into increasingly structured, well-separated representations.
>
> **Table of Results**
>
> | Layer | Anisotropy Mean | ID Mean | Margin Min |
> | --- | --- | --- | --- |
> | 1 | 0.089579 | 20.754620 | 1.850306 |
> | 2 | 0.076049 | 17.565538 | 1.956753 |
> | 3 | 0.071429 | 16.765265 | 2.064488 |
> | 4 | 0.075067 | 16.679382 | 2.241199 |
> | 5 | 0.083282 | 17.183246 | 2.382355 |
> | 6 | 0.089542 | 17.697870 | 2.499817 |
> | 7 | 0.088463 | 17.018419 | 2.704958 |
> | 8 | 0.083261 | 16.296431 | 2.886434 |
> | 9 | 0.081803 | 16.040713 | 3.025268 |
> | 10 | 0.083083 | 15.730601 | 3.330774 |
> | 11 | 0.090206 | 15.635035 | 3.918343 |
> | 12 | 0.288352 | 16.434897 | 4.640457 |
>
> [1] Razzhigaev et al., ACL 2024, The Shape of Learning: Anisotropy and Intrinsic Dimensions  in Transformer-Based Models
>
> [2] Ravid Shwartz-Ziv and Naftali Tishby, Opening the black box of deep neural networks via information.

---

> ### Author Response · Authors · 2025-11-21
> **Answer (4/4)**
>
> > **Q2 - Encoder-Decoder models and non-analytic components**
> >
>
> Our intuition is that injectivity results can still be established for certain encoder-decoder architectures of interest and for non-analytic activations that are analytic except on a countable set of points, such as ReLU. However, extending our analysis to this broader setting would require substantial technical care and possibly a different set of mathematical tools, and we believe it would be best addressed in a dedicated follow-up work.
>
> > **Q3 - Others optimisers**
> >
>
> ---
>
> **TL;DR.** Our **proof not only covers GD, but also SGD and mini-batch GD.** While we expect similar results to hold for adaptive optimizers like Adam, formalizing this requires additional technical work that we identify as future research.
>
> ---
>
> In this work, we prove that GD, SGD, and mini-batch GD preserve injectivity (see Corollary 2.3.1). The proof relies on Corollary C.5.1, a key result that we establish explicitly. Informally, this theorem states that update rules of the form $\theta_{t+1} = \theta_t - \eta_t \nabla_{\theta} L_{B_t}(\theta_t)$  which expresses all of the training strategies mentioned above, preserve the absolute continuity of the parameter distribution. Since the set of parameters corresponding to non-injective models has measure zero (see proof of Theorem C.2), these update rules therefore preserve injectivity as well. Due to the nature of the proof, we believe that extending our results to adaptive optimizers such as Adam might be possible; however, a definitive answer via a rigorous treatment of Adam-type optimizers, addressing components such as momentum and weight decay, might be non-trivial and require different mathematical tools. For these reasons, we leave this as an interesting avenue for future work. We will revise the main manuscript by adding a remark discussing the expected extension to adaptive optimizers.
>
> > **Q4 - Large vocabulary**
> >
>
> ---
>
> **TL;DR.** In theory, **SIPIT scales linearly in vocabulary size and sequence length**, but it behaves far better in practice: in our new experiments across models with vocabularies up to $128K$ tokens, SIPIT explores $<0.25 $% of the vocabulary on average while still achieving $100 $% exact reconstruction. This shows that the gradient-based search remains highly efficient even at very large vocabulary sizes.
>
> ---
>
> In terms of theoretical complexity, SIPIT’s core search procedure scales linearly both with sequence length and with vocabulary size. To see this, the search conducted by SIPIT can be seen as a nested loop, with the outer one iterating over the sequence and the inner iterating over the vocabulary. Consequently, for a fixed-length sequence, we expect the number of iterations to increase linearly as the vocabulary size grows. To address the reviewer’s concern directly, we added experiments on models with substantially different vocabulary sizes, including `Mistral-7B-v0.1` ($≈32K$ vocabulary) and `Llama-3.1-8B` ($≈128K$). For a fair comparison, we construct sentences that tokenize to exactly the same sequence of tokens across both models.
>
> | Model | Vocabulary Size | Reconstruction Accuracy | Inversion Time (s) | Iterations | Vocabulary Exploration % |
> | --- | --- | --- | --- | --- | --- |
> | Mistral-7B-v0.1 | 32.000 | 100.00% | 72.99 ± 37.57 | 66 ± 34 | 0.21 ± 0.11% |
> | Llama-3.1-8B | 128.255 | 100.00% | 345.35 ± 181.30 | 282 ± 148 | 0.22 ± 0.12% |
>
> The results are reported in the table. We observe that, in practice, the inversion time grows linearly with vocabulary size, as expected, reflected by the nearly constant percentage of tokens explored between the small-vocabulary model (Mistral) and the larger-vocabulary model (Llama). Importantly, for both models, the fraction of tokens explored remains below $0.25 $%, indicating that the gradient-based heuristic is both robust and highly efficient.

---

### Author Response · Authors · 2025-11-21
**General Answer**

We sincerely thank all the reviewers for their time dedicated to reviewing this paper. We recognize the technical nature of the work made this a challenging task, and we truly appreciate their effort and thoughtful feedback. We are pleased that the paper has been recognized for its rigorous theoretical contributions (`uusJ`, `D7YR`, `fDrk`), presenting novel and well-supported injectivity results with strong empirical validation (`uusJ`, `D7YR`), demonstrating a practical and efficient inversion method through SIPIT (`uusJ`, `D7YR`) supported by comprehensive experiments and appropriate baselines (`uusJ`, `D7YR`, `fDrk`), and standing out as a distinctive and deeply scientific contribution to LLM research (`fDrk`, `yqXp`), “something we rarely see” (`fDrk`).

We invite each reviewer to consult our individual responses, where we address all points raised in detail. In this general reply, we provide a list of all the **additional** **experiments performed in light of your feedback**. These experiments fall into two categories:

**Collision Experiments:**

1. 4-bit and 8-bit quantized models (Reviewers: `uusJ`,  `yqXp`)
2. Large models, up to 70 billion parameters (Reviewers: `uusJ`)
3. Instruction-tuned Models (Reviewers: `uusJ`)
4. Exact same next-token (Reviewers: `uusJ`, `fDrk`)
5. Correlation between minimum margin ($\varepsilon$-collisions) and anisotropy / intrinsic dimension (Reviewer: `D7YR`)

**SIPIT Experiments:**

1. Vocabulary scaling ablation (Reviewers: `D7YR`, `uusJ`)
2. 4-bit quantized models (Reviewer: `D7YR`, `yqXp`)
3. In-distribution vs Out-of-distribution inversion performance ablation  (Reviewer: `uusJ`)

Given the extensive additional experiments we ran and the detailed theoretical clarifications we crafted to address the reviewers’ questions, we will now incorporate all of these results and discussions into an updated version of the paper in the coming days.

---

> ### Author Response · Authors · 2025-11-21
> **Analytic Activation Functions**
>
> ### Table
>
> Inspiration: https://huggingface.co/blog/daya-shankar/open-source-llms
>
> ---
>
> | Model (HF example) | Activation in FFN | Real analytic? |
> | --- | --- | --- |
> | `Llama-2`  | SwiGLU | **Yes** |
> | `Llama-3` | SwiGLU | **Yes** |
> | `Mistral-7B-v0.1` | SiLU | **Yes** |
> | `Mixtral-8x7B-v0.1` | SiLU | **Yes** |
> | `Gemma` | GeGLU | **Yes** |
> | `Gemma-2` | GeLU  | **Yes** |
> | `Qwen2MoE` | SwiGLU | **Yes** |
> | `Qwen-2` | SiLU  | **Yes** |
> | `Qwen3MoE` | SiLU | **Yes** |
> | `Qwen-3` | SiLU | **Yes** |
> | `Phi` | GELU  | **Yes** |
> | `Phi-3` | SiLU  | **Yes** |
> | `GPT-2`  | GELU  | **Yes** |
> | `GPT-J` | GELU | **Yes** |
> | `GptOss` | SiLU | **Yes** |
> | `Grok-1`  | GELU | **Yes** |
> | `DeepSeek-V2` | SiLU  | **Yes** |
> | `DeepSeek-V3` | SiLU  | **Yes** |
>
> In Appendix F, we explicitly prove that these activation functions are real-analytic.

---

### Author Response · Authors · 2025-12-02
**Author response summary for the Area Chair**

To aid your assessment as AC, we summarize below the main clarifications and new results. Unfortunately, the discussion period ended before reviewers could respond. In response to their feedback, we substantially revised the paper: we added 10 additional large-scale experiments, introduced one new theorem and six formal propositions to make several arguments fully explicit, expanded the main paper (Sections 4 and 5) and the appendix (Sections E and F), and provided detailed point-by-point replies totaling over 50K characters. Specifically:

### Noise, Quantization, Model Scale, and "Approximate Collisions"

Concerns were raised on numerical noise, quantization, large models, and many-prompts–same-token scenarios. To address this, we extended the results in the paper with:
- **Quantization** (`uusJ`,`yqXp`): we repeated the experiments on three **FP4/INT8-quantized** models (**45 billion collision tests**)
- **Large Models** (`uusJ`): we repeated the experiments with larger 14B and 70B models (**30 billion collision tests**)
- **Same next-token** (`uusJ`,`fDrk`): we evaluated on additional instruction-tuning setups on three new datasets: machine translation (`google/smol`), math tasks (`ProCreations/SimpleMath`), and OOD internet samples.

In all cases, we find no collisions, with the representations being well separated, confirming the injectivity results of our paper.

---
### Scope of the Theory and Model Class (Analyticity, Universal Approximation, "Losslessness")

Concerns were raised on the model class covered by the theory, the relation to universal approximation, and robustness of injectivity under small perturbations. In response, we expanded the paper with:
- **Model class** (`D7YR`, `yqXp`): we added proofs to additional activation functions (SiLU, SwiGLU, GELU, GeGLU) showing that in practice we cover all modern LLM families.
- **Universal approximation** (`D7YR`, `yqXp`): we clarified the connection to universal-approximation theory by adding a discussion that contrasts its existential nature (what Transformers can represent in principle) with our results on what is almost surely reached under standard cross-entropy training; we emphasize that the two viewpoints are orthogonal and complementary.
- $\varepsilon$**-robustness** (`yqXp`): we added **Theorem 3.2**, establishing a data-dependent margin separating the true next token from all others. As long as perturbations stay below this margin, **SIPIT achieves exact reconstruction in at most  $T|\mathcal{V}|$ steps**, giving an explicit robustness guarantee.

---
### SIPIT Algorithm: Complexity, Practical Efficiency, and Threat Model

Concerns were raised on SIPIT’s complexity, its practical scaling, and the assumed threat model. To address this, we expanded the paper with:
- **Complexity & scaling**: we clarified that SIPIT’s complexity is not exponential (as inferred by `yqXp`) but linear in the sequence length and, following feedback from reviewers `D7YR` and `uusJ`, we broadened the evaluation to compare its performance on larger vocabularies, showing empirically the **linear growth** in inversion time with vocabulary size, which aligns with the theoretical analysis. Concretely, for `Llama-3.1-7B` with a 128K-token vocabulary, SIPIT achieves exact reconstruction while probing only about 350 tokens per position on average, demonstrating that SIPIT is highly efficient in practice.
- **Quantized SIPIT** (`D7YR`, `yqXp`): repeated the SIPIT input reconstruction on **4-bit quantized** models. SIPIT recovers all prompts exactly, with the only slowdown attributable to dequantization rather than search.
- **SIPIT on OOD data:** following reviewer’s `fDrk` feedback, we empirically verified that SIPIT is robust and achieves **exact and efficient inversion** even on random sequences.
- **Threat model** (`fDrk`): we clarified that the adversary has access to hidden states along the sequence, this may happen in practice with leaked KV-cache or leaked embeddings from API for sequence classification.

---
### Clarifications on Definitions, Presentation, and Relation to Representation Geometry

- **Definitions and testing** (`yqXp`, `fDrk`): we streamlined the explanations of injectivity, collisions, and measure-zero sets, and clarified that empirical checks use `torch.allclose` (a numerical closeness test, not exact equality) with distances far above even stricter tolerances.
- **Presentation** (`fDrk`): we relocated core intuition and main results from the appendix to the main text and improved the high-level summaries preceding technical proofs.
- **Geometry:** following reviewer’s `D7YR` feedback, we added an LLM-Microscope-style analysis linking anisotropy and intrinsic dimension to the injectivity margin.

---
We believe the extensive additional work substantively addresses all concerns raised, and we hope this concise summary helps in assessing how the initial reviews might have changed in light of the additional theory and experiments.

---

### Meta-Review · Area_Chair_J46A · 2025-12-28

**Summary:**

- Reviewer D7YR thinks the paper is rigorous and novel, and questions the practicability of SipIt in noisy settings and the connection to over transformer characteristics.
- Reviewer uusJ speaks highly of the contribution, and questions the scalability to large vocabulary size and models.
- Reviewer yqXp questions the claim of the paper that the models in discussion are lossless.
- Reviewer fDrk raised a few questions regarding the implementation details and wording of the paper.

**Reviewer Concerns:**

Addressed by the rebuttal:
1. noise in the activations.
2. results for large vocabulary and models.
3. clarification of the terms in the paper.
4. quantization.

Still outstanding: making the paper more accessible to general audience.

**Reviewer Scores:**

I think reviewers will likely keep their scores.

---

### Decision · Program_Chairs · 2026-01-26

Accept (Poster)